# Gradient Correction in Federated Learning with Adaptive Optimization

## Abstract

In federated learning (FL), model training performance is strongly impacted by data heterogeneity across clients. Client-drift compensation methods have recently emerged as a solution to this issue, introducing correction terms into local model updates. To date, these methods have only been considered under stochastic gradient descent (SGD)-based model training, while modern FL frameworks also employ adaptive optimizers (e.g., Adam) for improved convergence. However, due to the complex interplay between first and second moments found in most adaptive optimization methods, naively injecting correction terms can lead to performance degradation in heterogeneous settings. In this work, we propose `FAdamGC`, the first algorithm to integrate drift compensation into adaptive federated optimization. The key idea of `FAdamGC` is injecting a pre-estimation correction term that aligns with the moment structure of adaptive methods. We provide a rigorous convergence analysis of our algorithm under non-convex settings, showing that `FAdamGC` results in better rate and milder assumptions than naively porting SGD-based correction algorithms into adaptive optimizers. Our experimental results demonstrate that `FAdamGC` consistently outperform existing methods in total communication and computation cost across varying levels of data heterogeneity, showing the efficacy of correcting gradient information in federated adaptive optimization.

## 1 Introduction

Federated Learning (FL) has emerged as a popular framework for collaboratively training machine learning models across decentralized clients (Li et al., 2020; Kairouz et al., 2021). Despite its privacy advantages, FL presents unique challenges due to statistical heterogeneity across client data and limited communication bandwidth. These issues often lead to degraded convergence rates and suboptimal global performance. While stochastic gradient descent (SGD) remains the default choice for local updates in FL, adaptive optimizers such as AdaGrad, RMSProp, and Adam (Duchi et al., 2011; Graves, 2013; Kingma, 2014) have demonstrated superior performance in centralized settings, with pronounced efficacy in large language model (LLM) training due to their robustness in handling complex loss landscapes (Zhang et al., 2024). This has motivated the extension of adaptive optimizers to FL as well (Cheng et al., 2023; Reddi et al., 2020; Wang et al., 2022), including for federated LLM training where they are widely adopted to cope with the scale and variability across clients. Nonetheless, the performance of adaptive methods still deteriorate under FL's non-i.i.d. data distributions, highlighting a pressing need for methods that explicitly address the interaction between adaptive optimization and data heterogeneity.

To address data heterogeneity in FL, client-drift compensation methods, such as `SCAFFOLD` (Karim-ireddy et al., 2020) and `Proxskip` (Mishchenko et al., 2022), have been proposed, primarily in conjunction with SGD-based updates. These methods maintain control variates to estimate and correct for the discrepancy between local (client-side) and global (server-side) gradients, mitigating client-drift and enhancing convergence robustness. Nevertheless, drift compensation methods have not yet been developed adaptive optimization settings such as Adam (Kingma, 2014). Motivated by this, in this work, we investigate the following questions:

1. *Will adaptive optimization algorithms designed using **client-drift compensation** obtain performance advantages across FL systems as found with their SGD counterparts?*

2. *What is the most effective way to incorporate compensation into adaptive federated optimization to mitigate data heterogeneity while ensuring **theoretical convergence guarantees**?*

**Key Challenges.** The core difficulty in answering these questions stems from the nonlinear structure of adaptive updates, which involve element-wise normalization using gradient history. Due to the interplay between first and second moments in most adaptive optimization methods, naively injecting correction terms as in the SGD case fails to account for this complexity and, as we will see, can even harm performance in heterogeneous regimes. Consequently, designing effective compensation strategies for tracking first-order information in adaptive methods remains an *open and important challenge* for improving robustness in general federated optimization frameworks.

**Our Contributions.** We investigate how to correctly compensate client-drift in adaptive federated optimization to ensure stable convergence under data heterogeneity. Based on our insights, we propose a novel algorithm leveraging the Adam optimizer that efficiently mitigates data heterogeneity by injecting *pre-estimation corrections*, i.e., prior to computing the moment terms. Through rigorous convergence analysis and experimental evaluations, we demonstrate that our algorithm effectively stabilizes the global learning process of FL with adaptive optimizers. In particular, our method demonstrably enhances resilience of FL training to the level of non-i.i.d. data distributions across clients, addressing a critical limitation of adaptive federated optimization techniques.

Our main contributions are as follows:

- We propose `FAdamGC`, an Adam-based federated optimization algorithm stabilized with a novel gradient correction mechanism. By leveraging control variables to track global gradient information and implementing a selective client tracking scheme to enhance communication efficiency, `FAdamGC` compensates for client drift internally without the need for extra fine-tuning, efficiently mitigating model biases caused by non-i.i.d. data distributions in FL (Sec. 4.2). Furthermore, our analysis provides both theoretical and empirical insights into difference across pre- and post-estimation correction strategies (Sec. 5).

- We conduct a rigorous convergence analysis of our proposed algorithm, producing both a convergence guarantee for specialized gradient normalization without relying on the bounded gradient assumption and also generalized convergence guarantee for adaptive federated optimization. Our analysis provides insights into the stability and convergence speedup achieved by `FAdamGC` under data heterogeneity, and clarifies the distinct impact of applying parameter tracking at different stages of the local update process (Sec. 5).

- We perform extensive experiments of `FAdamGC` across diverse datasets and multiple FL settings, including image classification tasks using CNNs and sequence classification tasks using LLMs. Our results demonstrate substantial improvements in training accuracy and resource utilization compared with baselines under varying levels of non-i.i.d. client data distributions (Sec. 6).

## 2 RELATED WORKS

**Client-Drift Compensation in FL.** Gradient Tracking (GT) methods (Di Lorenzo & Scutari, 2016; Nedic et al., 2017; Tian et al., 2018; Koloskova et al., 2021; Takezawa et al., 2022; Wang et al., 2024) have been proposed to address data heterogeneity challenges in decentralized optimization algorithms through the incorporation of drift corrections. The core principle of GT lies in tracking global gradient information during each communication round, ensuring more accurate gradient estimates across the system. Algorithms such as `SCAFFOLD` (Karimireddy et al., 2020) and `Proxskip` (Mishchenko et al., 2022) have been designed based on this concept for the conventional client-server FL setting. Multiple works in serverless FL have also showed performance improvement from GT (Ge & Chang, 2023; Berahas et al., 2023; Alghunaim, 2024) in both accuracy and resource efficiency. Furthermore, studies have shown that with GT, under proper initialization of correction variables, assumptions on data heterogeneity required in FL analysis can be relaxed.

Recent advancements have also extended correction methods to address hierarchical network structures. `SDGT` was introduced as the first GT algorithm tailored for semi-decentralized networks (Chen et al., 2024), bridging the gap between fully decentralized and centralized topologies. Meanwhile, (Fang et al., 2024) proposed `MTGC`, a multi-timescale GT algorithm incorporating hierarchical tracking terms in multi-tier networks. Despite these advancements, existing works on GT in FL have focused on SGD-based training, leaving the integration of GT with adaptive optimizers largely unexplored and an open challenge.

**Adaptive Optimizers.** SGD optimizers rely on fixed or decaying learning rates, which often require careful tuning and may struggle with scenarios involving sparse gradients or noisy updates. To

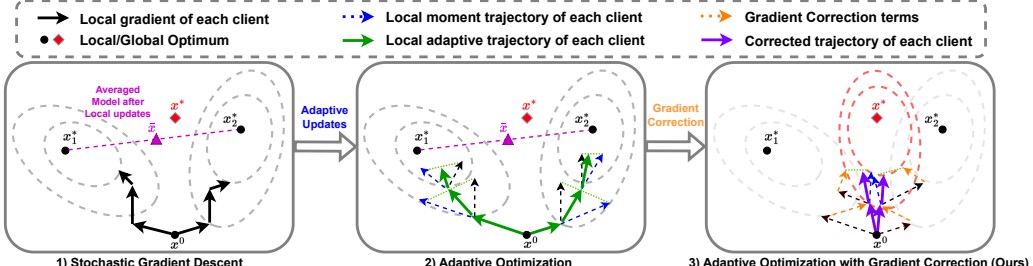

Figure 1: Visualization of the local update process under adaptive optimization with gradient correction. While adaptive methods help smooth the optimization trajectory, clients may still drift toward local optima due to data heterogeneity, preventing them from reaching globally optimal solutions even with federated cooperation. Gradient correction steers updates toward the global objective, mitigating client-drift to stabilize training. This combination blends the fast convergence of adaptive optimizers and the stability of correction-based methods.

address these limitations, adaptive optimizers dynamically adjust learning rates based on the gradient history, enabling more effective navigation of complex optimization landscapes. Among the most prominent adaptive optimizers are `AdaGrad` (Duchi et al., 2011) and `Adam` (Kingma, 2014). Recent advancements have further explored the decoupling of weight decay (Loshchilov, 2017) and the time-varying effects of regularization terms (Xie et al., 2024) in adaptive optimizers. More recently, novel optimizers such as Muon (Jordan et al., 2024), which introduce orthonormalized momentum matrices, have shown promising results in large-scale deep learning scenarios.

Several approaches have been proposed to integrate adaptive optimizers into FL. `FedAvg-M` and `SCAFFOLD-M` (Cheng et al., 2023) developed additional globally-tracked momentum terms to assist local updates. Methods like `FedAdam` and `FedAMS` employ an adaptive optimizer at the server to update the global model using aggregated client gradients (Reddi et al., 2020; Wang et al., 2022). On the other hand, Xie et al. (2019); Sun et al. (2023) incorporate adaptive optimization directly on local clients. Recently, Yan et al. (2025) proposed `PAdaMed`, an adaptive FL algorithm that employs gradient normalization to stabilize client updates and proves convergence without the bounded gradient assumption. While this work shares the high-level idea of using normalized gradients to control client behavior, it focuses on a simplified adaptive formulation without second-moment estimation.

## 3 BACKGROUND AND PRELIMINARIES

**System Model.** The problem we aim to solve follows the standard FL formulation:

$$\min_{x \in \mathbb{R}^d} f(x) = \frac{1}{n} \sum_{i=1}^{n} f_i(x), \tag{1}$$

where $n$ is the total number of clients (typically edge devices) in the system, indexed $i = 1, \ldots, n$. $f_i(x) = \mathbb{E}_{\xi_i \sim \mathcal{D}_i}[f_i(x; \xi_i)]$ is the empirical risk at client $i$ for model parameters $x \in \mathbb{R}^d$, where $\xi_i$ is an unbiased random sample drawn from the local empirical data distribution $\mathcal{D}_i$, constructed from the client's local training dataset. We assume the server is directly connected to each device as in the conventional FL architecture.

The training process operates on two distinct timescales: an outer timescale and an inner timescale. The outer timescale, denoted as $t = 1, 2, \ldots, T$, represents global aggregation rounds where the central server updates the global model. The inner timescale, denoted as $k = 1, \ldots, K$, represents local training steps performed by each client between global aggregations. We assume a fixed number of $K$ local updates occur between two consecutive global aggregation rounds.

For each global iteration $t$, the training procedure can be described in three iterative steps: (i) *Client Selection and Initialization*: At each global round $t$, the server selects a subset of clients $\mathcal{S}^t \subseteq \{1, \ldots, n\}$, where $|\mathcal{S}^t| = S \leq n$. The global model is broadcast to the selected clients to initialize local training. (ii) *Local Model Updates*: Each selected client performs $K$ local updates using a local optimizer, updating their local models based on their respective datasets. (iii) *Global Model Aggregation*: After completing $K$ local updates, the selected clients send their updated model parameters to the server. The server then aggregates these updates to refine the global model.

**Federated Adaptive Optimization.** The adaptive algorithm we focus in this work is Adaptive Moment Estimation (Adam), an optimization algorithm that combines the benefits of momentum and

adaptive learning rates (Kingma, 2014). At each iteration, Adam maintains an exponential moving average of the gradient (first moment) and the squared gradient (second moment). Given a stochastic gradient $g_i^{(t,k)}$ computed by client $i$ at step $t, k$, the update rules are:

$$m_i^{(t,k)} = \beta_1 m_i^{(t,k-1)} + (1 - \beta_1) g_i^{(t,k)} \qquad \text{(First moment)}$$

$$v_i^{(t,k)} = \beta_2 v_i^{(t,k-1)} + (1 - \beta_2) g_i^{(t,k)} \odot g_i^{(t,k)} \quad \text{(Second moment)}$$

$$x_i^{(t,k)} = x_i^{(t,k-1)} - \eta_l \frac{m_i^{(t,k-1)}}{\sqrt{v_i^{(t,k-1)} + \epsilon}}, \tag{2}$$

where $\beta_1, \beta_2 \in [0, 1)$ are decay rates, $\eta_l$ is the local learning rate, $\odot$ is the element-wise multiplcation, and $\epsilon$ is a small constant for numerical stability. The placement of adaptive optimizer in FL, on the server or the clients, has been a topic of ongoing debate (Sun et al., 2023). Prior work has shown that server-side adaptive methods, such as `FedAdam`, are more susceptible to gradient noise and tend to degrade as local updates $K$ increase. In contrast, approaches like `LocalAdam` (Sun et al., 2023), which apply Adam locally on clients and use averaging at the server, offer greater training robustness. Based on these findings, we adopt the design where adaptive optimizers are performed on clients, and the server applies averaging.

**Drift Compensation on SGD**. To address client-drift in SGD-based updates, `SCAFFOLD` employs control variates that adjust for local gradient discrepancies. Each client maintains a local correction term $y_i^t$, while the server maintains a global control variate $y^t$. The local update rule is:

$$x_i^{t,k+1} = x_i^{t,k} - \eta_l (g_i^{(t,k)} + y^t - y_i^t),$$

$$y_i^{t+1} = y_i^t - y^t + \frac{1}{\eta_l K}(x_i^t - x_i^{t,K}), \tag{3}$$

where $y_i^t$ and $y^t$ are the client and server control variates, respectively. These correction terms track the gradient differences between local and global objectives, mitigating the effect of client-drift.

# 4 DESIGN OF FAdamGC

## 4.1 MOTIVATION AND CHALLENGES

**Why is Drift Compensation Needed?** To motivate a more principled correction strategy for adaptive optimization in federated settings, we begin by considering the fixed-point solution $x^*$ that satisfies the global optimality condition $\nabla f(x^*) = 0$. For clarity, we also assume the moment estimates at convergence are zero, i.e., $m^* = v^* = 0$, consistent with typical Adam behavior under vanishing gradients. Under standard Adam-style updates, this fixed optimal point is not preserved when optimizing local functions $f_i$. Specifically, based on equation 2, the update rule $x^* \neq x^* - \eta_l \frac{\beta_1 m^* + (1-\beta_1)\nabla f_i(x^*)}{\sqrt{\beta_2 v^* + (1-\beta_2)\nabla f_i(x^*) \odot \nabla f_i(x^*) + \epsilon}}$ fails to satisfy the optimal point in general due to the fact that $\nabla f_i(x^*) \neq 0$ when $f_i$ differs from $f$. This misalignment arises from data heterogeneity across clients and leads to slower convergence or even divergence in non-IID settings. To address this challenge, various correction-based methods have been proposed to compensate for client-drift and stabilize training in SGD settings (Karimireddy et al., 2020; Chen et al., 2024; Fang et al., 2024). These techniques aim to align local updates by incorporating drift compensation, thereby restoring the fixed-point structure needed for consistent convergence across heterogeneous clients.

**Problem with Naive Application of Compensation.** A natural yet naive approach to track client-drift correction in adaptive federated optimization is to compute correction terms using the total model update $\frac{1}{\eta_l K}(x_i^{(t)} - x_i^{(t,K)})$ from all clients across the network. Similar to `SCAFFOLD`, the correction term $y_i^{(t+1)}$ averaged across all updates from $x_i^{(t,k)}$ from $k = 1$ to $K$, and this information is aggregated by the server to mitigate data heterogeneity across all clients in the next communication round $t + 1$. Specifically, given a local update direction $\Delta_i^{(t,k)}$, one could define the client update as $x_i^{(t,k+1)} = x_i^{(t,k)} - \eta_l(\Delta_i^{t,k} + y^{(t)} - y_i^{(t)})$, where $y_i^{(t)}$ and $y^{(t)}$ represent local and global correction buffers, respectively. The correction terms are then updated using equation 3. In the case of SGD, the correction term $y_i^{(t+1)}$ corresponds to the average of locally computed gradients: $y_i^{(t+1)} = \frac{1}{K}\sum_{k=1}^{K} \nabla f_i(x_i^{(t,k)}; \xi_i^{(t,k)})$. However, this equivalence breaks down when adaptive methods like Adam are used for local updates. In this setting, the update direction is no

longer a gradient, but instead involves adaptive scaling of moment estimates. Consequently, the correction term $y_i^t$ in this naive tracking setup becomes an average of local adaptive directions: $y_i^{(t+1)} = \frac{1}{K}\sum_{k=1}^{K} \frac{m_i^{(t,k)}}{\sqrt{\hat{v}_i^{(t,k)}}+\epsilon}$, where $m_i^{(t,k)}$ and $\hat{v}_i^{(t,k)}$ denote the first and second moment estimates at step $k$. We refer to this naive approach as *Federated Adaptive Moment Estimation with Naive Tracking* (FA-NT). Despite its simplicity, we show empirically that FA-NT fails to provide robust convergence under data heterogeneity in Appendix C. The failure stems from the incompatibility between the correction mechanism and the internal structure of adaptive methods. In particular, due to the nonlinearity and history-dependence introduced by moment estimation, the fixed-point condition is still not satisfied: $x^* \neq x^* - \eta_l (\frac{\beta_1 m^* + (1-\beta_1)\nabla f_i(x^*)}{\sqrt{\beta_2 v^* + (1-\beta_2)\nabla f_i(x^*) \odot \nabla f_i(x^*)}+\epsilon} + \frac{\nabla f(x^*)}{\sqrt{\nabla f(x^*) \odot \nabla f(x^*)}+\epsilon} - \frac{\nabla f_i(x^*)}{\sqrt{\nabla f_i(x^*) \odot \nabla f_i(x^*)}+\epsilon})$, because the globally optimal model $x^*$ may not be optimal for each client's local loss, i.e., $\nabla f_i(x^*) \neq 0$, due to data heterogeneity.

## 4.2 Federated Adam with Gradient Correction (FAdamGC)

**Key Idea.** Our idea is to mitigate the client-drift in adaptive FL by injecting a pre-estimation correction term that directly adjusts the gradient input to moment accumulation. Specifically, we observe that adding the correction $\nabla f(x^*) - \nabla f_i(x^*)$ before computing the moment terms ensures that $x^*$ becomes a fixed point of the modified update:

$$x^* = x^* - \eta_l \frac{\beta_1 m^* + (1-\beta_1)(\nabla f_i(x^*) + \overbrace{\nabla f(x^*) - \nabla f_i(x^*)}^{\text{pre-estimation correction}})}{\sqrt{\beta_2 v^* + (1-\beta_2)\nabla f_i(x^*) \odot \nabla f_i(x^*)} + \epsilon}. \quad (4)$$

Unlike post-estimation correction strategies such as those used in Naive Tracking, this approach aligns local updates more effectively with the global descent direction, thereby reducing the impact of data heterogeneity and stabilizing training. While the exact correction term $\nabla f(x^*) - \nabla f_i(x^*)$ is not accessible in practice, it can be approximated using gradient information from local updates. In settings with multiple local updates ($K > 1$), variance around the fixed-point scenario naturally arises because the server does not aggregate after every step. However, these deviations remain controllable under appropriate choices of local step sizes. Thus the fixed-point analysis serves as an intuitive guide: it highlights the structural conditions under which client drift is minimized and explains why the proposed pre-estimation correction provides robustness against data heterogeneity in realistic multi-step FL training.

**FAdamGC Algorithm.** Given this intuition, we propose *Federated Adaptive Moment Estimation with Gradient Correction* (FAdamGC). In contrast to FA-NT described in Sec. 4.1, in FAdamGC, gradient correction (GC) updates the correction buffer to track the averaged raw stochastic gradients *before* moment estimation: $y_i^{(t+1)} = \frac{1}{K}\sum_{k=1}^{K} g_i^{(t,k)}$. As shown in Figure 1, this gradient-level correction is then injected directly into the moment computation, effectively modifying both the first and second moment estimates.

As shown in Algorithm 1, during each global iteration $t$, the server samples a set of clients $\mathcal{S}^t$ with size $S$ for training. During the start of each local training interval, the server broadcasts the global model $x^{(t)}$ and the global correction term $y^{(t)}$ to all sampled clients $\mathcal{S}^t$. Then, for each sampled client $i$ at local iteration $k$, stochastic gradient $g_i^{(t,k)} = \nabla f_i(x_i^{(t,k)}, \xi_i^{(t,k)})$ is computed locally using the local model $x_i^{(t,k)}$. After each client $i$ computes $g_i^{(t,k)}$, the gradient correction is added to the gradient: $\hat{g}_i^{(t,k)} = g_i^{(t,k)} + y^{(t)} - y_i^{(t)}$. The adaptive local update direction $\Delta_i^{(t,k)}$ then calculated using $\hat{g}_i^{(t,k)}$. In this work, we use the Adam optimizer as shown in Line 9–10 of Algorithm 1, but it is possible for a more general framework where other adaptive optimizers are considered. A further evaluation on Gradient Correction with other adaptive optimizers is included in Appendix J.

**Enhanced Communication Efficiency via Selective Tracking.** Correction-based methods typically require additional communication overhead to update correction terms, which can offset their optimization benefits in bandwidth-constrained settings. To address this, we introduce a *Selective Tracking* mechanism that improves communication efficiency by updating correction terms on only a subset of clients. At each round $t$, only a randomly selected subset $\widetilde{\mathcal{S}}^t \subseteq \mathcal{S}^t$, with cardinality $\widetilde{S} \leq S$, participates in tracking updates. Our experiments demonstrate that even with $\widetilde{S} < S$, the

---

**Algorithm 1:** `FAdamGC`: Federated Adaptive Moment Estimation with Gradient Correction

---

**Input:** total rounds $T$, batch size $|\xi_i^{(t,k)}|$ for computing stochastic gradient, initial model $x^{(1)}$

1   Initialize $y_i^{(1)} = \nabla f_i(x^{(1)})$, $y^{(1)} = \nabla f(x^{(1)})$

2   **each global round** $t = 1, \ldots, T$ **do**

3     sample clients $\mathcal{S}^t \subseteq \{1, \ldots, n\}$ and sample clients for update tracking terms $\widetilde{\mathcal{S}}^t \subseteq \mathcal{S}^t$

4     server broadcasts $(x^{(t)}, y^{(t)})$ to all clients $i \in \mathcal{S}^t$

5     **each client** $i \in \mathcal{S}^t$ **in parallel do**

6       $x_i^{(t,1)} = x^{(t)}$, $m_i^{(t,1)} = 0$, $v_i^{(t,1)} = v_i^{(t)}$

7       **each local iteration** $k = 1, \ldots, K$ **do**

8         Compute batch gradient $g_i^{(t,k)}$, set moment estimation vector $\hat{g}_i^{(t,k)} = g_i^{(t,k)} + y^{(t)} - y_i^{(t)}$

9         Compute first & second moment with corrected gradient $m_i^{(t,k+1)} = \beta_1 m_i^{(t,k)} + (1 - \beta_1)\hat{g}_i^{(t,k)}$,

           $v_i^{(t,k+1)} = \beta_2 v_i^{(t,k)} + (1 - \beta_2)\hat{g}_i^{(t,k)} \odot \hat{g}_i^{(t,k)}$, and set $\hat{v}_i^{(t,k+1)} = \max(\hat{v}_i^{(t,k)}, v_i^{(t,k+1)})$

10        Let $\Delta_i^{(t,k)} = m_i^{(t,k+1)}/(\sqrt{\hat{v}_i^{(t,k+1)}} + \epsilon)$ and perform local update $x_i^{(t,k+1)} = x_i^{(t,k)} - \eta_l \Delta_i^{(t,k)}$

11       **if** $i \in \widetilde{\mathcal{S}}^t$ **then**

12         $y_i^{(t+1)} = \frac{1}{K} \sum_{k=1}^{K} g_i^{(t,k)}$

13       **else**

14         $y_i^{(t+1)} = y_i^{(t)}$

15       $v_i^{(t+1)} = v_i^{(t,K+1)}$

16     Server aggregates $x_i^{(t,K+1)} - x^{(t)}$ from clients $i \in \mathcal{S}^t$, and $y_i^{(t+1)} - y_i^{(t)}$ from clients $i \in \widetilde{\mathcal{S}}^t$.

17     $x^{(t+1)} = x^{(t)} + \frac{\eta_g}{S} \sum_{i \in \mathcal{S}^t} (x_i^{(t,K+1)} - x^{(t)})$.

18     $y^{(t+1)} = y^{(t)} + \frac{1}{n} \sum_{i \in \widetilde{\mathcal{S}}^t} (y_i^{(t+1)} - y_i^{(t)})$

---

proposed method achieves comparable performance to full participation while significantly reducing communication cost. After $K$ local steps, clients in $\mathcal{S}^t$ aggregate their models to update the global model $x^{(t)}$, while those in $\widetilde{\mathcal{S}}^t$ aggregate their correction terms to update $y^{(t)}$ on the server.

## 5   CONVERGENCE ANALYSIS

We present the convergence analysis of `FAdamGC` in this section. The detailed proofs, including of the intermediate lemmas, can be found in Appendix A and B.

**Assumption 5.1** (General Characteristics of Loss Functions). 1) Each local loss $f_i$ is $L$-smooth $\forall i \in \{1, \ldots, n\}$, i.e., $\|\nabla f_i(x_1) - \nabla f_i(x_2)\| \leq L\|x_1 - x_2\|$, $\forall x_1, x_2 \in \mathbb{R}^d$. 2) Consider $n_i^{(t,k)} = g_i^{(t,k)} - \nabla f_i(x_i^{(t,k)})$ as the unbiased noise of the gradient estimate through the SGD process for device $i$ at time $t, k$. The noise variance is upper bounded by $\sigma^2 > 0$, i.e., $\mathbb{E}[\|n_i^{(t,k)}\|^2] \leq \sigma^2 \, \forall i, t, k$.

In Theorem 5.2, we first analyze the convergence behavior for general non-convex loss functions in a special case where $\beta_2 = \epsilon = 0$. In this regime, the local update direction $\Delta_i^{t,k}$ is contractive, eliminating the need for bounded gradient assumptions and yielding tighter convergence bounds.

**Theorem 5.2.** Let $\beta_2 = \epsilon = 0$, by selecting $\eta_g \eta_l = \min\left\{\frac{\sqrt{\mathcal{F}S}}{\sqrt{\sigma^2 KTL}}, \frac{\mathcal{F}}{T}\right\}$, $\beta_1 = \sqrt[K]{\frac{KS - 2T}{2KS}}$, $\eta_l = \min\left\{\frac{1}{T}, \frac{\mathcal{F}}{K\sqrt{T}}\right\}$, under Assumption 5.1. By defining $\mathcal{F} = \mathbb{E}f(x^{(1)}) - f^*$, the iterates of `FAdamGC` can be bounded as:

$$\frac{1}{T} \sum_{t=1}^{T} \mathbb{E}\|\nabla f(x^{(t)})\|^2 = \mathcal{O}\left(\frac{L\mathcal{F}\sigma}{(SKT)^{\frac{1}{2}}} + \frac{(L\mathcal{F})^2}{T^2} + \frac{K^2(\sigma + L)^2}{T^2}\right) \tag{5}$$

**Novelty in the Proof.** A key novelty in the proof of Theorem 5.2 is that our local progression is internally controlled by the adaptive learning rate. When $\beta_2 = \epsilon = 0$, the local updates on each client will be bounded by the values of $\hat{v}_i^{(t,k)}$, i.e. $\|x_i^{(t,k)} - x_i^{(t,k-1)}\| \leq \eta_l$. This intrinsic bound eliminates the need for gradient boundedness assumptions and allows local updates to adapt flexibly to the geometry of the loss landscape, enabling more effective and assumption-light analysis.

Table 1: Convergence rate comparisons for $\frac{1}{T}\sum_{t=1}^{T}\|\nabla f(x^{(t)})\|^2$ across multiple adaptive methods. BDH stands for bounded data-heterogeneity, BG stands for bounded gradient norm, and $\mathcal{F}$ is the initial function gap $\mathbb{E}f(x^1) - f^*$. We can see that all methods have the same general $\mathcal{O}(1/\sqrt{nKT})$ non-convex convergence rate.

| Algorithms | Convergence Rate | Additional Assumptions |
|---|---|---|
| SCAFFOLD-M [2] (Cheng et al., 2023) | $\left(\frac{L\mathcal{F}\sigma^2}{nKT}\right)^{\frac{1}{2}} + \frac{L\mathcal{F}}{T}(1+n^{-\frac{1}{3}})$ | - |
| FedAdam (Reddi et al., 2020) | $\left(\frac{\mathcal{F}^2}{nKT}\right)^{\frac{1}{2}} + \frac{L\sigma^2}{G^2\sqrt{nKT}} + \frac{\sigma^2}{GKT} + \frac{L\sigma^2\sqrt{n}}{G^2\sqrt{K}T^{3/2}}$ | BG |
| FedAMS (Wang et al., 2022) | $\left(\frac{\mathcal{F}^2}{nKT}\right)^{\frac{1}{2}} + \frac{L\sqrt{nK}G^2}{\sqrt{\epsilon}T} + \frac{L^2K\sigma^2}{T} + \frac{G\sigma^2}{\sqrt{\epsilon^2 nKT}}$ | BG |
| PAdaMFed (Yan et al., 2025) | $\frac{(L+\sigma+\sqrt{L\sigma}+\mathcal{F})^2}{(nKT)^{\frac{1}{2}}} + \frac{(\sqrt{nK}\sigma+L)^2}{T}$ | - |
| FA-NT ($\beta_2 = \epsilon = 0$, Thm. C.3) | $\frac{L\mathcal{F}\sigma}{(nKT)^{\frac{1}{2}}} + \frac{(L\mathcal{F})^2}{T^2} + \frac{K^2(\sigma+L+nB)^2}{T^2}$ | BDH |
| FA-NT (Thm. C.2) | $\left(\frac{L\mathcal{F}\sigma^2}{nKT}\right)^{\frac{1}{2}} + \frac{L\mathcal{F}}{T} + \frac{KG^6}{\epsilon^2 T} + \frac{K^2(\sigma^2+(1+\epsilon^2)G^2)}{\epsilon^2 T}$ | BG |
| FAdamGC ($\beta_2 = \epsilon = 0$, Thm. 5.2) | $\frac{L\mathcal{F}\sigma}{(nKT)^{\frac{1}{2}}} + \frac{(L\mathcal{F})^2}{T^2} + \frac{K^2(\sigma+L)^2}{T}$ | - |
| FAdamGC (Thm. 5.4) | $\left(\frac{L\mathcal{F}\sigma^2}{nKT}\right)^{\frac{1}{2}} + \frac{L\mathcal{F}}{T} + \frac{KG^6}{\epsilon^2 T} + \frac{K(\sigma^2+(1+\epsilon^2)G^2)}{\epsilon^2 T}$ | BG |

**Remark.** This result in Theorem 5.2 demonstrates that `FAdamGC`, under milder assumptions than `FedAdam` and `FedAMS`, achieves convergence without requiring bounded gradients or explicit data heterogeneity conditions, which are properties shared by correction-based methods such as `SCAFFOLD`. In contrast, as shown in Theorem C.3, the naive tracking variant `FA-NT` still requires bounded data heterogeneity to ensure convergence. Our empirical results reinforce this theoretical distinction: the performance gap between `FAdamGC` and `FA-NT` widens as data heterogeneity increases, emphasizing the robustness of GC in practical federated settings. Notably, this convergence rate aligns with the behavior observed in `PAdaMed` (Table 1), as during the setting of $\beta_2 = 0$, both methods leverage gradient normalization in their update rules to regulate local client behavior and stabilize training under heterogeneous conditions.

We now present our theoretical result under any $\beta_2 > 0$, showing that the average of global loss gradient can attain linear speedup convergence to a stationary point under non-convex problems.

**Assumption 5.3** (Bounded Gradient). *The norm of the loss function $\ell(\cdot)$ is bounded by a constant $G$, i.e., $\|g_i^{(t)}\| \leq G, \ \forall i, t.$[1]*

**Theorem 5.4.** *Under Assumptions 5.1 and 5.3, and let the global $\eta_g$ and local $\eta_l$ step sizes satisfy $\eta_g\eta_l = \min\left\{\frac{(1-\beta_1)\beta_1}{8(G+\epsilon)KL}, \frac{(1-\beta_1)\beta_1}{12(G+\epsilon)TL}, \frac{(G+\epsilon)\sqrt{\mathcal{F}S}}{(1-\beta_1)\beta_1\sigma\sqrt{TKL}}\right\}$ and $\eta_l \leq \frac{(1-\beta_1)\beta_1\epsilon}{40(G+\epsilon)TL}$. For any $\beta_1, \beta_2 \in [0,1)$, the iterates of `FAdamGC` can be bounded as:*

$$\frac{1}{T}\sum_{t=1}^{T}\mathbb{E}\|\nabla f(x^{(t)})\|^2 = \mathcal{O}\left(\sqrt{\frac{L\mathcal{F}\sigma^2}{SKT}} + \frac{L\mathcal{F}}{T} + \frac{KG^6}{\epsilon^2 T} + \frac{K(\sigma^2+(1+\epsilon^2)G^2)}{\epsilon^2 T}\right). \tag{6}$$

**Novelty in the Proof.** A key technical contribution in the proof of Theorem 5.4 lies in efficiently bounding the deviation of the moment estimates. Unlike SGD-based methods where updates directly involve the current stochastic gradient $\nabla f_i(x_i^{(t,k)})$, the first moment $m_i^{(t,k)}$ involves a linear combination of historical gradients. This introduces significant challenges in controlling the deviation of $m_i^{(t,k)}$ during local training, since naively bounding this often leads to an unfavorable higher dependence on the number of local steps $K$. In our analysis, we show that by forgoing the bias correction design of Adam, the local deviation of $x_i^{(t,k)}$ can be controlled by any $\beta_1, \beta_2 \in [0,1)$.

**Remark.** Theorem 5.4 establishes that `FAdamGC` achieves linear speedup convergence to a stationary point, with the global model $x^{(t)}$ satisfying a rate of $\mathcal{O}(1/\sqrt{SKT})$ for sufficiently large $T$. This primary term aligns with existing methods in Table 1. When compared to the rates of `FedAdam` and `FedAMS` in Table 1, we observe a critical difference, where both methods lack dependence on the gradient variance $\sigma$ in their dominanting terms. As a result, their convergence cannot be effectively

---

[1]The bounded gradient assumption is a necessary condition for `Adam`-based methods, as controlling the behavior of the second moment relies on a universal bound on the gradient's magnitude. This assumption is widely adopted in numerous analysis of `Adam`-based algorithms (Kingma, 2014; Zou et al., 2019; Reddi et al., 2020; Sun et al., 2023).

[2]SCAFFOLD-M is not an adaptive algorithm thus does explicitly does not requires additional assumptions on bounded gradient nor bounded data heterogeneity to derive the results.

Table 2: Comparison of `FAdamGC` with multiple baselines on multiple datasets under full client participation. For all CIFAR-10 experiments, the target accuracy is 75%, and for CIFAR-100, the target accuracy is set at 50%, while for TinyImageNet, it is set at 30%. The target accuracy for SST-2 is set to 85% and for the other language tasks are set to 75%. We see that `FAdamGC` outperforms all baselines in most experiments under both settings.

| Settings | Task Type | Dataset | FedAvg-M (Cheng et al., 2023) | SCAFFOLD-M (Cheng et al., 2023) | FedAdam (Reddi et al., 2020) | FedAMS (Wang et al., 2022) | LocalAdam | PAdaMFed (Yan et al., 2025) | FA-NT | FAdamGC |
|---|---|---|---|---|---|---|---|---|---|---|
| Total Global Rounds | Image Tasks | CIFAR-10 | $1014.5_{\pm250.3}$ | $544.3_{\pm59.9}$ | $2532.5_{\pm343.3}$ | $2388.8_{\pm286.6}$ | $589.5_{\pm74.0}$ | $360.3_{\pm20.8}$ | $394.8_{\pm31.3}$ | $\mathbf{310.0}_{\pm16.8}$ |
| | | CIFAR-100 | $998.5_{\pm88.5}$ | $621.8_{\pm55.4}$ | $1854.0_{\pm185.3}$ | $1654_{\pm156.4}$ | $678.3_{\pm40.6}$ | $511.0_{\pm15.2}$ | $530.3_{\pm17.6}$ | $\mathbf{323.8}_{\pm16.3}$ |
| | | TinyImageNet | $215.5_{\pm10.4}$ | $242.2_{\pm15.4}$ | $543.2_{\pm45.2}$ | $463.5_{\pm35.7}$ | $177.3_{\pm8.3}$ | $87.5_{\pm6.4}$ | $157.0_{\pm6.4}$ | $\mathbf{66.3}_{\pm4.4}$ |
| | Language Tasks | 20NewsGroups | $245.5_{\pm27.6}$ | $214.0_{\pm17.7}$ | $247.0_{\pm8.1}$ | $224.5_{\pm5.4}$ | $156.8_{\pm7.8}$ | $149.3_{\pm3.4}$ | $155.0_{\pm7.0}$ | $\mathbf{143.3}_{\pm4.1}$ |
| | | QNLI | $171.3_{\pm41.2}$ | $162.5_{\pm37.4}$ | $137.5_{\pm10.2}$ | $145.0_{\pm9.8}$ | $117.0_{\pm10.4}$ | $87.3_{\pm13.2}$ | $99.8_{\pm11.2}$ | $\mathbf{55.5}_{\pm16.3}$ |
| | | QQP | $316.5_{\pm64.3}$ | $299.0_{\pm70.3}$ | $245.3_{\pm20.5}$ | $267.8_{\pm22.1}$ | $213.0_{\pm53.8}$ | $84.7_{\pm4.3}$ | $196.3_{\pm5.1}$ | $\mathbf{63.0}_{\pm4.6}$ |
| | | SST-2 | $150.5_{\pm36.1}$ | $129.8_{\pm32.5}$ | $84.3_{\pm4.2}$ | $78.8_{\pm4.4}$ | $47.0_{\pm5.8}$ | $42.3_{\pm7.9}$ | $48.8_{\pm8.4}$ | $\mathbf{30.3}_{\pm7.3}$ |
| Simulated Run Time (minutes) | Image Tasks | CIFAR-10 | $182.6_{\pm45.1}$ | $157.8_{\pm17.38}$ | $303.9_{\pm41.2}$ | $286.7_{\pm34.4}$ | $70.7_{\pm8.9}$ | $120.1_{\pm7.0}$ | $82.7_{\pm6.6}$ | $\mathbf{65.1}_{\pm3.5}$ |
| | | CIFAR-100 | $179.7_{\pm15.9}$ | $180.3_{\pm16.1}$ | $222.5_{\pm22.2}$ | $198.5_{\pm18.8}$ | $81.4_{\pm4.9}$ | $170.3_{\pm5.1}$ | $111.4_{\pm3.7}$ | $\mathbf{68.0}_{\pm3.4}$ |
| | | TinyImageNet | $38.8_{\pm1.9}$ | $70.2_{\pm4.5}$ | $65.2_{\pm5.4}$ | $55.6_{\pm4.3}$ | $21.3_{\pm1.0}$ | $29.2_{\pm2.1}$ | $33.0_{\pm1.3}$ | $\mathbf{13.9}_{\pm0.9}$ |
| | Language Tasks | 20NewsGroups | $34.4_{\pm3.9}$ | $35.1_{\pm2.9}$ | $31.6_{\pm1.0}$ | $28.7_{\pm0.7}$ | $\mathbf{20.1}_{\pm1.0}$ | $22.9_{\pm0.52}$ | $22.6_{\pm1.0}$ | $20.9_{\pm0.6}$ |
| | | QNLI | $24.8_{\pm6.0}$ | $27.4_{\pm6.3}$ | $18.2_{\pm1.4}$ | $19.2_{\pm1.3}$ | $15.5_{\pm1.4}$ | $13.4_{\pm2.0}$ | $15.0_{\pm1.7}$ | $\mathbf{8.4}_{\pm2.5}$ |
| | | QPP | $76.7_{\pm15.6}$ | $79.6_{\pm18.7}$ | $56.5_{\pm4.7}$ | $61.7_{\pm5.1}$ | $49.1_{\pm12.4}$ | $28.0_{\pm1.3}$ | $48.7_{\pm1.3}$ | $\mathbf{15.6}_{\pm1.1}$ |
| | | SST-2 | $127.7_{\pm36.1}$ | $114.7_{\pm32.5}$ | $74.5_{\pm4.2}$ | $69.7_{\pm4.4}$ | $41.3_{\pm5.8}$ | $38.5_{\pm7.2}$ | $43.1_{\pm8.4}$ | $\mathbf{26.8}_{\pm7.3}$ |

influenced by tuning the batch size. In contrast, the rate of `FAdamGC` explicitly incorporates $\sigma$, offering improved adaptability in real-world FL deployments. When compared with the rate of `FA-NT`, despite sharing the same dominating term rate, `FA-NT` incurs an additional $K$-value in the final term, with $\mathcal{O}\left(\frac{K^2(\sigma^2+(1+\epsilon^2)G^2)}{\epsilon^2 T}\right)$ instead of $\mathcal{O}\left(\frac{K(\sigma^2+(1+\epsilon^2)G^2)}{\epsilon^2 T}\right)$, and `FA-NT` imposes stricter constraints on the selection of local step sizes. Notably, both results rely on the bounded gradient assumption, which limits the theoretical separation between Naive Tracking and GC. The detailed proof for `FA-NT` can be found in Appendix D and E.

## 6 EXPERIMENTS

**Setup.** In the baseline comparisons on image tasks, we consider three widely used datasets: CIFAR-10, CIFAR-100 (Krizhevsky et al., 2009) and TinyImageNet (Le & Yang, 2015). For all three datasets, we adopt the ResNet-18 model. We set the total clients as $n = 100$, the client sampling rate $\frac{S}{n}$ to 10%, and set the number of local iterations $K = 60$. Furthermore, we conducted experiments on Large Language Models (LLMs). We tested on a Parameter-Efficient Fine-Tuning (PEFT) algorithm where only a limited amount of the LLM's parameters are trained using Low Rank Adaptation (LoRA) (Hu et al., 2022). We use the GPT-2 model (Radford et al., 2019), and set the total number of clients as $n = 100$ and the client sampling rate to 10%. We tested on two datasets, 20NewsGroups and the GLUE benchmark (Lang, 1995; Wang, 2018). Non-i.i.d. data are generated by distributing each dataset among clients through a Dirichlet distribution with parameter $\alpha = 0.1$. Mean and standard deviation are based on four random trials. Learning rates for each dataset are listed in Appendix H.

We compared our algorithm with several FL methods: 1) `FedAvg-M`, where local updates are performed using SGD optimizer with momentum, 2) `SCAFFOLD-M`, where local updates are performed using SGD optimizer with client-drift correction and momentum, 3) `FedAdam/FedAMS` where the Adam is used for the server updates, and 4) `LocalAdam`, where the local updates are performed using Adam optimizer. We perform a grid search through $\eta_l \in [10^{-4}, 10^{-1}]$ and $\eta_g \in [10^{-3}, 1]$, and plot the best performing results. We set $(\beta_1, \beta_2) = (0.9, 0.99)$ and $\epsilon = 10^{-8}$ for all Adam optimizers. All mean and standard deviation is based on four random trials.

We evaluate two key metrics: 1) *Total global rounds*, measured by the number of communication rounds $T$, which reflects the computational efficiency of each method; and 2) *Simulated run time (SRT)*, which estimates the training duration with each client gradient computation performed on a NVIDIA A100 Tensor Core GPU and client-server communication occurs over 100 Mbps links. Formally, for a total of global rounds, we compute: $SRT = \sum_{t=1}^{T}[\tau_{comp}(K) + \tau_{comm}V(t)]$, where $\tau_{comp}(K)$ denotes the local computation time in round $t$, $V(t)$ is the number of model-sized vectors transmitted in that round and $\tau_{comm}$ is the per-vector communication time. In this way, if an algorithm requires additional communication, such as sending correction tensors, $V(t)$ increases accordingly. This metric captures the practical impact of both computation and communication on overall system performance. Further discussion on communication and computation cost is included in Appendix I.

**Baseline Comparison on Image Tasks.** Table 2 compares the total cost required to reach a target accuracy across our proposed methods and several baselines. We assume full client participation for fair comparison with existing convergence rates. For our methods, the tracking subset size is set to $\widetilde{S} = S/2$. In our image task experiments, the total run time is largely dominated by communication,

which accounts for 10 to 25 more times than local training. This makes communication cost the primary performance bottleneck. Additional comparison of our method under different $\beta_2$ values are shown in Appendix G, showing the effectiveness of second moment estimation.

Under both evaluation methods, `FAdamGC` steadily outperforms the baselines under all datasets. The superior performance can be attributed to two key factors: 1) the use of adaptive updates via the Adam optimizer, which enables faster, geometry-agnostic local convergence; and 2) the carefully designed gradient correction mechanism. Unlike `FA-NT`, `FAdamGC`'s gradient correction leads to more effective mitigation of data heterogeneity and significantly improved convergence stability. Including second-moment information ($\beta_2 > 0$) further stabilizes corrected updates by adaptively scaling gradients to prevent erratic steps and by enhancing robustness against data heterogeneity, as evidenced by consistently better performance of Adam-based methods over SGD-based ones. This benefit disappears when $\beta_2 = 0$, explaining the observed empirical performance gap between `FAdaMFed` and `FAdamGC`.

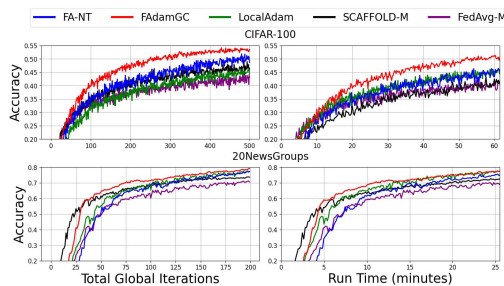

Figure 2: Comparison of achieved accuracy over global iterations and run time on CIFAR-100 and 20NewsGroups. `FAdamGC` steadily outperform baselines under different evaluation methods.

Figure 2 further illustrates the convergence trends under both metrics. While `FA-NT` achieves strong performance in terms of total rounds, it incurs higher communication overhead, limiting its practical efficiency. In contrast, `FAdamGC` achieves faster and more stable convergence while maintaining communication efficiency, demonstrating its robustness in heterogeneous federated settings.

**Baseline Comparison on Language Tasks.** In these PEFT tasks, the model weights transmitted between the server and each client constitute only 1.9% of the total parameters stored on the client side. As a result, local training time dominates, being 10 to 30 times longer than the communication time, this indicates that the primary bottleneck in this setting is computational cost.

Table 2 presents the performance of our method in language tasks. In these experiments, the size of the sample set used for model aggregation is equal to the sample set for tracking term aggregation, i.e., $\widetilde{S} = S$. The local epochs between two consecutive global aggregations is set to one. The results demonstrate that while the improvement introduced by NT is less pronounced compared to its impact in image tasks, GC consistently yields significant enhancements over the baselines. When evaluating the run time, `AdamGC` is able to achieve better results than most algorithms, *emphasizing its ability to capture and leverage first-order information effectively during adaptive optimization*.

**Impact of Data Heterogeneity.** Figure 3 illustrates the performance improvement of our algorithm compared to `LocalAdam` under varying levels of non-iid data. We vary the Dirichlet parameter $\alpha$ from 0.1 to 1 to represent levels of non-i.i.d. When evaluating communication rounds, the gap between `LocalAdam` and `FAdamGC` is more pronounced under high data heterogeneity. In contrast, for more i.i.d. settings, the performance gap between `FAdamGC` and `LocalAdam` becomes negligible. When evaluating the run time, we can see that `FAdamGC` still outperforms `LocalAdam` under high data heterogeneity. We also see that `FAdamGC` mitigates data heterogeneity better than `FA-NT`, this observation aligns with Theorem 5.2 and C.3, showing that GC deals with data heterogeneity better than Naive Tracking.

**Communication Efficiency under Different $\widetilde{S}$.** Figure 4 evaluates the performance of our proposed methods under different subset sizes $\tilde{S}$ used for updating tracking terms. We set the total clients to be $n = 100$. In these set of experiments, we increase the client sample size from $S = 10$ to $S = 50$. This allows a wider range of $\widetilde{S}$ value to compare the difference in terms of communication efficiency. We then compare the communication and computation cost for both algorithms across various $\widetilde{S}$ values ranging from 1 to 50. The results reveal that, for both `FA-NT` and `FAdamGC`, the total iterations to achieve certain accuracy under increases slowly as the $\widetilde{S}$ value decreases. *This finding demonstrates the possibility to significantly reduce the total number of communications required by the drift compensation process without compromising training performance*. The implication is particularly valuable when communication costs are a major bottleneck. These findings are further

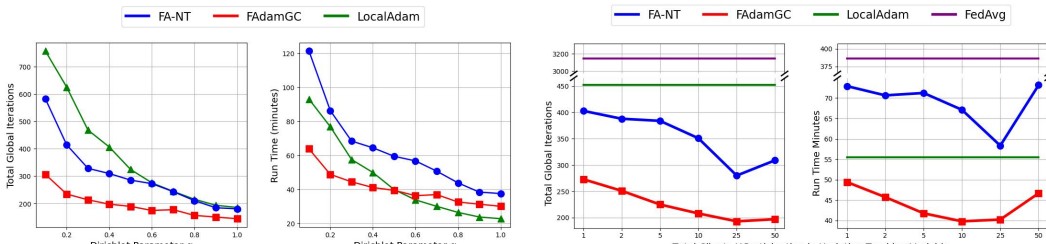

Figure 3: Comparison of the total cost of Adam-based methods under varying Dirichlet parameters on CIFAR-100 to attain 50% accuracy.

Figure 4: Comparison of cost to attain certain accuracy between different tracking sampling rates on CIFAR-100 with $S = 50$, where the target accuracy is 50%.

substantiated by the run time plots in Figure 4. The plots highlight that *the appropriate $\widetilde{S}$ values not only reduces communication overhead but also maintains superior performance compared to all other configurations and baseline methods*. This advantage underscores the robustness of `FAdamGC`, which effectively balances communication efficiency and convergence. By leveraging a reduced set size $\widetilde{S}$, `FAdamGC` achieves steady improvements over baselines while preserving its performance.

## 7 CONCLUSION

In this paper, we introduce Gradient Correction, a method to incorporate client-drift compensation into adaptive FL algorithms. By incorporating gradient correction tracking into local adaptive optimizers, we propose a novel algorithms `FAdamGC`. Through rigorous theoretical analysis, we demonstrate that our algorithm achieve linear speedup convergence to a stationary point while showing the naively injecting correction terms into adaptive FL may lead to sub-optimal results with higher dependence on data heterogeneity. Comprehensive numerical evaluations confirm that our method outperform all baselines, delivering superior training performance in heterogeneous data settings.

**Reproducibility Statement.** This paper provides all the necessary information to reproduce the main experimental results. The datasets used are all publicly available, while the model used, training details, and hyperparameters are documented in Sec. 6 and Appendix H. The implementation code is included in the supplementary material of the submission.

**LLM Usage.** ChatGPT (GPT-5) was used solely as a language assistive tool to enhance manuscript clarity by polishing grammar and rephrasing sentences. It was not involved in research ideation, methodology design, data analysis, or experimental implementation. All scientific content, theoretical interpretations, and study results are executed by the authors.

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

# Appendix

## A    THEORETICAL ANALYSIS FOR FADAMGC (THEOREM 5.4)

We first define $c^k$ as the sum of all moving average coefficients to compute the first order moment $m_i^{(t,k)}$:

$$c^{(k,k')} = (1 - \beta_1)\beta_1^{k-k'} \tag{7}$$

$$c^k = \sum_{k'=1}^{k} c^{(k,k')} < 1 \tag{8}$$

We then define the expected first order moment $\tilde{m}_i^{(t,k)}$ as the following:

$$\tilde{m}_i^{(t,k)} \triangleq \sum_{k'=1}^{k} c^{(k,k')} \left( \nabla f_i(x_i^{(t,k)}) - \nabla f_i(\gamma_i^{(t,k)}) + \frac{1}{n}\sum_{i=1}^{n} \nabla f_i(\gamma_i^{(t,k)}) \right) \tag{9}$$

Where $\gamma_i^{(t,k)}$ is an auxilary variable that tracks the GT terms:

$$\gamma_i^{(t,k)} = \begin{cases} x_i^{(t-1,k)} & i \in \mathcal{Y}^{t-1} \\ \gamma_i^{(t-1,k)} & i \notin \mathcal{Y}^{t-1} \end{cases} \tag{10}$$

We further define the local deviation term $\Xi^{(t)}$ as:

$$\Xi^{(t)} = \frac{1}{n}\sum_{i=1}^{n}\sum_{k=1}^{K} \mathbb{E}\| \sum_{k'=1}^{k} c^{(k,k')}\nabla f_i(x_i^{(t,k')}) - c^k\nabla f_i(x^{(t)})\|^2 \tag{11}$$

*Proof.* Given global iteration $t$, the update of the model at the server can be written as:

$$x^{(t+1)} = x^{(t)} + \eta_g \frac{1}{S}\sum_{i\in\mathcal{S}^{(t)}} (x_i^{(t,K+1)} - x^{(t)}) \tag{12}$$

$$= x^{(t)} - \eta_g\eta_l \frac{1}{S}\sum_{i\in\mathcal{S}^{(t)}}\sum_{k=1}^{K} \frac{m_i^{(t,k)}}{\sqrt{\hat{v}_i^{(t,k)}} + \epsilon} \tag{13}$$

By injecting Assumption 5.1, we can get the following inequality:

$$\mathbb{E}f(x^{(t+1)}) \leq \mathbb{E}f(x^{(t)}) - \eta_g\eta_l\mathbb{E}\underbrace{\left\langle \nabla f(x^{(t)}), \frac{1}{S}\sum_{i\in\mathcal{S}^{(t)}}\sum_{k=1}^{K} \frac{m_i^{(t,k)}}{\sqrt{\hat{v}_i^{(t,k)}} + \epsilon} \right\rangle}_{\text{Term I}} \tag{14}$$

$$+ \eta_g^2\eta_l^2 \frac{L}{2}\mathbb{E}\underbrace{\left\| \frac{1}{S}\sum_{i\in\mathcal{S}^{(t)}}\sum_{k=1}^{K} \frac{m_i^{(t,k)}}{\sqrt{\hat{v}_i^{(t,k)}} + \epsilon} \right\|^2}_{\text{Term II}} \tag{15}$$

For term I, we first define the average of all square root second moment:

$$\bar{v}^{(t)} = \frac{1}{n}\sum_{i=1}^{n} \sqrt{v_i^{(t)}} \tag{16}$$

Term I can be upper bounded as:

$$- \eta_g\eta_l\mathbb{E}\left\langle \nabla f(x^{(t)}), \frac{1}{S}\sum_{i\in\mathcal{S}^{(t)}}\sum_{k=1}^{K} \frac{m_i^{(t,k)}}{\sqrt{\hat{v}_i^{(t,k)}} + \epsilon} \right\rangle$$

$$= -\eta_g \eta_l \mathbb{E} \left\langle \nabla f(x^{(t)}), \frac{1}{S} \sum_{i \in \mathcal{S}^{(t)}} \sum_{k=1}^{K} \frac{\tilde{m}_i^{(t,k)}}{\sqrt{\hat{v}_i^{(t,k)}} + \epsilon} \right\rangle$$

$$= -\eta_g \eta_l \mathbb{E} \left\langle \nabla f(x^{(t)}), \frac{1}{n} \sum_{i=1}^{n} \sum_{k=1}^{K} \frac{\sum_{k'=1}^{k} c^{(k,k')} \nabla f_i(x_i^{(t,k')})}{\sqrt{\hat{v}_i^{(t,k)}} + \epsilon} \right\rangle$$

$$- \eta_g \eta_l \mathbb{E} \left\langle \nabla f(x^{(t)}), \frac{1}{n} \sum_{i,k} \frac{\sum_{k'=1}^{k} c^{(k,k')} \left( \frac{1}{K} \sum_{k''=1}^{K} \left( \frac{1}{n} \sum_{i'=1}^{n} \nabla f_{i'}(\gamma_{i'}^{(t,k'')}) - \nabla f_i(\gamma_i^{(t,k'')}) \right) \right)}{\sqrt{\hat{v}_i^{(t,k)}} + \epsilon} \right\rangle \quad (17)$$

Using the fact that $\sum_{i=1}^{n} \left( \nabla f_i(\gamma_i^{(t,k)}) - \frac{1}{n} \sum_{i'=1}^{n} \nabla f_i'(\gamma_i'^{(t,k)}) \right) = 0$, we can show that:

$$- \eta_g \eta_l \mathbb{E} \left\langle \nabla f(x^{(t)}), \frac{1}{S} \sum_{i \in \mathcal{S}^{(t)}} \sum_{k=1}^{K} \frac{m_i^{(t,k)}}{\sqrt{\hat{v}_i^{(t,k)}} + \epsilon} \right\rangle$$

$$= -\eta_g \eta_l \mathbb{E} \left\langle \nabla f(x^{(t)}), \frac{1}{n} \sum_{i=1}^{n} \sum_{k=1}^{K} \left( \frac{\sum_{k'=1}^{k} c^{(k,k')} \nabla f_i(x_i^{(t,k')})}{\sqrt{\hat{v}_i^{(t,k)}} + \epsilon} - \frac{\sum_{k'=1}^{k} c^{(k,k')} \nabla f_i(x_i^{(t,k')})}{\bar{v}^{(t)} + \epsilon} \right) \right\rangle$$

$$- \eta_g \eta_l \mathbb{E} \left\langle \nabla f(x^{(t)}), \frac{1}{n} \sum_{i=1}^{n} \sum_{k=1}^{K} \left( \frac{\sum_{k'=1}^{k} c^{(k,k')} \nabla f_i(x_i^{(t,k')})}{\bar{v}^{(t)} + \epsilon} - \frac{c^k \nabla f_i(x^{(t)})}{\bar{v}^{(t)} + \epsilon} + \frac{c^k \nabla f_i(x^{(t)})}{\bar{v}^{(t)} + \epsilon} \right) \right\rangle$$

$$\leq -\eta_g \eta_l K \frac{(1-\beta_1)\beta_1}{G+\epsilon} \mathbb{E} \| \nabla f(x^{(t)}) \|^2$$

$$- \eta_g \eta_l \mathbb{E} \left\langle \nabla f(x^{(t)}), \frac{1}{n} \sum_{i=1}^{n} \sum_{k=1}^{K} \left( \frac{\sum_{k'=1}^{k} c^{(k,k')} \nabla f_i(x_i^{(t,k')})}{\sqrt{\hat{v}_i^{(t,k)}} + \epsilon} - \frac{\sum_{k'=1}^{k} c^{(k,k')} \nabla f_i(x_i^{(t,k')})}{\bar{v}^{(t)} + \epsilon} \right) \right\rangle$$

$$- \eta_g \eta_l \mathbb{E} \left\langle \nabla f(x^{(t)}), \frac{1}{n} \sum_{i=1}^{n} \sum_{k=1}^{K} \left( \frac{\sum_{k'=1}^{k} c^{(k,k')} \nabla f_i(x_i^{(t,k')})}{\bar{v}^{(t)} + \epsilon} - \frac{c^k \nabla f_i(x^{(t)})}{\bar{v}^{(t)} + \epsilon} \right) \right\rangle$$

$$\leq -\frac{\eta_g \eta_l K}{2} \frac{(1-\beta_1)\beta_1}{G+\epsilon} \mathbb{E} \| \nabla f(x^{(t)}) \|^2$$

$$+ \eta_g \eta_l \frac{G+\epsilon}{(1-\beta_1)\beta_1 \epsilon^2} \frac{1}{n} \sum_{i=1}^{n} \sum_{k=1}^{K} \mathbb{E} \| \sum_{k'=1}^{k} c^{(k,k')} \nabla f_i(x_i^{(t,k')}) - c^k \nabla f_i(x^{(t)}) \|^2$$

$$+ \eta_g \eta_l K \frac{G+\epsilon}{(1-\beta_1)\beta_1} \mathbb{E} \| \frac{1}{nK} \sum_{i=1}^{n} \sum_{k=1}^{K} \frac{\sum_{k'=1}^{k} c^{(k,k')} \nabla f_i(x_i^{(t,k')})}{\sqrt{\hat{v}_i^{(t,k)}} + \epsilon} - \frac{\sum_{k'=1}^{k} c^{(k,k')} \nabla f_i(x_i^{(t,k')})}{\bar{v}^{(t)} + \epsilon} \|^2$$

$$\leq -\frac{\eta_g \eta_l K}{2} \frac{(1-\beta_1)\beta_1}{G+\epsilon} \mathbb{E} \| \nabla f(x^{(t)}) \|^2$$

$$+ \eta_g \eta_l \frac{G+\epsilon}{(1-\beta_1)\beta_1 \epsilon^2} \frac{1}{n} \sum_{i=1}^{n} \sum_{k=1}^{K} \mathbb{E} \| \sum_{k'=1}^{k} c^{(k,k')} \nabla f_i(x_i^{(t,k')}) - c^k \nabla f_i(x^{(t)}) \|^2$$

$$+ \eta_g \eta_l K \frac{G^2(G+\epsilon)}{(1-\beta_1)\beta_1} \mathbb{E} \| \frac{1}{nK} \sum_{i=1}^{n} \sum_{k=1}^{K} \frac{\sqrt{\hat{v}_i^{(t,k)}} - \bar{v}^{(t)}}{(\sqrt{\hat{v}_i^{(t,k)}} + \epsilon)(\bar{v}^{(t)} + \epsilon)} \|^2$$

$$\leq -\frac{\eta_g \eta_l K}{2} \frac{(1-\beta_1)\beta_1}{G+\epsilon} \mathbb{E} \| \nabla f(x^{(t)}) \|^2$$

$$+ \eta_g \eta_l \frac{G+\epsilon}{(1-\beta_1)\beta_1 \epsilon^2} \frac{1}{n} \sum_{i=1}^{n} \sum_{k=1}^{K} \mathbb{E} \| \sum_{k'=1}^{k} c^{(k,k')} \nabla f_i(x_i^{(t,k')}) - c^k \nabla f_i(x^{(t)}) \|^2$$

$$+ \eta_g \eta_l K \frac{G^2(G+\epsilon)}{(1-\beta_1)\beta_1 \epsilon^2} \mathbb{E} \| \bar{v}^{(t+1)} - \bar{v}^{(t)} \|^2 \quad (18)$$

Term II can be upper bounded as:

$$\frac{\eta_g^2 \eta_l^2 L}{2} \mathbb{E} \left\| \frac{1}{S} \sum_{i \in \mathcal{S}^{(t)}} \sum_{k=1}^{K} \frac{m_i^{(t,k)}}{\sqrt{\hat{v}_i^{(t,k)}} + \epsilon} \right\|^2$$

$$\leq 2\eta_g^2 \eta_l^2 K^2 L \mathbb{E} \|\nabla f(x^{(t)})\|^2 + \frac{4\eta_g^2 \eta_l^2 KL}{\epsilon^2} \frac{1}{n} \sum_{i=1}^{n} \sum_{k=1}^{K} \| \sum_{k'=1}^{k} c^{(k,k')} \nabla f_i(x_i^{(t,k')}) - c^k \nabla f_i(x^{(t)}) \|^2$$

$$+ \frac{4\eta_g^2 \eta_l^2 (1-\epsilon)^2}{\epsilon^2} K^2 L G^2 + \eta_g^2 \eta_l^2 K L \sigma^2 \tag{19}$$

If we choose $\eta_g \eta_l \leq \frac{(1-\beta_1)\beta_1}{8KL(G+\epsilon)}$, we can combine Term I and II and get:

$$\mathbb{E} f(x^{(t+1)}) \leq \mathbb{E} f(x^{(t)}) - \frac{\eta_g \eta_l K}{4} \frac{(1-\beta_1)\beta_1}{G+\epsilon} \mathbb{E} \|\nabla f(x^{(t)})\|^2 + \frac{2\eta_g \eta_l}{\epsilon^2} \frac{G+\epsilon}{(1-\beta_1)\beta_1} \Xi^{(t)}$$

$$+ \eta_g \eta_l K \frac{G^2(G+\epsilon)}{(1-\beta_1)\beta_1 \epsilon^2} \mathbb{E} \|\bar{v}^{(t+1)} - \bar{v}^{(t)}\|^2 + \frac{2\eta_g^2 \eta_l^2 (1-\epsilon)^2}{\epsilon^2} K^2 L G^2 + \frac{\eta_g^2 \eta_l^2 K L \sigma^2}{S} \tag{20}$$

By using Lemma A.1, we can bound $\Xi^{(t)}$ and get:

$$\mathbb{E} f(x^{(t+1)}) \leq \mathbb{E} f(x^{(t)}) - \frac{\eta_g \eta_l K}{4} \frac{(1-\beta_1)\beta_1}{G+\epsilon} \mathbb{E} \|\nabla f(x^{(t)})\|^2$$

$$+ \frac{2\eta_g \eta_l}{\epsilon^2} \frac{G+\epsilon}{(1-\beta_1)\beta_1} \frac{6\eta_l^2 K^2 L^2}{\epsilon^2} \left( 6(1-\beta_2)G^6 + 8(1-\beta_1)(\sigma^2 + G^2) \right)$$

$$+ \eta_g \eta_l K \frac{G^2(G+\epsilon)}{(1-\beta_1)\beta_1 \epsilon^2} \mathbb{E} \|\bar{v}^{(t+1)} - \bar{v}^{(t)}\|^2 + \frac{2\eta_g^2 \eta_l^2 (1-\epsilon)^2}{\epsilon^2} K^2 L G^2 + \frac{\eta_g^2 \eta_l^2 K L \sigma^2}{S} \tag{21}$$

We can reorganize the inequality and get:

$$\frac{\eta_g \eta_l K}{4} \frac{(1-\beta_1)\beta_1}{G+\epsilon} \mathbb{E} \|\nabla f(x^{(t)})\|^2$$

$$\leq \mathbb{E} f(x^{(t)}) - \mathbb{E} f(x^{(t+1)})$$

$$+ \frac{2\eta_g \eta_l}{\epsilon^2} \frac{G+\epsilon}{(1-\beta_1)\beta_1} \frac{6\eta_l^2 K^2 L^2}{\epsilon^2} \left( 6(1-\beta_2)G^6 + 8(1-\beta_1)(\sigma^2 + G^2) \right)$$

$$+ \eta_g \eta_l K \frac{G^2(G+\epsilon)}{(1-\beta_1)\beta_1 \epsilon^2} \mathbb{E} \|\bar{v}^{(t+1)} - \bar{v}^{(t)}\|^2$$

$$+ \frac{2\eta_g^2 \eta_l^2 (1-\epsilon)^2}{\epsilon^2} K^2 L G^2 + \frac{\eta_g^2 \eta_l^2 K L \sigma^2}{S} \tag{22}$$

By summing up all global iterations $T$ and dividing both sides with constants, we get:

$$\frac{1}{T} \sum_{i=1}^{T} \mathbb{E} \|\nabla f(x^{(t)})\|^2 \leq \frac{4(G+\epsilon)(\mathbb{E} f(x^{(1)}) - \mathbb{E} f(x^{(T+1)}))}{\eta_l \eta_g K (1-\beta_1)\beta_1 T}$$

$$+ \frac{(G+\epsilon)^2}{(1-\beta_1)^2 \beta_1^2} \frac{6\eta_l^2 K L^2}{\epsilon^4} \left( 6(1-\beta_2)G^6 + 8(1-\beta_1)(\sigma^2 + G^2) + G^2 \right)$$

$$+ \frac{G^4(G+\epsilon)^2}{(1-\beta_1)^2 \beta_1^2 \epsilon^2 T}$$

$$+ \frac{8\eta_g \eta_l (G+\epsilon)}{(1-\beta_1)\beta_1 \epsilon^2} L(KG^2 + \sigma^2) \tag{23}$$

Finally, by defining $\mathcal{F} = \mathbb{E} f(x^{(1)}) - f^*$ and bounding $\eta_l \leq \frac{(1-\beta_1)\beta_1 \epsilon}{40(G+\epsilon)\sqrt{T}L}$, $\eta_g \eta_l \leq \frac{(1-\beta_1)\beta_1}{12(G+\epsilon)TL}$, and a specific step size

$$\eta_g \eta_l = \min\left( \frac{(1-\beta_1)\beta_1}{8(G+\epsilon)KL}, \frac{(1-\beta_1)\beta_1}{12(G+\epsilon)TL}, \frac{(G+\epsilon)\sqrt{S}}{(1-\beta_1)\beta_1 \sigma \sqrt{TKL}} \right) \tag{24}$$

, we can get:

$$\frac{1}{T}\sum_{i=1}^{T}\mathbb{E}\|\nabla f(x^{(t)})\|^2 \le \frac{L\mathcal{F}}{T}$$

$$+ 2\sqrt{\frac{L\mathcal{F}\sigma^2}{SKT}}$$

$$+ \left(\frac{2}{(1-\beta_1)^2\beta_1^2} + K(1-\beta_2)\right)\frac{G^6}{\epsilon^2 T}$$

$$+ K(1-\beta_1)\frac{\sigma^2 + G^2}{\epsilon^2 T}$$

$$= \mathcal{O}\left(\sqrt{\frac{L\mathcal{F}\sigma^2}{SKT}} + \frac{L\mathcal{F}}{T} + \frac{KG^6}{\epsilon^2 T} + \frac{K(\sigma^2 + (1+\epsilon^2)G^2)}{\epsilon^2 T}\right) \tag{25}$$

$\square$

**Lemma A.1.** *Under Assumption 5.1, the local deviation term $\Xi^{(t)}$ can be bounded as the following:*

$$\Xi^{(t)} \le \frac{6\eta_l^2 K^2 L^2}{\epsilon^2}\left(6(1-\beta_2)G^6 + 8(1-\beta_1)(\sigma^2 + G^2)\right) \tag{26}$$

*Proof.* We first define the unbiased version of $m_i^{(t,k)}$ taking expectation on all stochastic gradients $g_i^{(t,k)}$. Then, we can bound the deviation term as:

$$\Xi^{(t)} = \frac{1}{n}\sum_{i=1}^{n}\sum_{k=1}^{K}\mathbb{E}\|\sum_{k'=1}^{k}c^{(k,k')}\nabla f_i(x_i^{(t,k')}) - c^k\nabla f_i(x^{(t)})\|^2 \tag{27}$$

$$\le \frac{L^2}{n}\sum_{k=1}^{K}\underbrace{\sum_{i=1}^{n}\sum_{k'=1}^{k}c^{(k,k')}\mathbb{E}\|x_i^{(t,k')} - x^{(t)}\|^2}_{e^{t,k}} \tag{28}$$

We can further bound $e^{t,k}$, for some nonnegative constant $a$, we can get:

$$\sum_{i=1}^{n}\sum_{k'=1}^{k}c^{(k,k')}\mathbb{E}\|x_i^{(t,k')} - x^{(t)}\|^2 \tag{29}$$

$$= \sum_{i=1}^{n}\sum_{k'=1}^{k}c^{(k,k')}\mathbb{E}\left\|x_i^{(t,k'-1)} - \eta_l\frac{m_i^{t,k'-1}}{\sqrt{\hat{v}_i^{t,k'-1}} + \epsilon} - x^{(t)}\right\|^2 \tag{30}$$

$$\le (1+a)\sum_{i=1}^{n}\sum_{k'=1}^{k}c^{(k,k')}\mathbb{E}\left\|x_i^{(t,k'-1)} - x^{(t)}\right\|^2 + (1+\frac{1}{a})\eta_l^2\sum_{i=1}^{n}\sum_{k'=1}^{k}c^{(k,k')}\mathbb{E}\left\|\frac{m_i^{t,k'-1}}{\sqrt{\hat{v}_i^{t,k'-1}} + \epsilon}\right\|^2 \tag{31}$$

From equation 70, we can show that:

$$c^{(k,k')} = (1-\beta_1)\beta_1^{k-k'} = c^{(k-1,k'-1)} \tag{32}$$

Thus, we can further bound the terms as:

$$\sum_{i=1}^{n}\sum_{k'=1}^{k}c^{(k,k')}\mathbb{E}\|x_i^{(t,k')} - x^{(t)}\|^2 \tag{33}$$

$$\leq (1+a)\sum_{i=1}^{n}\sum_{k'=0}^{k-1} c^{(k-1,k')}\mathbb{E}\|x_i^{(t,k')} - x^{(t)}\|^2 + (1+\frac{1}{a})\eta_l^2\sum_{i=1}^{n}\sum_{k'=0}^{k-1} c^{(k-1,k')}\mathbb{E}\left\|\frac{m_i^{t,k'}}{\sqrt{\hat{v}_i^{t,k'}}+\epsilon}\right\|^2 \tag{34}$$

$$= (1+a)\sum_{i=1}^{n}\sum_{k'=1}^{k-1} c^{(k-1,k')}\mathbb{E}\|x_i^{(t,k')} - x^{(t)}\|^2 + (1+\frac{1}{a})\eta_l^2\sum_{i=1}^{n}\sum_{k'=1}^{k-1} c^{(k-1,k')}\mathbb{E}\left\|\frac{m_i^{t,k'}}{\sqrt{\hat{v}_i^{t,k'}}+\epsilon}\right\|^2 \tag{35}$$

$$= (1+a)e^{t,k-1} + (1+\frac{1}{a})\eta_l^2 \underbrace{\sum_{i=1}^{n}\sum_{k'=1}^{k-1} c^{(k-1,k')}\mathbb{E}\left\|\frac{m_i^{t,k'}}{\sqrt{\hat{v}_i^{t,k'}}+\epsilon}\right\|^2}_{s^{t,k-1}} \tag{36}$$

We then bound $s^{t,k}$:

$$\sum_{i=1}^{n}\sum_{k'=1}^{k} c^{(k,k')}\mathbb{E}\left\|\frac{m_i^{t,k'}}{\sqrt{\hat{v}_i^{t,k'}}+\epsilon}\right\|^2 \tag{37}$$

$$\leq 2\sum_{i=1}^{n}\sum_{k'=1}^{k} c^{(k,k')}\mathbb{E}\left\|\frac{m_i^{t,k'}}{\sqrt{\hat{v}_i^{t,k'}}+\epsilon} - \frac{m_i^{t,k'}}{\sqrt{\beta_2\hat{v}_i^{t,k'-1}}+\epsilon}\right\|^2 + 2\sum_{i=1}^{n}\sum_{k'=1}^{k} c^{(k,k')}\mathbb{E}\left\|\frac{m_i^{t,k'}}{\sqrt{\beta_2\hat{v}_i^{t,k'-1}}+\epsilon}\right\|^2 \tag{38}$$

$$\leq 2G^2\sum_{i=1}^{n}\sum_{k'=1}^{k} c^{(k,k')}\mathbb{E}\left\|\frac{1}{\sqrt{\hat{v}_i^{t,k'}}+\epsilon} - \frac{1}{\sqrt{\beta_2\hat{v}_i^{t,k'-1}}+\epsilon}\right\|^2 \tag{39}$$

$$+ 2\sum_{i=1}^{n}\sum_{k'=1}^{k} c^{(k,k')}\mathbb{E}\left\|\frac{\beta_1 m_i^{t,k'-1} + (1-\beta_1)\hat{g}_i^{t,k'}}{\sqrt{\beta_2\hat{v}_i^{t,k'-1}}+\epsilon}\right\|^2 \tag{40}$$

$$\leq 2(1-\beta_2)G^6\frac{n}{\epsilon^2} + 2\beta_1\sum_{i=1}^{n}\sum_{k'=1}^{k-1} c^{(k-1,k')}\mathbb{E}\left\|\frac{m_i^{t,k'}}{\sqrt{\beta_2\hat{v}_i^{t,k'}}+\epsilon}\right\|^2 \tag{41}$$

$$+ \frac{2(1-\beta_1)}{\epsilon^2}\sum_{k'=1}^{k} c^{(k,k')}\mathbb{E}\left\|\hat{g}_i^{t,k'} - \nabla f_i(x_i^{t,k'}) + \nabla f_i(x_i^{t,k'}) - \nabla f_i(x^t) + \nabla f_i(x^t)\right\|^2 \tag{42}$$

$$\leq \frac{8L^2}{\epsilon^2}(1-\beta_1)e^{t,k} + \underbrace{\left(6(1-\beta_2)G^6 + 8(1-\beta_1)(\sigma^2 + G^2)\right)\frac{n}{\epsilon^2}}_{C_1} \tag{43}$$

We thus get the recursive relationship between $e^{t,k}$ and $s^{t,k}$:

$$\begin{cases} e^{t,k} & \leq (1+a)e^{t,k-1} + (1+\frac{1}{a})\eta_l^2 s^{t,k-1} \\ s^{t,k} & \leq \frac{8L^2(1-\beta_1)}{\epsilon^2}e^{t,k} + C_1 \end{cases} \tag{44}$$

If we restrict the choice of the momentum term with $(1-\beta_1) < \frac{\epsilon^2}{16KL^2}$ and let $\eta_l \leq \frac{\epsilon}{4\sqrt{(1-\beta_1)}KL}$, we can let $a = 1$ and get:

$$e^{t,k} \leq (1+\frac{1}{K-1})e^{t,k-1}2\eta_l^2 C_1 \tag{45}$$

By unrolling the recursion, we get:

$$e^{t,k} \leq \sum_{k'=1}^{k} k'(1+\frac{1}{K-1})^{k'}2\eta_l^2 C_1 \leq 6K\eta_l^2 C_1 \tag{46}$$

Finally, plug equation 46 back to the definition of $\Xi^t$ and we get:

$$\Xi^{(t)} \leq \frac{L^2}{n} \sum_{k=1}^{K} e^{t,k} \tag{47}$$

$$\leq \frac{6\eta_l^2 K^2 L^2}{\epsilon^2} \left( 6(1-\beta_2)G^6 + 8(1-\beta_1)(\sigma^2 + G^2) \right) \tag{48}$$

$\square$

## B  ANALYSIS OF FADAMGC FOR SPECIAL CASES (THEOREM. 5.2)

Similar to Appendix E, with the adaptive stepsize no longer relying on an estimation of the second order moment but the norm of the first order information, we now have $\|\frac{m_i^{t,k}}{\|\sqrt{v_i^{t,k}}\|}\| \leq 1$ for any $\beta_1 \in (0,1)$.

We first write out the update from using $L$-smoothness, we first define an arbitrary vector $q^t \in \mathbb{R}^d$ that will be determined later.

$$\mathbb{E}f(x^{t+1}) - f(x^t)$$

$$\leq -K\eta_g\eta_l\mathbb{E}\left\langle \nabla f(x^t), \frac{1}{SK}\sum_{i,k}\frac{m_i^{t,k}}{\sqrt{\hat{v}_i^{t,k}}} \right\rangle + \frac{\eta_l^2\eta_g^2 K^2 L}{2}$$

$$= -K\eta_g\eta_l\mathbb{E}\left\langle \nabla f(x^t) - q^t, \frac{1}{SK}\sum_{i,k}\frac{m_i^{t,k}}{\sqrt{\hat{v}_i^{t,k}}} \right\rangle - K\eta_g\eta_l\mathbb{E}\left\langle q^t, \frac{1}{SK}\sum_{i,k}\frac{m_i^{t,k}}{\sqrt{\hat{v}_i^{t,k}}} \right\rangle + \frac{\eta_l^2\eta_g^2 K^2 L}{2}$$

$$= K\eta_g\eta_l(\mathbb{E}\|\nabla f(x^t) - q^t\| - \mathbb{E}\|q^t\|) + K\eta_g\eta_l\mathbb{E}\|q^t\|\left\| \frac{1}{SK}\sum_{i,k}\frac{m_i^{t,k}}{\sqrt{\hat{v}_i^{t,k}}} - \frac{q^t}{\|q^t\|} \right\| + \frac{\eta_l^2\eta_g^2 K^2 L}{2} \tag{49}$$

If we let $q = \frac{1}{K}\sum_{k=1}^{K} c^k \nabla f(x^t)$, then we can get:

$$\mathbb{E}f(x^{t+1}) - f(x^t)$$

$$\leq -K\eta_g\eta_l(1 - 2\beta_1^K)\mathbb{E}\|\nabla f(x^t)\| + K\eta_g\eta_l \underbrace{\mathbb{E}\|q^t\|\left\| \frac{1}{SK}\sum_{i,k}\frac{m_i^{t,k}}{\sqrt{\hat{v}_i^{t,k}}} - \frac{q^t}{\|q^t\|} \right\|}_{R_1} + \frac{\eta_l^2\eta_g^2 K^2 L}{2} \tag{50}$$

For $R_1$, we can further bound it as:

$$R_1 = \mathbb{E}\left( \|q^t\|\left\| \frac{1}{SK}\sum_{i,k}\frac{m_i^{t,k}}{\sqrt{\hat{v}_i^{t,k}}} + \frac{m_i^{t,k}}{\|q^t\|} - \frac{m_i^{t,k}}{\|q^t\|} - \frac{q^t}{\|q^t\|} \right\| \right)$$

$$\leq \mathbb{E}\|q^t\|\|m_i^{t,k}\|\left\| \frac{1}{SK}\sum_{i,k}\frac{\|q^t\| - \sqrt{\hat{v}_i^{t,k}}}{\sqrt{\hat{v}_i^{t,k}}\|q^t\|} \right\| + \mathbb{E}\|\frac{1}{SK}\sum_{i,k}m_i^{t,k} - c^k\nabla f(x^t)\|$$

$$\leq \mathbb{E}\|q^t\|\left\| \frac{1}{SK}\sum_{i,k}\frac{\frac{1}{K}\sum_{k=1}^{K}c^k\|q^t\| - \hat{g}_i^{t,k}}{\|q^t\|} \right\| + \mathbb{E}\|\frac{1}{SK}\sum_{i,k}m_i^{t,k} - c^k\nabla f(x^t)\|$$

$$\leq \left( \mathbb{E}\|\frac{1}{SK}\sum_{i,k}\hat{g}_i^{t,k} - \nabla f(x^t)\| + \mathbb{E}\|\frac{1}{SK}\sum_{i,k}m_i^{t,k} - c^k\nabla f(x^t)\| \right) \tag{51}$$

We can then bound $\mathbb{E}\|\frac{1}{SK}\sum_{i,k}\hat{g}_i^{t,k} - \nabla f(x^t)\|$ using $L$-smoothness, and by using the definition of $\gamma_i^{t,k}$ from equation 10 we can get:

$$\mathbb{E}\|\frac{1}{SK}\sum_{i,k}\hat{g}_i^{t,k} - \nabla f(x^t)\| = \mathbb{E}\|\frac{1}{SK}\sum_{i,k}g_i^{t,k} + y^t - y_i^t - \nabla f_i(x^t) + \nabla f_i(x^t) - \nabla f(x^t)\|$$

$$\leq \mathbb{E}\|\frac{1}{SK}\sum_{i,k}g_i^{t,k} - \nabla f_i(x_i^{t,k}) + \nabla f_i(x_i^{t,k}) - \nabla f_i(x^t)\|$$

$$+ \mathbb{E}\|y^t - y_i^t - \nabla f(x^t) + \nabla f_i(x^t)\|$$

$$\leq \underbrace{\sqrt{\mathbb{E}\|\frac{1}{SK}\sum_{i,k}g_i^{t,k} - \nabla f_i(x_i^{t,k}) + \nabla f_i(x_i^{t,k}) - \nabla f_i(x^t)\|^2}}_{R_2}$$

$$+ \frac{2}{nK}\sum_{i,k}\mathbb{E}\|\gamma_i^{t,k} - x^t\| \tag{52}$$

We can further bound $R_2$ with the bound $\eta_l \leq \frac{1}{KL}$ and the fact $\|\frac{m_i^{t,k}}{\|\sqrt{v_i^{t,k}}\|}\| \leq 1$:

$$R_2 = \sqrt{\mathbb{E}\|\frac{1}{SK}\sum_{i,k}g_i^{t,k} - \nabla f_i(x_i^{t,k}) + \nabla f_i(x_i^{t,k}) - \nabla f_i(x^t)\|^2}$$

$$\leq \sqrt{\mathbb{E}\|\frac{1}{SK}\sum_{i,k}g_i^{t,k} - \nabla f_i(x_i^{t,k})\|^2}$$

$$+ \sqrt{\mathbb{E}\langle\frac{1}{SK}\sum_{i,k}g_i^{t,k} - \nabla f_i(x_i^{t,k}), \frac{1}{SK}\sum_{i,k}\nabla f_i(x_i^{t,k}) - \nabla f_i(x^t)\rangle}$$

$$+ \sqrt{\mathbb{E}\|\frac{1}{SK}\sum_{i,k}\nabla f_i(x_i^{t,k}) - \nabla f_i(x^t)\|^2}$$

$$\leq \sqrt{\mathbb{E}\|\frac{1}{SK}\sum_{i,k}g_i^{t,k} - \nabla f_i(x_i^{t,k})\|^2}$$

$$+ \sqrt{\mathbb{E}\|\frac{1}{SK}\sum_{i,k}g_i^{t,k} - \nabla f_i(x_i^{t,k})\|\|\frac{1}{SK}\sum_{i,k}\nabla f_i(x_i^{t,k}) - \nabla f_i(x^t)\|}$$

$$+ \sqrt{\mathbb{E}\|\frac{1}{SK}\sum_{i,k}\nabla f_i(x_i^{t,k}) - \nabla f_i(x^t)\|^2}$$

$$\leq \frac{\sigma}{\sqrt{SK}} + \frac{\sqrt{\sigma\eta_l KL}}{(SK)^{\frac{1}{4}}} + \eta_l KL \tag{53}$$

Combine the result with equation 52 and we get:

$$\mathbb{E}\|\frac{1}{SK}\sum_{i,k}\hat{g}_i^{t,k} - \nabla f(x^t)\| \leq \frac{\sigma}{\sqrt{SK}} + \frac{\sqrt{\sigma\eta_l KL}}{(SK)^{\frac{1}{4}}} + \eta_l KL + \frac{2}{nK}\sum_{i=1}\mathbb{E}\|\gamma_i^{t,k} - x^t\| \tag{54}$$

We can do a similar thing for $\mathbb{E}\|m_i^{t,k} - c^k\nabla f(x^t)\|$:

$$\mathbb{E}\|\frac{1}{SK}\sum_{i,k}m_i^{t,k} - c^k\nabla f(x^t)\| \leq \mathbb{E}\|\frac{1}{SK}\sum_{i,k}\sum_{k'=1}^{k}c^{k,k'}\hat{g}_i^{t,k} - \nabla f(x^t)\| \tag{55}$$

$$\leq \frac{\sigma}{\sqrt{SK}} + \frac{\sqrt{\sigma\eta_l KL}}{(SK)^{\frac{1}{4}}} + \eta_l KL + \frac{2}{nK}\sum_{i,k}\sum_{k'=1}^{k}c^{k,k'}\mathbb{E}\|\gamma_i^{t,k'} - x^t\| \tag{56}$$

By defining the effect of the gradient correction term as $\mathcal{E}^t = \frac{1}{nK}\sum_{i=1}\mathbb{E}\|\gamma_i^{t,k} - x^t\|$ and using Lemma B.1, we get:

$$\mathbb{E}f(x^{t+1}) - f(x^t)$$
$$\leq -K\eta_g\eta_l(1 - 2\beta_1^K)\mathbb{E}\|\nabla f(x^t)\| + 2K^2L\eta_g\eta_l^2 + \frac{K\eta_g\eta_l\sigma}{\sqrt{SK}}$$
$$+ \frac{K\eta_g\eta_l\sqrt{\sigma\eta_lKL}}{(SK)^{\frac{1}{4}}} + 4K\eta_g\eta_l\mathcal{E}^t + \frac{\eta_l^2\eta_g^2K^2L}{2} \tag{57}$$

By constructing a Lyapunov function using $\mathcal{E}^t$ and $f(x)$, we can get the following inequality:

$$\left(\mathbb{E}f(x^{t+1}) + 8\eta_g\eta_lK\frac{n}{Y}\mathcal{E}^{t+1}\right)$$
$$\leq \left(\mathbb{E}f(x^t) + 8\eta_g\eta_lK\frac{n}{Y}\mathcal{E}^t\right) - K\eta_g\eta_l(1 - 2\beta_1^K)\mathbb{E}\|\nabla f(x^t)\| + 2K^2L\eta_g\eta_l^2$$
$$+ \frac{K\eta_g\eta_l\sigma}{\sqrt{SK}} + \frac{K\eta_g\eta_l\sqrt{\sigma\eta_lKL}}{(SK)^{\frac{1}{4}}} + \frac{\eta_l^2\eta_g^2K^2L}{2} + \frac{16n^2}{Y^2}\eta_g^2\eta_l^2K^2 + 2\eta_g\eta_l^2K^2 \tag{58}$$

Then, by unfolding the iterations and letting $y_i^0 = \nabla f_i(x^0)$, we can get:

$$\frac{1}{T}\sum_{t=1}^T\mathbb{E}\|\nabla f(x^t)\| \leq \frac{\mathbb{E}f(x^1) - f^*}{K\eta_g\eta_l(1 - 2\beta_1^K)T} + \frac{\eta_g\eta_lKL}{2(1 - \beta_1^K)} + \frac{16n^2\eta_g\eta_lK}{Y^2(1 - \beta_1^K)}$$
$$+ \frac{2K(1+L)\eta_l}{1 - 2\beta_1^K} + \frac{\sigma}{\sqrt{SK}(1 - 2\beta_1^K)} + \frac{\sqrt{\sigma\eta_lKL}}{(SK)^{\frac{1}{4}}(1 - 2\beta_1^K)} \tag{59}$$

Finally, by letting $\eta_g\eta_l = \min(\frac{\sqrt{\mathcal{F}S}}{\sqrt{\sigma^2KTL}}, \frac{\mathcal{F}}{T})$, $\beta_1 = \sqrt[\kappa]{\frac{KS-2T}{2KS}}$, $\eta_l = \min(\frac{1}{T}, \frac{\mathcal{F}}{K\sqrt{T}})$, we get:

$$\frac{1}{T}\sum_{t=1}^T\mathbb{E}\|\nabla f(x^t)\| \lesssim \frac{\sqrt{L\mathcal{F}\sigma}}{(1 - 2\beta_1)(SKT)^{\frac{1}{4}}} + \frac{L\mathcal{F}}{(1 - 2\beta_1)T} + \frac{LK}{(1 - 2\beta_1)T} + \frac{K\sigma}{T} \tag{60}$$

**Lemma B.1.** *The effect of gradient correction $\mathcal{E}^t$ can be iteratively bounded as:*
$$\mathcal{E}^t \leq (1 - \frac{Y}{2n})\mathcal{E}^{t-1} + 2\frac{n}{Y}\eta_g\eta_lK + 2\frac{Y}{n}\eta_lK \tag{61}$$

*Proof.* Base on the iterative relation of $\gamma_i^{t,k}$, we can show that:

$$\mathbb{E}\|\gamma_i^{t,k} - x^t\|$$
$$\leq (1 - \frac{Y}{n})\mathbb{E}\|\gamma_i^{t-1,k} - x^t\| + \frac{Y}{n}\|s_i^{t-1,k} - x^t\|$$
$$\leq (1 - \frac{Y}{n})(1+b)\|\gamma_i^{t-1,k} - x^{-1}\| + (1 - \frac{Y}{n})\frac{1}{b}\|x^t - x^{t-1}\|$$
$$+ 2\frac{Y}{n}\|x_i^{t-1,k} - x^{t-1}\| + 2\frac{Y}{n}\|x^t - x^{t-1}\|$$
$$= (1 - \frac{Y}{n})(1+b)\|\gamma_i^{t-1,k} - x^{-1}\| + (2\frac{Y}{n} + (1 - \frac{Y}{n})\frac{1}{b})\eta_l\eta_gK + 2\frac{Y}{n}\eta_lK \tag{62}$$

If we let $b = \frac{Y}{2(n-Y)}$, then we can further bound the terms as:

$$\mathbb{E}\|\gamma_i^{t,k} - x^t\| \leq (1 - \frac{Y}{2n})\|\gamma_i^{t-1,k} - x^{t-1}\| + 2\frac{n}{Y}\eta_g\eta_lK + 2\frac{Y}{n}\eta_lK \tag{63}$$

By summing up all $k$ we can yield:

$$\mathcal{E}^t \leq (1 - \frac{Y}{2n})\mathcal{E}^{t-1} + 2\frac{n}{Y}\eta_g\eta_lK + 2\frac{Y}{n}\eta_lK \tag{64}$$

$\square$

## C  THE ALGORITHM AND CONVERGENCE RATE FOR FA-NT

In this section we show the full algorithm for how we implemented Naive Tracking, where the updates is a direct implementation of how SCAFFOLD performs their updates onto a LocalAdam-based FL method.

---

**Algorithm 2:** FA-NT: Federated Adaptive Moment Estimtion with Naive Tracking

---

**Input:** $T$, minibatch size, $|\xi_i^{(t,k)}|$, initial model $x^{(1)}$

1  **each global round** $t = 1, \ldots, T$ **do**
2  $\quad$ randomly sample clients $\mathcal{S}^t \subseteq \{1, \ldots, n\}$.
3  $\quad$ randomly sample clients for update tracking terms $\widetilde{\mathcal{S}}^t \subseteq \mathcal{S}^t$
4  $\quad$ server broadcasts $(x^{(t)}, y^{(t)})$ to all clients $i \in \mathcal{S}^t$
5  $\quad$ **each client** $i \in \mathcal{S}^t$ **in parallel do**
6  $\quad\quad$ $x_i^{(t,1)} = x^{(t)}, m_i^{(t,1)} = 0, v_i^{(t,1)} = v_i^{(t)}$
7  $\quad\quad$ **each local iteration** $k = 1, \ldots, K$ **do**
8  $\quad\quad\quad$ compute mini-batch gradient $g_i^{(t,k)}$, set moment estimation vector $\hat{g}_i^{(t,k)} = g_i^{(t,k)}$
9  $\quad\quad\quad$ Compute first moment $m_i^{(t,k+1)} = \beta_1 m_i^{(t,k)} + (1 - \beta_1)\hat{g}_i^{(t,k)}$, second moment
$\quad\quad\quad$ $v_i^{(t,k+1)} = \beta_2 v_i^{(t,k)} + (1 - \beta_2)\hat{g}_i^{(t,k)} \odot \hat{g}_i^{(t,k)}$, and set $\hat{v}_i^{(t,k+1)} = \max(\hat{v}_i^{(t,k)}, v_i^{(t,k+1)})$
10 $\quad\quad\quad$ Let $\Delta_i^{(t,k)} = m_i^{(t,k+1)}/(\sqrt{\hat{v}_i^{(t,k+1)}} + \epsilon)$, and update
$\quad\quad\quad$ $x_i^{(t,k+1)} = x_i^{(t,k)} - \eta_l(\Delta_i^{(t,k)} + y^{(t)} - y_i^{(t)})$
11 $\quad\quad$ **if** $i \in \widetilde{\mathcal{S}}^t$ **then**
12 $\quad\quad\quad$ $y_i^{(t+1)} = y_i^{(t)} - y^{(t)} + \frac{1}{K\eta_l}(x^{(t)} - x_i^{(t,K+1)})$
13 $\quad\quad$ **else**
14 $\quad\quad\quad$ $y_i^{(t+1)} = y_i^{(t)}$
15 $\quad\quad$ $v_i^{(t+1)} = v_i^{(t,K+1)}$
16 $\quad$ Server aggregates $x_i^{(t,K+1)} - x^{(t)}$ from clients $i \in \mathcal{S}^t$, and $y_i^{(t+1)} - y_i^{(t)}$ from clients $i \in \mathcal{Y}^t$.
17 $\quad$ $x^{(t+1)} = x^{(t)} + \eta_g \frac{1}{S} \sum_{i \in \mathcal{S}^t} (x_i^{(t,K+1)} - x^{(t)})$.
18 $\quad$ $y^{(t+1)} = y^{(t)} + \frac{1}{n} \sum_{i \in \mathcal{Y}^t} (y_i^{(t+1)} - y_i^{(t)})$

---

A potential advantage of `FA-NT` over `FAdamGC` appears when clients have full participation ($S = n$). In this setting, we can define a new correction term $z_i^t = y^t - y_i^t$ that combines the effect of both the global term $y^t$ and the local terms $y_i^t$, and change the update into:

$$x_i^{(t,k+1)} = x_i^{(t,k)} - \eta_l(\Delta_i^{t,k} + z_i^t) \tag{65}$$

$$z_i^{t+1} = z_i^t + \frac{1}{K\eta_g\eta_l}(x_i^{t,K+1} - x^{(t+1)}) \tag{66}$$

After this reformulation, `FA-NT` now only requires transmitting the model parameter $x_i$ between clients and servers, which makes the average communication cost for each client equivalent to `FedAvg` and half the cost of `SCAFFOLD`.

Now, we present the convergence rate of `FA-NT`, both under general $\beta_1, \beta_2$ and under the special case $\beta_2 = \epsilon = 0$. We first introduce an additional assumption on data heterogeneity that is required for our analysis:

**Assumption C.1** (Bounded Data-Heterogeneity). The norm $\|\nabla f_i(x) - \nabla f(x)\|$ is bounded by a constant $B$, i.e., $\|\nabla f_i(x) - \nabla f(x)\| \leq B, \forall x$.

**Theorem C.2.** *Under Assumptions 5.1, 5.3, and the global and local step size satisfies conditions* $\eta_g\eta_l = \min(\frac{(1-\beta_1)\beta_1}{8(G+\epsilon)KL}, \frac{(1-\beta_1)\beta_1}{12(G+\epsilon)TL}, \frac{(1-\beta_1)\beta_1\sqrt{n}}{30(G+\epsilon)\sqrt{T}L})$, *define* $\mathcal{F} = \mathbb{E}f(x^{(1)}) - f^*$, *and consider the following conditions for local step size* $\eta_l$:

$$\eta_l \leq \min\left(\frac{(1-\beta_1)\beta_1\epsilon}{40(G+\epsilon)T^{3/2}L}, \frac{(1-\beta_1)\beta_1\epsilon}{30(G+\epsilon)KL}, \right). \tag{C.2}$$

When satisfying Conditions equation C.2, the iterates of FA-NT can be bounded as:

$$\frac{1}{T}\sum_{t=1}^{T}\mathbb{E}\|\nabla f(x^t)\|^2 = \mathcal{O}\left(\sqrt{\frac{L\mathcal{F}\sigma^2}{nKT}} + \frac{L\mathcal{F}}{T} + \frac{KG^6}{\epsilon^2 T} + \frac{K^2(\sigma^2 + (1+\epsilon^2)G^2)}{\epsilon^2 T}\right). \quad (67)$$

**Theorem C.3.** *Let $\beta_2 = \epsilon = 0$, by selecting $\eta_g\eta_l = \sqrt{\frac{nK}{T}}, \beta_1 = \sqrt[K]{\frac{Kn-2T}{2Kn}}, \eta_l \leq \frac{1}{T}$, under Assumptions 5.1, C.1, the iterates of FA-NT can be bounded as:*

$$\frac{1}{T}\sum_{t=1}^{T}\mathbb{E}\|\nabla f(x^t)\| = \mathcal{O}\left(\sqrt{\frac{L\mathcal{F}}{nKT}} + \frac{L\mathcal{F}}{T} + \frac{LK}{T} + \frac{K(\sigma+nB)}{T}\right) \quad (68)$$

Theorem C.2 shows in general choice of estimation parameters $\beta_1, \beta_2$, FA-NT requires stricter constraints on step sizes than FAdamGC, while from Thorem C.3, we show that under special cases, FA-NT requires more assumptions than FAdamGC to ensure convergence. The detailed proof of both theorems will be in Appendix D and E.

## D    THEORETICAL ANALYSIS OF FA-NT UNDER GENERAL HYPER-PARAMETERS

We first define the following auxilary definitions that will be helpful throughout the proof.

We define $c^k$ as the sum of all moving average coefficients to compute the first order moment $m_i^{(t,k)}$:

$$c^{(k,k')} = (1-\beta_1)\beta_1^{k-k'} \quad (69)$$

$$c^k = \sum_{k'=1}^{k} c^{(k,k')} < 1 \quad (70)$$

We first define the unbiased version of $m_i^{(t,k)}$ taking expectation on all stochastic gradients $g_i^{(t,k)}$. We define $\tilde{m}_i^{(t,k)}$ as the following:

$$\tilde{m}_i^{(t,k)} \triangleq \sum_{k'=1}^{k} c^{(k,k')}\nabla f_i(x_i^{(t,k)}) \quad (71)$$

We define an auxilary variable $\alpha_i^{t,k}$:

$$\alpha_i^{t,k} = \begin{cases} m_i^{t-1,k}/(\sqrt{v_i^{t-1,k}} + \epsilon), & i \in \mathcal{Y}^{t-1} \\ \alpha_i^{t-1,k}, & i \notin \mathcal{Y}^{t-1} \end{cases} \quad (72)$$

We define the tracking variable drift term as:

$$\Gamma^{(t)} = \frac{1}{nK}\sum_{i=1}^{n}\sum_{k=1}^{K}\mathbb{E}\left\|\alpha_i^{t,k} - \nabla f_i(x^{(t)})\right\|^2 \quad (73)$$

We define the local update deviation term as:

$$\mathcal{E}^{(t)} = \frac{1}{n}\sum_{i=1}^{n}\sum_{k=1}^{K}\mathbb{E}\|\tilde{m}_i^{(t,k)} - c^k\nabla f_i(x^{(t)})\|^2 \quad (74)$$

*Proof.* Given global iteration $t$, the update of the model at the server can be written as:

$$x^{(t+1)} = x^{(t)} + \eta_g\frac{1}{S}\sum_{i\in\mathcal{S}^{(t)}}(x_i^{(t,K+1)} - x^{(t)}) \quad (75)$$

$$= x^{(t)} - \eta_g\eta_l\frac{1}{S}\sum_{i\in\mathcal{S}^{(t)}}\sum_{k=1}^{K}\frac{m_i^{(t,k)}}{\sqrt{\hat{v}_i^{(t,k)}} + \epsilon} \quad (76)$$

By injecting Assumption 5.1, we can get the following inequality:

$$\mathbb{E}f(x^{(t+1)}) \leq \mathbb{E}f(x^{(t)}) - \eta_g\eta_l\mathbb{E}\underbrace{\left\langle \nabla f(x^{(t)}), \frac{1}{|\mathcal{S}^{(t)}|}\sum_{i\in\mathcal{S}^{(t)}}\sum_{k=1}^{K}\frac{m_i^{(t,k)}}{\sqrt{\hat{v}_i^{(t,k)}}+\epsilon}\right\rangle}_{\text{Term I}} \tag{77}$$

$$+ \eta_g^2\eta_l^2\frac{L}{2}\mathbb{E}\underbrace{\left\|\frac{1}{|\mathcal{S}^{(t)}|}\sum_{i\in\mathcal{S}^{(t)}}\sum_{k=1}^{K}\frac{m_i^{(t,k)}}{\sqrt{\hat{v}_i^{(t,k)}}+\epsilon}\right\|^2}_{\text{Term II}} \tag{78}$$

For term I, we first define the average of all square root second moment:

$$\bar{v}^{(t)} = \frac{1}{n}\sum_{i=1}^{n}\sqrt{v_i^{(t)}} \tag{79}$$

Then we can upper bound it by Assumption 5.1:

$$-\eta_g\eta_l\mathbb{E}\left\langle \nabla f(x^{(t)}), \frac{1}{S}\sum_{i\in\mathcal{S}^{(t)}}\sum_{k=1}^{K}\frac{m_i^{(t,k)}}{\sqrt{\hat{v}_i^{(t,k)}}+\epsilon}\right\rangle \tag{80}$$

$$-\eta_g\eta_l\mathbb{E}\left\langle \nabla f(x^{(t)}), \frac{1}{S}\sum_{i\in\mathcal{S}^{(t)}}\sum_{k=1}^{K}\frac{\tilde{m}_i^{(t,k)}}{\sqrt{\hat{v}_i^{(t,k)}}+\epsilon}\right\rangle \tag{81}$$

$$= -\eta_g\eta_l\mathbb{E}\left\langle \nabla f(x^{(t)}), \frac{1}{n}\sum_{i=1}^{n}\sum_{k=1}^{K}\left(\frac{\tilde{m}_i^{(t,k)}}{\sqrt{\hat{v}_i^{(t,k)}}+\epsilon} - \frac{\tilde{m}_i^{(t,k)}}{\bar{v}^{(t)}+\epsilon} + \frac{\tilde{m}_i^{(t,k)}}{\bar{v}^{(t)}+\epsilon} - \frac{c^k\nabla f_i(x^{(t)})}{\bar{v}^{(t)}+\epsilon} + \frac{c^k\nabla f_i(x^{(t)})}{\bar{v}^{(t)}+\epsilon}\right)\right\rangle \tag{82}$$

$$\overset{(a)}{\leq} -\eta_g\eta_l K\frac{(1-\beta_1)\beta_1}{G+\epsilon}\mathbb{E}\|\nabla f(x^{(t)})\|^2 \tag{83}$$

$$-\eta_g\eta_l K\mathbb{E}\left\langle \nabla f(x^{(t+1)}), \frac{1}{nK}\sum_{i=1}^{n}\sum_{k=1}^{K}\left(\frac{\tilde{m}_i^{(t,k)}}{\sqrt{\hat{v}_i^{(t,k)}}+\epsilon} - \frac{\tilde{m}_i^{(t,k)}}{\bar{v}^{(t)}+\epsilon} + \frac{\tilde{m}_i^{(t,k)}}{\bar{v}^{(t)}+\epsilon} - \frac{c^k\nabla f_i(x^{(t)})}{\bar{v}^{(t)}+\epsilon}\right)\right\rangle \tag{84}$$

$$\leq -\frac{\eta_g\eta_l K}{2}\frac{(1-\beta_1)\beta_1}{G+\epsilon}\mathbb{E}\|\nabla f(x^{(t)})\|^2 + \eta_g\eta_l\frac{G+\epsilon}{(1-\beta_1)\beta_1\epsilon^2}\frac{1}{n}\sum_{i=1}^{n}\sum_{k=1}^{K}\mathbb{E}\|\tilde{m}_i^{(t,k)} - c^k\nabla f_i(x^{(t)})\|^2 \tag{85}$$

$$+ \eta_g\eta_l K\frac{G+\epsilon}{(1-\beta_1)\beta_1}\mathbb{E}\left\|\frac{1}{nK}\sum_{i=1}^{n}\sum_{k=1}^{K}\frac{\tilde{m}_i^{(t,k)}}{\sqrt{\hat{v}_i^{(t,k)}}+\epsilon} - \frac{\tilde{m}_i^{(t,k)}}{\bar{v}^{(t)}+\epsilon}\right\|^2 \tag{86}$$

$$\leq -\frac{\eta_g\eta_l K}{2}\frac{(1-\beta_1)\beta_1}{G+\epsilon}\mathbb{E}\|\nabla f(x^{(t)})\|^2 + \eta_g\eta_l\frac{G+\epsilon}{(1-\beta_1)\beta_1\epsilon^2}\frac{1}{n}\sum_{i=1}^{n}\sum_{k=1}^{K}\mathbb{E}\|\tilde{m}_i^{(t,k)} - c^k\nabla f_i(x^{(t)})\|^2 \tag{87}$$

$$+ \eta_g\eta_l K\frac{G^2(G+\epsilon)}{(1-\beta_1)\beta_1}\mathbb{E}\left\|\frac{1}{nK}\sum_{i=1}^{n}\sum_{k=1}^{K}\frac{\sqrt{\hat{v}_i^{(t,k)}}-\bar{v}^{(t)}}{(\sqrt{\hat{v}_i^{(t,k)}}+\epsilon)(\bar{v}^{(t)}+\epsilon)}\right\|^2 \tag{88}$$

$$\overset{(b)}{\leq} -\frac{\eta_g\eta_l K}{2}\frac{(1-\beta_1)\beta_1}{G+\epsilon}\mathbb{E}\|\nabla f(x^{(t)})\|^2 + \eta_g\eta_l\frac{G+\epsilon}{(1-\beta_1)\beta_1\epsilon^2}\frac{1}{n}\sum_{i=1}^{n}\sum_{k=1}^{K}\mathbb{E}\|\tilde{m}_i^{(t,k)} - c^k\nabla f_i(x^{(t)})\|^2 \tag{89}$$

$$+ \eta_g \eta_l K \frac{G^2(G+\epsilon)}{(1-\beta_1)\beta_1\epsilon^2} \mathbb{E}\|\bar{v}^{(t+1)} - \bar{v}^{(t)}\|^2 \tag{90}$$

Where (a) the fact that $(1-\beta_1)\beta_1 \leq c^k \leq \beta_1$, and (b) uses the fact that $\hat{v}_i^{(t,1)} \leq \hat{v}_i^{(t,2)} \leq \ldots \leq \hat{v}_i^{(t,K+1)}$.

For term II, we can bound it as:

$$\frac{\eta_g^2\eta_l^2 L}{2} \mathbb{E}\left\|\frac{1}{|\mathcal{S}^{(t)}|}\sum_{i\in\mathcal{S}^{(t)}}\sum_{k=1}^{K}\frac{m_i^{(t,k)}}{\sqrt{\hat{v}_i^{(t,k)}}+\epsilon}\right\|^2 = \eta_g^2\eta_l^2 L\mathbb{E}\left\|\frac{1}{n}\sum_{i=1}^{n}\sum_{k=1}^{K}\frac{\tilde{m}_i^{(t,k)}}{\sqrt{\hat{v}_i^{(t,k)}}+\epsilon}\right\|^2 + \frac{\eta_g^2\eta_l^2 KL\sigma^2}{n} \tag{91}$$

$$= \eta_g^2\eta_l^2 L\mathbb{E}\left\|\frac{1}{n}\sum_{i=1}^{n}\sum_{k=1}^{K}\frac{\tilde{m}_i^{(t,k)}}{\sqrt{\hat{v}_i^{(t,k)}}+\epsilon} - \nabla f_i(x^{(t)}) + \nabla f_i(x^{(t)})\right\|^2 + \frac{\eta_g^2\eta_l^2 KL\sigma^2}{n} \tag{92}$$

$$\leq 2\eta_g^2\eta_l^2 K^2 L\mathbb{E}\|\nabla f(x^{(t)})\|^2 \tag{93}$$

$$+ 2\eta_g^2\eta_l^2 L\frac{K}{n}\sum_{i=1}^{n}\sum_{k=1}^{K}\left\|\frac{\tilde{m}_i^{(t,k)}}{\sqrt{\hat{v}_i^{(t,k)}}+\epsilon} - \frac{c^k\nabla f_i(x^{(t)})}{\sqrt{\hat{v}_i^{(t,k)}}+\epsilon} + \frac{c^k\nabla f_i(x^{(t)})}{\sqrt{\hat{v}_i^{(t,k)}}+\epsilon} - \nabla f_i(x^{(t)})\right\|^2 + \frac{\eta_g^2\eta_l^2 KL\sigma^2}{n} \tag{94}$$

$$\leq 2\eta_g^2\eta_l^2 K^2 L\mathbb{E}\|\nabla f(x^{(t)})\|^2 + \frac{4\eta_g^2\eta_l^2 KL}{\epsilon^2}\frac{1}{n}\sum_{i=1}^{n}\sum_{k=1}^{K}\|\tilde{m}_i^{(t,k)} - c^k\nabla f_i(x^{(t)})\|^2 \tag{95}$$

$$+ \frac{4\eta_g^2\eta_l^2(1-\epsilon)^2}{\epsilon^2}K^2 LG^2 + \frac{\eta_g^2\eta_l^2 KL\sigma^2}{n} \tag{96}$$

If we choose $\eta_g\eta_l \leq \frac{(1-\beta_1)\beta_1}{8KL(G+\epsilon)}$, we can combine Term I and II and get:

$$\mathbb{E}f(x^{(t+1)}) \leq \mathbb{E}f(x^{(t)}) - \frac{\eta_g\eta_l K}{4}\frac{(1-\beta_1)\beta_1}{G+\epsilon}\mathbb{E}\|\nabla f(x^{(t)})\|^2 + \frac{2\eta_g\eta_l}{\epsilon^2}\frac{G+\epsilon}{(1-\beta_1)\beta_1}\mathcal{E}^{(t)} \tag{97}$$

$$+ \eta_g\eta_l K\frac{G^2(G+\epsilon)}{(1-\beta_1)\beta_1\epsilon^2}\mathbb{E}\|\bar{v}^{(t+1)} - \bar{v}^{(t)}\|^2 + \frac{2\eta_g^2\eta_l^2(1-\epsilon)^2}{\epsilon^2}K^2 LG^2 + \frac{\eta_g^2\eta_l^2 KL\sigma^2}{n} \tag{98}$$

By using Lemma D.1, we can formulate the following:

$$\mathbb{E}f(x^{(t+1)}) \leq \mathbb{E}f(x^{(t)})\left(-\frac{\eta_g\eta_l K}{4}\frac{(1-\beta_1)\beta_1}{G+\epsilon} + \frac{2\eta_g\eta_l}{\epsilon^2}\frac{G+\epsilon}{(1-\beta_1)\beta_1}48\eta_l^2 K^3 L^2\right)\mathbb{E}\|\nabla f(x^{(t)})\|^2 \tag{99}$$

$$+ \frac{2\eta_g\eta_l}{\epsilon^2}\frac{G+\epsilon}{(1-\beta_1)\beta_1}96K^3 L^2\eta_l^2\Gamma^{(t)} \tag{100}$$

$$+ \eta_g\eta_l K\frac{G^2(G+\epsilon)}{(1-\beta_1)\beta_1\epsilon^2}\mathbb{E}\|\bar{v}^{(t+1)} - \bar{v}^{(t)}\|^2 \tag{101}$$

$$+ \frac{2\eta_g^2\eta_l^2(1-\epsilon)^2}{\epsilon^2}K^2 LG^2 + \frac{2\eta_g\eta_l}{\epsilon^2}\left(\frac{G+\epsilon}{(1-\beta_1)\beta_1}\right)144K^3 L^2\eta_l^2\left(\frac{G^2+\sigma^2}{\epsilon^2}\right) \tag{102}$$

$$+ \frac{\eta_g^2\eta_l^2 KL\sigma^2}{n} \tag{103}$$

By choosing the local step size as $\eta_l \leq \frac{\epsilon(1-\beta_1)\beta_1}{30(G+\epsilon)KL}$, we can get:

$$\frac{\eta_g\eta_l K}{8}\frac{(1-\beta_1)\beta_1}{G+\epsilon}\mathbb{E}\|\nabla f(x^{(t)})\|^2 \tag{104}$$

$$\leq \mathbb{E}f(x^{(t)}) - \mathbb{E}f(x^{(t+1)}) \tag{105}$$

$$+ \frac{2\eta_g\eta_l}{\epsilon^2}\frac{G+\epsilon}{(1-\beta_1)\beta_1}96K^3L^2\eta_l^2\Gamma^{(t)} \tag{106}$$

$$+ \eta_g\eta_l K\frac{G^2(G+\epsilon)}{(1-\beta_1)\beta_1\epsilon^2}\mathbb{E}\|\bar{v}^{(t+1)} - \bar{v}^{(t)}\|^2 \tag{107}$$

$$+ \frac{2\eta_g^2\eta_l^2(1-\epsilon)^2}{\epsilon^2}K^2LG^2 + \frac{2\eta_g\eta_l}{\epsilon^2}\left(\frac{G+\epsilon}{(1-\beta_1)\beta_1}\right)144K^3L^2\eta_l^2\left(\frac{G^2+\sigma^2}{\epsilon^2}\right) \tag{108}$$

$$+ \frac{\eta_g^2\eta_l^2KL\sigma^2}{n} \tag{109}$$

By moving constants across the inequality and taking average over all iterations, we can get:

$$\frac{1}{T}\sum_{t=1}^{T}\mathbb{E}\|\nabla f(x^{(t)})\|^2 \leq \frac{12(G+\epsilon)(\mathbb{E}f(x^{(1)}) - \mathbb{E}f(x^{(T+1)}))}{\eta_g\eta_l K(1-\beta_1)\beta_1 T} \tag{110}$$

$$+ \frac{12}{K\epsilon^2}\left(\frac{G+\epsilon}{(1-\beta_1)\beta_1}\right)^2 96K^3L^2\eta_l^2\sum_{t=1}^{T}\Gamma^{(t)} \tag{111}$$

$$+ \frac{12KG^2(G+\epsilon)^2}{(1-\beta_1)^2\beta_1^2\epsilon^2 T}\mathbb{E}\|\bar{v}^{(T+1)} - \bar{v}^{(1)}\|^2 \tag{112}$$

$$+ \frac{24\eta_g\eta_l(1-\epsilon)^2KLG^2(G+\epsilon)}{(1-\beta_1)\beta_1\epsilon^2} \tag{113}$$

$$+ \frac{12}{\epsilon^2}\left(\frac{G+\epsilon}{(1-\beta_1)\beta_1}\right)^2 144K^2L^2\eta_l^2\left(\frac{G^2+\sigma^2}{\epsilon^2}\right) \tag{114}$$

$$+ \frac{12\eta_g\eta_l L(G+\epsilon)}{(1-\beta_1)\beta_1 n}\sigma^2 \tag{115}$$

By using Lemma D.2, we can bound $\sum_{t=1}^{T}\Gamma^{(t)}$ with:

$$\sum_{t=1}^{T}\Gamma^{(t)} \leq \sum_{t=1}^{T}(1-\frac{Y}{2n})^{T-t}\Gamma^{(1)} + \sum_{t=1}^{T}t\left(\frac{7n}{Y}G^2 + \frac{Y}{n}\frac{G^2+\sigma^2}{\epsilon^2}\right) \tag{116}$$

$$= T^2\frac{Y}{n}\left(\frac{7n}{Y}G^2 + \frac{Y}{n}\frac{G^2+\sigma^2}{\epsilon^2}\right) \tag{117}$$

Finally, by bounding $\eta_l \leq \frac{(1-\beta_1)\beta_1\epsilon}{12(G+\epsilon)T^{3/2}L}$, $\eta_g\eta_l \leq \frac{(1-\beta_1)\beta_1}{12(G+\epsilon)TL}$, and a specific step size

$$\eta_g\eta_l = \min\left(\frac{(1-\beta_1)\beta_1}{8KL(G+\epsilon)}, \frac{(1-\beta_1)\beta_1}{12(G+\epsilon)TL}, \frac{(G+\epsilon)\sqrt{\mathcal{F}n}}{(1-\beta_1)\beta_1\sigma\sqrt{TKL}}\right) \tag{118}$$

then by defining $\mathcal{F} = \mathbb{E}f(x^1) - f^*$, we can get the convergence rate:

$$\frac{1}{T}\sum_{t=1}^{T}\mathbb{E}\|\nabla f(x^{(t)})\|^2 \lesssim \frac{L\mathcal{F}}{T} + \sqrt{\frac{L\mathcal{F}\sigma^2}{nKT}} \tag{119}$$

$$+ \frac{KG^6}{(1-\beta_1)^2\beta_1^2\epsilon^2 T} + K^2(1+\frac{Y^2}{n^2})\frac{G^2+\epsilon^2 G^2}{\epsilon^2 T} \tag{120}$$

$$+ K^2(1+\frac{Y^2}{n^2})\frac{\sigma^2}{\epsilon^2 T} \tag{121}$$

$$= \mathcal{O}\left(\sqrt{\frac{L\mathcal{F}\sigma^2}{nKT}} + \frac{L\mathcal{F}}{T} + \frac{KG^6}{\epsilon^2 T} + \frac{K^2(\sigma^2+(1+\epsilon^2)G^2)}{\epsilon^2 T}\right) \tag{122}$$

$$\square$$

**Lemma D.1.** *Under Assumption 5.1, the local devation term $\mathcal{E}^{(t)} = \frac{1}{n}\sum_{i=1}^{n}\sum_{k=1}^{K}\mathbb{E}\|\tilde{m}_i^{(t,k)} - c^k\nabla f_i(x^{(t)})\|^2$ can be bounded as the following:*

$$\mathcal{E}^{(t)} \le 48K^3L^2\eta_l^2\mathbb{E}\|\nabla f(x^{(t)})\|^2 + 96K^3L^2\eta_l^2\Gamma^{(t)} + 144K^3L^2\eta_l^2\frac{G^2+\sigma^2}{\epsilon^2} \tag{123}$$

*Proof.*

$$\frac{1}{n}\sum_{i=1}^{n}\sum_{k=1}^{K}\mathbb{E}\|\tilde{m}_i^{(t,k)} - c^k\nabla f_i(x^{(t)})\|^2 \tag{124}$$

$$= \frac{1}{n}\sum_{i=1}^{n}\sum_{k=1}^{K}\mathbb{E}\|\sum_{k'=1}^{k}c^{(k,k')}\left(\nabla f_i(x_i^{(t,k')}) - \nabla f_i(x^{(t)})\right)\|^2 \tag{125}$$

$$\le \frac{L^2}{n}\sum_{i=1}^{n}\sum_{k=1}^{K}\sum_{k'=1}^{k}c^{(k,k')}\mathbb{E}\|x_i^{(t,k')} - x^{(t)}\|^2 \tag{126}$$

We can simplify the formulation by first unfolding each local step $x_i^{(t,k')}$:

$$\frac{1}{n}\sum_{i=1}^{n}\mathbb{E}\|x_i^{(t,k')} - x^{(t)}\|^2 \tag{127}$$

$$\le (1+\frac{1}{K-1})\frac{1}{n}\sum_{i=1}^{n}\mathbb{E}\|x_i^{(t,k'-1)} - x^{(t)}\|^2 \tag{128}$$

$$+ K\frac{1}{n}\sum_{i=1}^{n}\mathbb{E}\left\|\eta_l\left(\frac{m_i^{(t,k'-1)}}{\sqrt{\hat{v}_i^{(t,k'-1)}}+\epsilon} - \nabla f_i(x^{(t)}) + y^{(t)} - y_i^{(t)} + \nabla f_i(x^{(t)}) - \nabla f(x^{(t)}) + \nabla f(x^{(t)})\right)\right\|^2 \tag{129}$$

$$\le (1+\frac{1}{K-1})\frac{1}{n}\sum_{i=1}^{n}\mathbb{E}\|x_i^{(t,k'-1)} - x^{(t)}\|^2 + 3K\eta_l^2\mathbb{E}\|\nabla f(x^{(t)})\|^2 + 3K\eta_l^2\Gamma^{(t)} + \frac{12K\eta_l^2(G^2+\sigma^2)}{\epsilon^2} \tag{130}$$

$$\le \sum_{r=1}^{k'}(1+\frac{1}{K-1})^r\left(4K\eta_l^2\mathbb{E}\|\nabla f(x^{(t)})\|^2 + 8K\eta_l^2\frac{1}{nK}\sum_{i=1}^{n}\sum_{k''=1}^{K}\|\alpha_i^{t,k''} - \nabla f_i(x^{(t)})\|^2\right) \tag{131}$$

$$+ \sum_{r=1}^{k'}(1+\frac{1}{K-1})^r\frac{12K\eta_l^2(G^2+\sigma^2)}{\epsilon^2} \tag{132}$$

Using the fact that $(1+\frac{1}{K-1})^r \le 2e \le 6$, we can get that:

$$\frac{1}{n}\sum_{i=1}^{n}\sum_{k=1}^{K}\mathbb{E}\|m_i^{(t,k)} - c^k\nabla f_i(x^{(t)})\|^2 \le 48K^3L^2\eta_l^2\mathbb{E}\|\nabla f(x^{(t)})\|^2 \tag{133}$$

$$+ 96K^3L^2\eta_l^2\Gamma^{(t)} + 144K^3L^2\eta_l^2\frac{1}{\epsilon^2}(G^2+\sigma^2) \tag{134}$$

$$\square$$

**Lemma D.2.** *Under Assumption 5.1, the tracking variable drift term $\Gamma^{(t)} = \frac{1}{nK}\sum_{i=1}^{n}\sum_{k=1}^{K}\mathbb{E}\left\|\alpha_i^{t,k} - \nabla f_i(x^{(t)})\right\|^2$ can be bounded as:*

$$\Gamma^{(t)} \le (1-\frac{Y}{2n})\Gamma^{(t-1)} + \frac{7n}{Y}G^2 + \frac{Y}{n}\frac{G^2+\sigma^2}{\epsilon^2} \tag{135}$$

*Proof.* By using the definition of $\alpha_i^{t,k}$, we can get the following relation:

$$\Gamma^{(t)} = \frac{1}{nK} \sum_{i=1}^{n} \sum_{k=1}^{K} \mathbb{E} \left\| \alpha_i^{t,k} - \nabla f_i(x^{(t)}) \right\|^2 \tag{136}$$

$$\leq (1 - \frac{Y}{n}) \frac{1}{nK} \sum_{i=1}^{n} \sum_{k=1}^{K} \mathbb{E} \left\| \alpha_i^{t-1,k} - \nabla f_i(x^{(t)}) \right\|^2 \tag{137}$$

$$+ \frac{Y}{n} \frac{1}{nK} \sum_{i=1}^{n} \sum_{k=1}^{K} \mathbb{E} \left\| \frac{m_i^{t-1,k}}{\sqrt{v_i^{t-1,k}} + \epsilon} - \nabla f_i(x^{(t)}) \right\|^2 \tag{138}$$

$$\leq (1 - \frac{Y}{n})(1 + \frac{Y}{2n}) \frac{1}{nK} \sum_{i=1}^{n} \sum_{k=1}^{K} \mathbb{E} \left\| \alpha_i^{t-1,k} - \nabla f_i(x^{(t-1)}) \right\|^2 + (1 - \frac{Y}{n})(1 + \frac{2n}{Y})G^2 \tag{139}$$

$$+ \frac{Y}{n} \frac{1}{nK} \sum_{i=1}^{n} \sum_{k=1}^{K} \mathbb{E} \left\| \frac{m_i^{t-1,k}}{\sqrt{v_i^{t-1,k}} + \epsilon} - \nabla f_i(x^{(t)}) \right\|^2 \tag{140}$$

$$\leq (1 - \frac{Y}{2n})\Gamma^{(t-1)} + (\frac{5n}{Y} + \frac{2Y}{n})G^2 + \frac{Y}{n} \frac{1}{nK} \sum_{i=1}^{n} \sum_{k=1}^{K} \mathbb{E} \left\| \frac{m_i^{t-1,k}}{\sqrt{v_i^{t-1,k}} + \epsilon} \right\|^2 \tag{141}$$

$$\leq (1 - \frac{Y}{2n})\Gamma^{(t-1)} + (\frac{7n}{Y})G^2 \tag{142}$$

$$+ \frac{Y}{n} \frac{1}{nK} \sum_{i=1}^{n} \sum_{k=1}^{K} \mathbb{E} \| \sum_{k'=1}^{k} c^{k,k'} \nabla f_i(x_i^{t-1,k'}, \xi_i^{t-1,k'}) \|^2 \left\| \frac{1}{\sqrt{v_i^{t-1,k}} + \epsilon} \right\|^2 \tag{143}$$

$$\leq (1 - \frac{Y}{2n})\Gamma^{(t-1)} + (\frac{7n}{Y})G^2 + \frac{Y}{n} \frac{G^2 + \sigma^2}{\epsilon^2} \tag{144}$$

$\square$

# E   THEORETICAL ANALYSIS OF FA-NT UNDER $\beta_2 = 0$

With the adaptive stepsize no longer relying on an estimation of the second order moment but the norm of the first order information, we now have $\|\frac{m_i^{t,k}}{\|\sqrt{v_i^{t,k}}\|}\| \leq 1$ for any $\beta_1 \in (0, 1)$. We first write out the update from using $L$-smoothness, we first define an arbitrary vector $q^t \in \mathbb{R}^d$ that will be determined later.

$$\mathbb{E}f(x^{t+1}) - f(x^t) \tag{145}$$

$$\leq -K\eta_g\eta_l \mathbb{E} \left\langle \nabla f(x^t), \frac{1}{SK} \sum_{i,k} \frac{m_i^{t,k}}{\sqrt{\hat{v}_i^{t,k}}} \right\rangle + \frac{\eta_l^2 \eta_g^2 K^2 L}{2} \tag{146}$$

$$= -K\eta_g\eta_l \mathbb{E} \left\langle \nabla f(x^t) - q^t, \frac{1}{SK} \sum_{i,k} \frac{m_i^{t,k}}{\sqrt{\hat{v}_i^{t,k}}} \right\rangle - K\eta_g\eta_l \mathbb{E} \left\langle q^t, \frac{1}{SK} \sum_{i,k} \frac{m_i^{t,k}}{\sqrt{\hat{v}_i^{t,k}}} \right\rangle + \frac{\eta_l^2 \eta_g^2 K^2 L}{2} \tag{147}$$

$$= K\eta_g\eta_l (\mathbb{E}\|\nabla f(x^t) - q^t\| - \mathbb{E}\|q^t\|) + K\eta_g\eta_l \mathbb{E}\|q^t\| \left\| \frac{1}{SK} \sum_{i,k} \frac{m_i^{t,k}}{\sqrt{\hat{v}_i^{t,k}}} - \frac{q^t}{\|q^t\|} \right\| + \frac{\eta_l^2 \eta_g^2 K^2 L}{2} \tag{148}$$

If we let $q = \frac{1}{K} \sum_{k=1}^{K} c^k \nabla f(x^t)$, then we can get:

$$\mathbb{E}f(x^{t+1}) - f(x^t) \tag{149}$$

$$\leq -K\eta_g\eta_l(1-2\beta_1^K)\|\nabla f(x^t)\| + K\eta_g\eta_l \underbrace{\mathbb{E}\|q^t\| \left\|\frac{1}{SK}\sum_{i,k}\frac{m_i^{t,k}}{\sqrt{\hat{v}_i^{t,k}}} - \frac{q^t}{\|q^t\|}\right\|}_{R_1} + \frac{\eta_l^2\eta_g^2 K^2 L}{2} \quad (150)$$

For $R_1$, we can further bound it as:

$$R_1 = \mathbb{E}\|q^t\| \left\|\frac{1}{SK}\sum_{i,k}\frac{m_i^{t,k}}{\sqrt{\hat{v}_i^{t,k}}} + \frac{m_i^{t,k}}{\|q^t\|} - \frac{m_i^{t,k}}{\|q^t\|} - \frac{q^t}{\|q^t\|}\right\| \quad (151)$$

$$\leq \mathbb{E}\|q^t\|\|m_i^{t,k}\| \left\|\frac{1}{SK}\sum_{i,k}\frac{\|q^t\| - \sqrt{\hat{v}_i^{t,k}}}{\sqrt{\hat{v}_i^{t,k}}\|q^t\|}\right\| + \mathbb{E}\|m_i^{t,k} - c^k\nabla f(x^t)\| \quad (152)$$

$$\leq \mathbb{E}\|q^t\| \left\|\frac{1}{SK}\sum_{i,k}\frac{\frac{1}{K}\sum_{k=1}^K c^k\|q^t\| - \hat{g}_i^{t,k}}{\|q^t\|}\right\| + \mathbb{E}\|m_i^{t,k} - c^k\nabla f(x^t)\| \quad (153)$$

$$\leq \left(\mathbb{E}\|\frac{1}{SK}\sum_{i,k}\hat{g}_i^{t,k} - \nabla f(x^t)\| + \mathbb{E}\|\frac{1}{SK}\sum_{i,k}m_i^{t,k} - c^k\nabla f(x^t)\|\right) \quad (154)$$

We can then bound $\mathbb{E}\|\hat{g}_i^{t,k} - \nabla f(x^t)\|$ using $L$-smoothness and bounded-data heterogeneity:

$$E\|\frac{1}{SK}\sum_{i,k}\hat{g}_i^{t,k} - \nabla f(x^t)\| = \mathbb{E}\|\frac{1}{SK}\sum_{i,k}g_i^{t,k} - \nabla f_i(x^t) + \nabla f_i(x^t) - \nabla f(x^t)\| \quad (155)$$

$$\leq \sqrt{\|\frac{1}{SK}\sum_{i,k}g_i^{t,k} - \nabla f_i(x_i^{t,k})\|^2} \quad (156)$$

$$+ \sqrt{\langle\frac{1}{SK}\sum_{i,k}g_i^{t,k} - \nabla f_i(x_i^{t,k}), \frac{1}{SK}\sum_{i,k}\nabla f_i(x_i^{t,k}) - \nabla f_i(x^t)\rangle}$$

$$(157)$$

$$+ \sqrt{\|\frac{1}{SK}\sum_{i,k}\nabla f_i(x_i^{t,k}) - \nabla f_i(x^t)\|^2 + B} \quad (158)$$

$$\leq \sqrt{\|\frac{1}{SK}\sum_{i,k}g_i^{t,k} - \nabla f_i(x_i^{t,k})\|^2} \quad (159)$$

$$+ \sqrt{L\frac{1}{SK}\sum_{i,k}\|\Delta_i^{t,k} - y^t + y_i^t\|\|\frac{1}{SK}\sum_{i,k}\nabla f_i(x_i^{t,k}) - \nabla f_i(x^t)\|}$$

$$(160)$$

$$+ \sqrt{\|\frac{1}{SK}\sum_{i,k}\nabla f_i(x_i^{t,k}) - \nabla f_i(x^t)\|^2 + B} \quad (161)$$

$$\overset{(a)}{\leq} \frac{\sigma}{\sqrt{SK}} + \frac{\sqrt{\sigma\eta_l 3KL}}{(SK)^{\frac{1}{4}}} + \eta_l 3KL + B \quad (162)$$

Where (a) holds true by initializating $y_i^0 = \nabla f_i(x^0)/\|\nabla f_i(x^0)\|$, then we can get $\|y_i^t\| \leq 1$ and $\|y^t\| \leq 1$ for any $t \geq 0$.

We can do a similar thing for $\mathbb{E}\|m_i^{t,k} - c^k\nabla f(x^t)\|$:

$$\mathbb{E}\|\frac{1}{SK}\sum_{i,k}m_i^{t,k} - c^k\nabla f(x^t)\| \leq \mathbb{E}\|\frac{1}{SK}\sum_{i,k}\sum_{k'=1}^k c^{k,k'}\hat{g}_i^{t,k} - \nabla f(x^t)\| \quad (163)$$

$$\leq \frac{\sigma}{\sqrt{SK}} + \frac{\sqrt{\sigma\eta_l 3KL}}{(SK)^{\frac{1}{4}}} + \eta_l 3KL + B \tag{164}$$

By combining the results above into equation 150, we can get:

$$\frac{1}{T}\sum_{t=1}^{T}\mathbb{E}\|\nabla f(x^t)\| \lesssim \frac{\mathbb{E}f(x^1) - f^*}{K\eta_g\eta_l(1 - 2\beta^K)T} + \frac{\eta_g\eta_l KL}{2(1 - 2\beta_1^K)} + \frac{3\eta_l KL}{1 - 2\beta^K} + \frac{K(\sigma/\sqrt{S} + B)}{(1 - 2\beta_1^K)} + \frac{\sqrt{\sigma\eta_l KL}}{(SK)^{\frac{1}{4}}(1 - 2\beta_1^K)} \tag{165}$$

Finally, by letting $\eta_g\eta_l = \min(\frac{\sqrt{\mathcal{F}S}}{\sqrt{\sigma^2 KTL}}, \frac{\mathcal{F}}{T})$, $\beta = \sqrt[K]{\frac{KS - 2T}{2KS}}$, $\eta_l \leq \min(\frac{1}{T}, \frac{\mathcal{F}}{K\sqrt{T}})$, we can get:

$$\frac{1}{T}\sum_{t=1}^{T}\mathbb{E}\|\nabla f(x^t)\| \lesssim \frac{\sqrt{L\mathcal{F}\sigma}}{(1 - 2\beta_1)(SKT)^{\frac{1}{4}}} + \frac{L\mathcal{F}}{(1 - \beta_1)T} + \frac{KL}{(1 - 2\beta_1)T} + \frac{K(\sigma + \sqrt{S}B)}{T} \tag{166}$$

# F    ADDITIONAL EXPERIMENTS ON CIFAR DATASETS

In this section we plot more training results on CIFAR datasets under different sampling rate and different choice of local iterations $K$. Compare between Figure 5 and Figure 6, we can see that although `FAdamGC` outperforms `FA-NT` in most cases, there are still certain scenarios (sample rate = 10%, $K = 10$) where Naive Tracking seems to perform better than GC. However, as the sample rate increases, in both $K = 10$ and $K = 60$ set of experiments, `FAdamGC` gains more steady improvement. Similar observation can also be found from experiments on CIFAR10 in Figure 7 and 8.

Additionally, we present the mean and variance of the training curves computed over four random trials in Figure 10. The results indicate that our method achieves the lowest variance across diverse data heterogeneity conditions, highlighting its robustness and training stability.

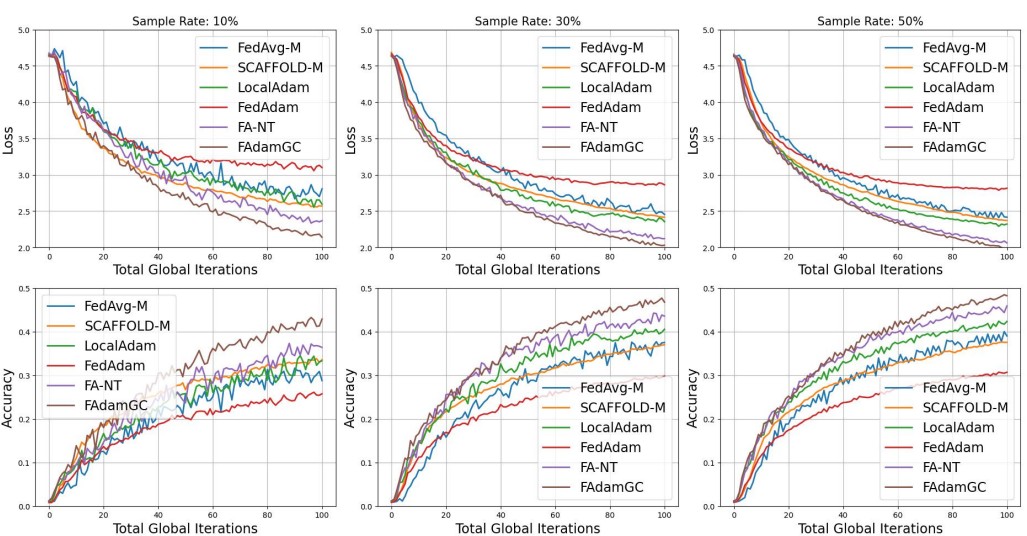

Figure 5: Experimental results on CIFAR100 under different sample rate of clients and $K = 60$.

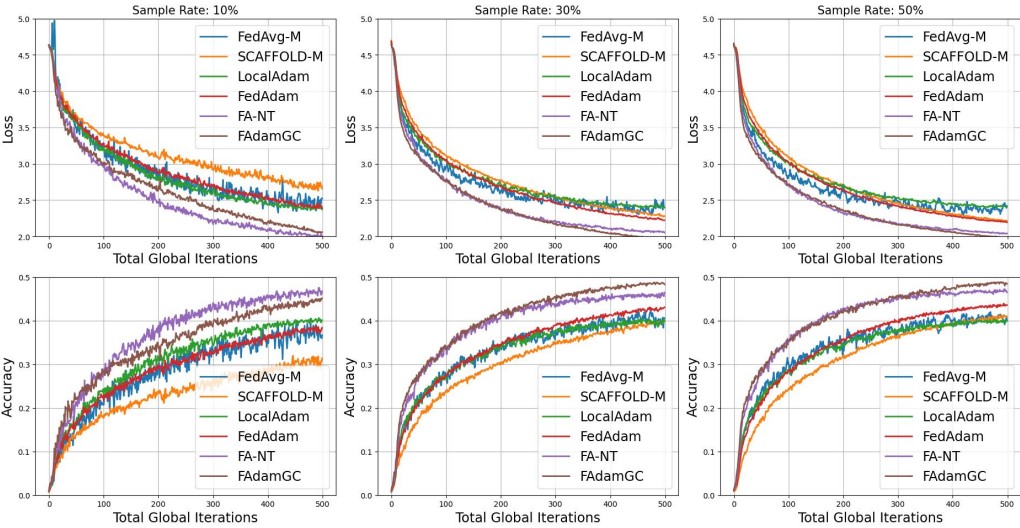

Figure 6: Experimental results on CIFAR100 under different sample rate of clients and $K = 10$.

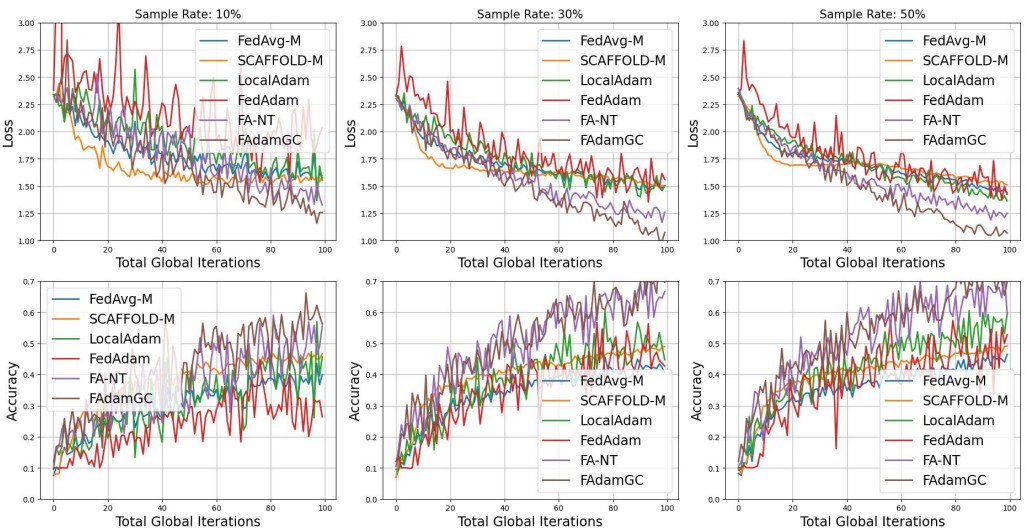

Figure 7: Experimental results on CIFAR10 under different sample rate of clients and $K = 60$.

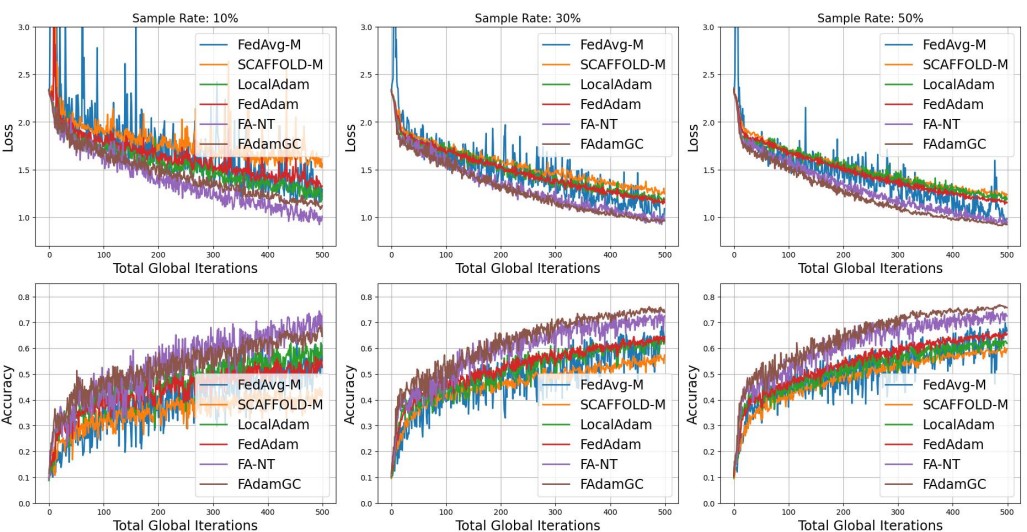

Figure 8: Experimental results on CIFAR10 under different sample rate of clients and $K = 10$.

# G  COMPARISON BETWEEN $\beta_2 = 0$ AND NON ZERO $\beta_2$ IN FADAMGC AND FA-NT

With Theorems 5.2 and C.3 established, a natural question arises: *Is second-moment estimation necessary in federated learning?* While our analysis demonstrates that setting $\beta_2 = 0$ allows for convergence under weaker assumptions, empirical results consistently show improved performance when $\beta_2 > 0$. This suggests that second-moment information remains valuable in practice, and that tighter theoretical guarantees for FAdamGC and FA-NT may be attainable, particularly if future analysis can bypass the need for bounded gradient assumptions. We show in Table 3 that of all proposed methods, a large $\beta_2$ consistly outperforms the case of $\beta_2 = 0$.

# H  CHOSEN HYPERPARAMETERS

We showed all the learning rate we used in Sec. 6, obtained through grid search.

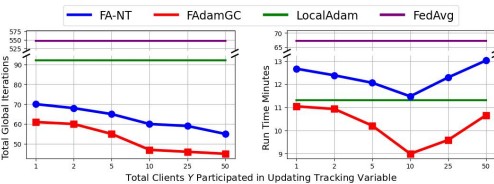

Figure 9: Comparison of total cost to attain certain accuracy between different tracking sampling rate TinyImageNet with $S = 50$, where the target accuracy is 30%.

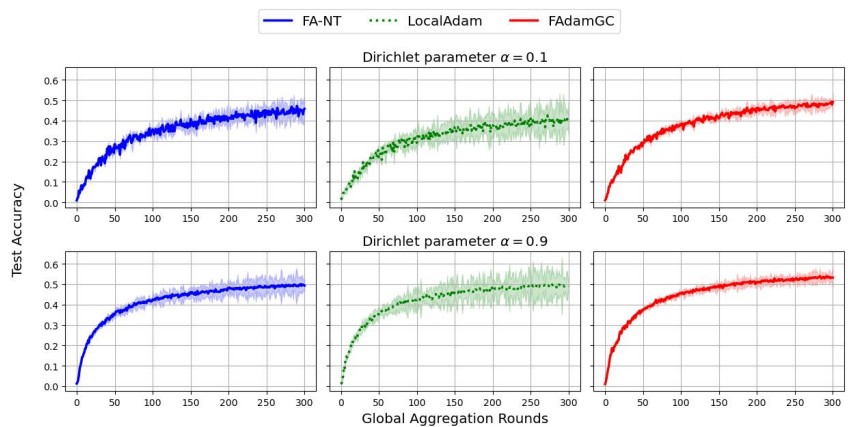

Figure 10: Mean and variance across 4 random trials under different data heterogeneity.

Table 3: The comparison of our methods under different constraints on $\beta_2$ values

| Settings | Dataset | LocalAdam | FA-NT ($\beta_2 = 0$) | FAdamGC ($\beta_2 = 0$) | FAdamGC ($\beta_2 = 0.3$) | FAdamGC ($\beta_2 = 0.7$) | FA-NT | FAdamGC |
|---|---|---|---|---|---|---|---|---|
| Total Communication Rounds | CIFAR-10 | $589.5_{\pm74.0}$ | $401.5_{\pm33.9}$ | $358.8_{\pm21.4}$ | $334.3_{\pm20.2}$ | $318.5_{\pm15.3}$ | $394.8_{\pm31.3}$ | $\mathbf{310.0_{\pm16.8}}$ |
| | CIFAR-100 | $678.3_{\pm40.6}$ | $867.5_{\pm25.3}$ | $527.0_{\pm16.5}$ | $413.3_{\pm15.5}$ | $370.5_{\pm14.7}$ | $530.3_{\pm17.6}$ | $\mathbf{323.8_{\pm16.3}}$ |
| | TinyImageNet | $177.3_{\pm8.3}$ | $164.3_{\pm6.7}$ | $85.5_{\pm5.7}$ | $75.3_{\pm5.7}$ | $74.0_{\pm6.6}$ | $157.0_{\pm6.4}$ | $\mathbf{66.3_{\pm4.4}}$ |
| Simulated Run Time (minutes) | CIFAR-10 | $72.38_{\pm74.0}$ | $83.44_{\pm33.9}$ | $74.56_{\pm21.4}$ | $71.78_{\pm4.34}$ | $68.44_{\pm4.34}$ | $82.07_{\pm31.3}$ | $\mathbf{64.42_{\pm16.8}}$ |
| | CIFAR-100 | $83.36_{\pm40.6}$ | $180.27_{\pm25.3}$ | $109.56_{\pm16.5}$ | $88.79_{\pm3.33}$ | $79.57_{\pm3.16}$ | $110.21_{\pm17.6}$ | $\mathbf{67.2_{\pm16.3}}$ |
| | TinyImageNet | $21.78_{\pm8.3}$ | $34.15_{\pm6.7}$ | $17.77_{\pm5.7}$ | $16.17_{\pm1.22}$ | $15.89_{\pm1.42}$ | $32.63_{\pm6.4}$ | $\mathbf{13.78_{\pm4.4}}$ |

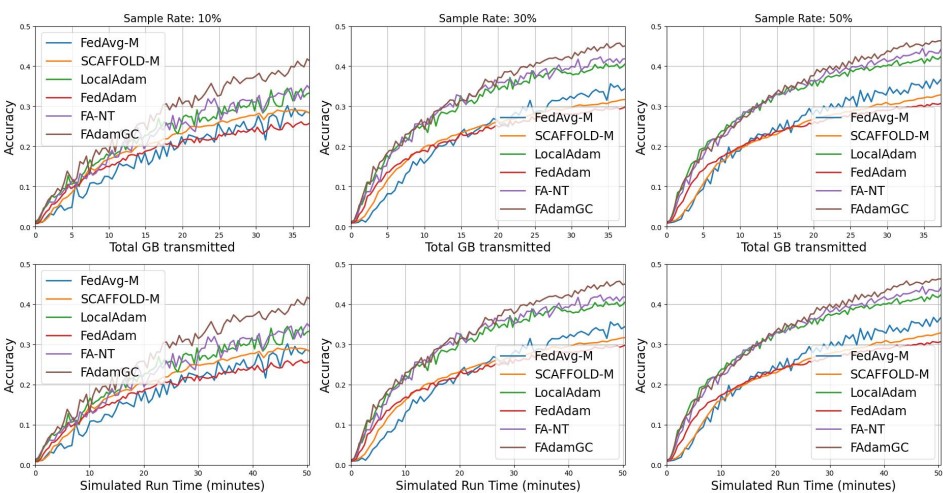

Figure 11: Results on CIFAR100 under $K = 10$ evaluated under Total transmitted GBs and Simulated Run Time (SRT).

Table 4: The hyperparameters for image classification tasks.

| Learning Rate | Dataset | FedAvg-M | SCAFFOLD-M | FedAdam | FedAMS | LocalAdam | FA-NT | FAdamGC |
|---|---|---|---|---|---|---|---|---|
| $\eta_g$ | CIFAR-10 | 1 | 1 | $1 \times 10^{-3}$ | $1 \times 10^{-3}$ | 1 | 1 | 1 |
| | CIFAR-100 | 1 | 1 | $1 \times 10^{-3}$ | $1 \times 10^{-3}$ | 1 | 1 | 1 |
| | TinyImageNet | $3 \times 10^{-1}$ | $3 \times 10^{-1}$ | $1 \times 10^{-3}$ | $1 \times 10^{-3}$ | $1 \times 10^{-1}$ | $3 \times 10^{-1}$ | $3 \times 10^{-1}$ |
| $\eta_l$ | CIFAR-10 | $1 \times 10^{-2}$ | $1 \times 10^{-2}$ | $3 \times 10^{-2}$ | $3 \times 10^{-2}$ | $1 \times 10^{-3}$ | $1 \times 10^{-3}$ | $1 \times 10^{-3}$ |
| | CIFAR-100 | $1 \times 10^{-2}$ | $1 \times 10^{-2}$ | $3 \times 10^{-2}$ | $3 \times 10^{-2}$ | $1 \times 10^{-3}$ | $1 \times 10^{-3}$ | $1 \times 10^{-3}$ |
| | TinyImageNet | $1 \times 10^{-2}$ | $1 \times 10^{-2}$ | $1 \times 10^{-2}$ | $1 \times 10^{-2}$ | $1 \times 10^{-3}$ | $1 \times 10^{-3}$ | $1 \times 10^{-3}$ |

Table 5: The hyperparameters for language tasks.

| Learning Rate | Dataset | FedAvg-M | SCAFFOLD-M | FedAdam | FedAMS | LocalAdam | FA-NT | FAdamGC |
|---|---|---|---|---|---|---|---|---|
| $\eta_g$ | 20NEWSGROUPS | $1 \times 10^{-1}$ | $1 \times 10^{-1}$ | $1 \times 10^{-2}$ | $1 \times 10^{-2}$ | $1 \times 10^{-1}$ | $1 \times 10^{-1}$ | $1 \times 10^{-1}$ |
| | QQP | $1 \times 10^{-1}$ | $1 \times 10^{-1}$ | $1 \times 10^{-3}$ | $1 \times 10^{-3}$ | $3 \times 10^{-1}$ | $3 \times 10^{-1}$ | $3 \times 10^{-1}$ |
| | QNLI | $1 \times 10^{-1}$ | $1 \times 10^{-1}$ | $1 \times 10^{-3}$ | $1 \times 10^{-3}$ | $3 \times 10^{-1}$ | $3 \times 10^{-1}$ | $3 \times 10^{-1}$ |
| | SST-2 | $1 \times 10^{-1}$ | $1 \times 10^{-1}$ | $1 \times 10^{-3}$ | $1 \times 10^{-3}$ | $3 \times 10^{-1}$ | $3 \times 10^{-1}$ | $3 \times 10^{-1}$ |
| $\eta_l$ | 20NEWSGROUPS | $1 \times 10^{-2}$ | $1 \times 10^{-2}$ | $1 \times 10^{-3}$ | $1 \times 10^{-3}$ | $5 \times 10^{-3}$ | $5 \times 10^{-3}$ | $5 \times 10^{-3}$ |
| | QQP | $1 \times 10^{-2}$ | $1 \times 10^{-2}$ | $3 \times 10^{-4}$ | $3 \times 10^{-4}$ | $1 \times 10^{-4}$ | $1 \times 10^{-4}$ | $1 \times 10^{-4}$ |
| | QNLI | $1 \times 10^{-2}$ | $1 \times 10^{-2}$ | $3 \times 10^{-4}$ | $3 \times 10^{-4}$ | $3 \times 10^{-5}$ | $3 \times 10^{-5}$ | $3 \times 10^{-5}$ |
| | SST-2 | $1 \times 10^{-2}$ | $1 \times 10^{-2}$ | $3 \times 10^{-4}$ | $3 \times 10^{-4}$ | $1 \times 10^{-3}$ | $1 \times 10^{-3}$ | $1 \times 10^{-3}$ |

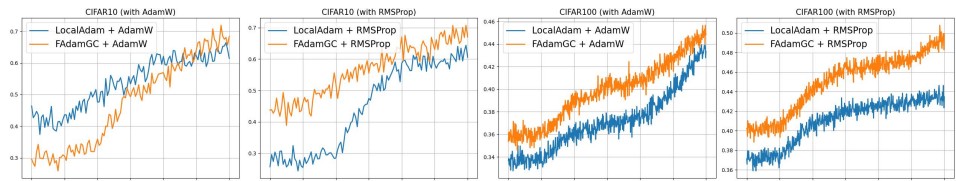

Figure 12: Performance of FAdamGC variants and LocalAdam variants over different time steps on CIFAR-10 dataset, the step sizes are chosen over a grid search.

# I COMMUNICATION AND COMPUTATION COST EVALUATION

## I.1 STORAGE OVERHEAD ON CLIENTS

In `FAdamGC`, each client locally maintains weights, gradients, and first/second-order moments (standard in adaptive methods such as Adam), plus one additional tracking vector. The global tracking term is broadcast and not persisted locally. Therefore, compared to LocalAdam, our method requires only one extra vector of model size. In large-scale settings, this remains practical under low-rank finetuning (e.g., LoRA), which we adopt in our LLM experiments. In Table 6, we compare the total amount of parameters required to be stored in the memory during local training on edge devices. We can observe that when the parameters are stored in FP32, the total memory usage varies between 0.1 to 0.5 GB, which is reasonable for most existing edge devices such as the Nvidia Jetson Nano or Raspberry Pi.

## I.2 COMMUNICATION VOLUME METRIC AND RESULTS

We quantify communication cost using a pure communication volume metric:

$$\text{CommVol} = \left( \sum_{t=1}^{T} N_t^{\text{up}} + N_t^{\text{down}} \right) \times \text{bytes}(\theta),$$

where $N_t^{\text{up}}$ and $N_t^{\text{down}}$ are the numbers of model-sized tensors transmitted (uplink and downlink) between server and clients at round $t$, and $\text{bytes}(\theta)$ is the size (in bytes) of the deployed model (ResNet-18 here). This yields a direct measure of total gigabytes (GB) transmitted until a target accuracy is reached, independent of local computation. As shown in Table 7, `FAdamGC` consistently incurs the lowest communication volume, outperforming both adaptive and non-adaptive baselines under this metric.

Table 6: Total parameters stored in memory per client during local training (counts shown as number of scalars; approximate memory in parentheses assumes FP32 at 4 bytes/parameter).

| Task | FedAvg-M | SCAFFOLD-M | FedAdam | FedAMS | LocalAdam | FAdamGC |
|---|---|---|---|---|---|---|
| Image task (ResNet-18) | 35.1M ($\sim$0.14 GB) | 46.8M ($\sim$0.18 GB) | 23.4M ($\sim$0.09 GB) | 23.4M ($\sim$0.09 GB) | 46.8M ($\sim$0.18 GB) | 58.5M ($\sim$0.23 GB) |
| Language task | 125.48M ($\sim$0.50 GB) | 126.22M ($\sim$0.50 GB) | 124.74M ($\sim$0.49 GB) | 124.74M ($\sim$0.49 GB) | 126.22M ($\sim$0.50 GB) | 126.96M ($\sim$0.51 GB) |

Table 7: Total communication volume in GB transmitted until a target accuracy is reached for ResNet-18. Values are mean $\pm$ std.

| Dataset | FedAvg-M | SCAFFOLD-M | FedAdam | FedAMS | LocalAdam | FA-NT | FAdamGC |
|---|---|---|---|---|---|---|---|
| CIFAR-10 | $132.53 \pm 32.70$ | $94.80 \pm 10.43$ | $220.71 \pm 29.94$ | $208.05 \pm 24.91$ | $51.34 \pm 6.44$ | $60.15 \pm 4.78$ | $47.22 \pm 2.47$ |
| CIFAR-100 | $130.37 \pm 11.56$ | $108.25 \pm 9.64$ | $161.41 \pm 16.14$ | $144.15 \pm 13.62$ | $59.04 \pm 3.80$ | $80.76 \pm 2.72$ | $49.34 \pm 2.49$ |
| TinyImageNet | $28.08 \pm 1.36$ | $42.14 \pm 2.68$ | $47.23 \pm 3.93$ | $40.29 \pm 3.10$ | $15.41 \pm 0.72$ | $23.87 \pm 0.97$ | $10.08 \pm 0.67$ |

Additionally, Fig. 11 presents the training curves of multiple algorithms evaluated in terms of total transmitted gigabytes and simulated run time. We can observe that even considering the communication overhead of gradient correction methods, across different numbers of sampled clients, `FAdamGC` still consistently outperforms existing methods. This demonstrates both stability and communication efficiency of the combination of gradient correction and adaptive optimization in FL.

## J GENERALIZATION OF GRADIENT CORRECTION CONCEPT

The pre-moment gradient correction of `FAdamGC` is readily applicable to other adaptive methods whose updates can be summarized as "estimate statistics of the gradient, then normalize the step". This includes RMSProp and AdamW (with decoupled weight decay handled in a straightforward way). The fixed-point consistency and the key steps of our proof extend with minor, mechanical changes (mainly to the way second-moment estimators are bounded). Under gradient correction, the update rule for RMSProp under our correction framework becomes: $x_i^{t,k+1} = x_i^{t,k} - \eta_l \frac{g_i^{t,k} - y_i^t + y^t}{\sqrt{\hat{v}_i^{t,k}} + \epsilon}$ where $g_i^{t,k}$ is the local stochastic gradient and $\hat{v}_i^{t,k}$ is the running average of squared gradients. For AdamW, the correction operates identically to `FAdamGC`, with the only difference being the application of decoupled weight decay after the gradient computation. This decoupling does not interfere with the correction logic or analysis, and the fixed-point condition still holds.

Figure 12 presents `FAdamGC`'s convergence behavior when applied to various adaptive optimizers. We observe that all adaptive optimizers exhibit comparable convergence trends and achieve stable convergence. We can also see that FAdamGC steadily outperforms LocalAdam under every optimizer, showing the robust improvement gradient correction introduces to adaptive optimizers.

