# OpenReview forum: "Gradient Correction in Federated Learning with Adaptive Optimization"
_ICLR.cc/2026/Conference — Submitted to ICLR 2026_

### Official Review · Reviewer_XP38 · 2025-10-19

**Soundness:** 3
**Presentation:** 2
**Contribution:** 3
**Rating:** 6
**Confidence:** 3

**Summary:**

This paper addresses the problem of client-drift caused by data heterogeneity in FL with adaptive optimizers. The authors propose FAdamGC, the first algorithm to integrate gradient-tracking-based drift compensation into adaptive federated optimization. The authors conduct a rigorous theoretical convergence analysis under non-convex settings and perform extensive empirical studies, showing that their method outperforms prior correction schemes in terms of communication and computation efficiency across varying data heterogeneity.

**Strengths:**

1. FAdamGC is the first work to consider integrating gradient-tracking-based client-drift compensation with adaptive optimizers in federated learning, thereby addressing a meaningful challenge in the field.
2. The proposed pre-estimation correction method is technically sound: it ensures that the gradient correction term aligns with the complex update format of adaptive methods (which a naive “post-estimation” approach fails to achieve).
3. The paper presents a rigorous convergence analysis and extensive empirical studies, which together convincingly demonstrate the superior performance of the proposed method.

**Weaknesses:**

The writing and experimental setup of this work can be further improved, please refer to the question part.

**Questions:**

1. For the system model: in order to be rigorous, it is recommended to write $f_i$ in the form of the empirical risk and defining $\mathcal{D}_i$ as the empirical distribution constructed by i.i.d. sampling from the local data distribution — that is, the local training set.
2. Regarding the Adam‐optimizer used in this work (lines 160–161): it appears that the correction terms (bias correction) have not been included. Would adopting the bias‐corrected version of Adam affect the theoretical analysis in this paper?
3. In the section “Problem with naive application of compensation,” the authors could provide further explanation as to why — in the SGD setting — despite the heterogeneity of data and thus $\nabla f_i(x^*) \neq 0$, SGD can still use the original correction term.
4. For the experimental part, it is recommended to use momentum for estimating the gradient‐correction term, and then compare this method against the “pre-estimation correction term” method proposed in this paper.
5. The authors could also consider including, in the communication‐rounds/simulated-time versus loss/accuracy curves, the mean values and corresponding standard deviations across multiple runs, in order to reflect the robustness of different algorithms/settings.

---

> ### Author Response · Authors · 2025-11-19
>
> We appreciate the reviewer’s time, effort, and constructive feedback on our manuscript.
>
> > ### question 1
> >  For the system model: in order to be rigorous, it is recommended to write $f_i$ in the form of the empirical risk and defining
> $\mathcal{D}_i$ as the empirical distribution constructed by i.i.d. sampling from the local data distribution — that is, the local training set.
>
> We have updated the system model section accordingly in the revised manuscript.
>
> ```
> The problem we aim to solve follows the standard FL formulation:
> \[
>     \textstyle \min_{x\in \mathbb{R}^d}f(x) &\textstyle= \frac{1}{n}\sum_{i=1}^n f_i(x),
> \]
> where  $n$ is the total number of clients (typically edge devices) in the system, indexed $i = 1, \ldots, n$.  $ f_i(x) = \mathbb{E}_{\xi_i \sim \mathcal{D}_i}[f_i(x; \xi_i)] $ is the empirical risk at client $ i $ for model parameters $ x \in \mathbb{R}^d $, where $ \xi_i $ is an unbiased random sample drawn from the local empirical data distribution $\mathcal{D}_i$, constructed from the client’s local training dataset. We assume the server is directly connected
> to each device as in the conventional FL architecture.
> ```
> > ### question 2
> > Regarding the Adam‐optimizer used in this work (lines 160–161): it appears that the correction terms (bias correction) have not been included. Would adopting the bias‐corrected version of Adam affect the theoretical analysis in this paper?
>
> No, the only thing that will change will be the value of $c^{(k)}$ that is used in the proof of all theorems. This value will now equal one rather than being strictly less than one, which will not effect the final convergence bound.
>
> > ### question 3
> > In the section “Problem with naive application of compensation,” the authors could provide further explanation as to why — in the SGD setting — despite the heterogeneity of data and thus $\nabla f_i(x^*) \neq 0$, SGD can still use the original correction term.
>
> In the SGD setting, although each client’s gradient at the global optimum is generally non-zero due to data heterogeneity ($\nabla f_i(x^\*) \neq 0$), the ideal correction term converges to $\nabla f(x^\*) - \nabla f_i(x^\*)$. As a result, the fixed-point update for SGD becomes:
> $$
> x^\* = x^\* - \eta_l ( \nabla f_i(x^\*) + \nabla f(x^\*) - \nabla f_i(x^\*) ).
> $$
> This explains why the original correction term suffices in SGD despite heterogeneity, as it effectively cancels out the bias induced by local client gradients at the optimum.
>
> > ### question 4
> > For the experimental part, it is recommended to use momentum for estimating the gradient‐correction term, and then compare this method against the “pre-estimation correction term” method proposed in this paper.
>
> We appreciate the reviewer’s constructive suggestion to use momentum for estimating the gradient-correction term and to compare it against our proposed method. This indeed amounts to designing a new variant of the algorithm, which could potentially offer performance improvements. We have implement and currently running experiments on this alternative approach and will include the discussion in the revised manuscript before the end of the discussion period.
>
> > ### question 5
> > The authors could also consider including, in the communication‐rounds/simulated-time versus loss/accuracy curves, the mean values and corresponding standard deviations across multiple runs, in order to reflect the robustness of different algorithms/settings
>
> We appreciate the reviewer's valuable suggestion. While including mean and variance in all plots would provide a comprehensive view of robustness, it also significantly increases visual clutter and reduces clarity in key figures. To balance detail and readability, we have included a subset of plots with mean and standard deviation curves in the appendix with additional discussion (see Appendix F). These supplementary plots demonstrate the low inter-run variance and consistent trends across different datasets, reinforcing our claims about robustness.
>
> ```
> Additionally, we present the mean and variance of the training curves computed over four random trials in Figure 10. The results indicate that our method achieves the lowest variance across diverse data heterogeneity conditions, highlighting its robustness and training stability.
> ```

---

### Official Review · Reviewer_Gf9b · 2025-10-30

**Soundness:** 1
**Presentation:** 2
**Contribution:** 1
**Rating:** 0
**Confidence:** 4

**Summary:**

The paper introduces FAdamGC, a novel algorithm to mitigate client drift in federated learning with adaptive optimizers like Adam. Arguing that naive correction fails, the authors propose a "pre-estimation" correction applied to raw gradients before moment computation. They provide convergence analysis and extensive experiments on image and language tasks to demonstrate FAdamGC's superior performance and stability in heterogeneous settings

**Strengths:**

1. Addresses the critical and timely problem of making adaptive optimizers robust to data heterogeneity in FL.
2. Presents extensive and convincing empirical results across diverse tasks (CNNs, LLM PEFT) and metrics, consistently outperforming strong baselines.

**Weaknesses:**

1. (Major) The central claim of providing the first convergence guarantee for an adaptive FL method without a bounded gradient assumption appears to be incorrect. The paper "Problem-Parameter-Free Federated Learning" has already presented an adaptive FL algorithm and proven its convergence without this assumption. This significantly diminishes the claimed novelty of Theorem 5.2. The authors must acknowledge this prior art, thoroughly discuss it, and compare their theoretical framework, proof techniques, and resulting convergence rates. The current framing of the theoretical contribution is misleading and needs to be fundamentally revised.

[1] Yan, Wenjing, et al. "Problem-Parameter-Free Federated Learning." The Thirteenth International Conference on Learning Representations. 2025.

2. The core algorithmic idea, termed "Gradient Correction" or "pre-estimation correction," where control variates are used to correct the raw gradient before it enters the moment accumulators, is presented as a key contribution. However, this exact mechanism was already proposed and analyzed in the paper "[1] Problem-Parameter-Free Federated Learning." The current paper fails to cite and compare against this work.

3. The main theoretical result that forgoes the bounded gradient assumption (Theorem 5.2) is derived for the special case where β2 = 0. This setting reduces the optimizer to a method more akin to momentum with normalization, rather than the full Adam algorithm. Theorem 5.4. is based on bounded gradient assumption, which seems more strong. The paper should be more transparent about this limitation in the main text and abstract, as the current presentation might imply the result holds for the general Adam optimizer.

**Questions:**

1. What are the primary technical hurdles that prevent the extension of the bounded-gradient-free analysis from the β2 = 0 case to the general case where β2 > 0?

2. The empirical results for the language tasks (e.g., QQP, QNLI) show exceptionally large performance gains for FAdamGC. Do the authors have a hypothesis for why this specific correction mechanism is so particularly effective in the context of Parameter-Efficient Fine-Tuning (PEFT) with LoRA?

3. In Figure 4, it is shown that the number of clients participating in tracking updates (Se) can be reduced significantly with only a minor impact on performance. Is there a theoretical relationship between the degree of data heterogeneity (α) and the minimum required Se to maintain stable convergence?

---

> ### Author Response · Authors · 2025-11-13
> **Replies to major comments**
>
> We would like to begin by addressing this reviewer’s comments, as we greatly value the feedback and believe it will be instrumental in guiding our rebuttal and subsequent revisions.
>
> We thank the reviewer for highlighting the related work [1], which was inadvertently omitted in our original submission. After carefully reviewing the paper, we will revise our manuscript to include a detailed discussion of its approach and clarify the distinctions between our framework and theirs.
>
> We note the following similarities between the two works:
>
> ---
> 1.  The algorithm in [1] performs a gradient normalization during local updates without any adaptive estimation, this makes it conceptually similar to our algorithm's special case $\beta_2 = 0$.
>
> 2.  Because both use normalized gradients rather than adaptive second-moment estimation ($\beta_2=0$), the convergence guarantees are analogous, as Theorem 1 in [1] and Theorem 5.2 in our work both avoid the bounded-gradient assumption by leveraging similar normalization properties.
> ---
>
> These overlaps mean our work is not the first to present a pre-estimation gradient correction with normalization. Nonetheless, our study pursues different objectives and introduces distinct analyses and mechanisms to further advance adaptive federated optimization:
>
> ---
> ## 1. Generalized Framework with $\beta_2$ Parameter (Adaptive Moment Estimation).
>
> Although both methods are momentum-based algorithms with gradient normalization, our method retains full **Adam-style adaptive moment estimation** ($\beta_1, \beta_2 \in [0,1)$), which is essential in practice—particularly as $\beta_2 \to 1$ for stable and smooth convergence across diverse edge conditions. Our theoretical analysis also rigorously covers this generalized setting, whereas [1] focuses on the $\beta_2 = 0$ with only gradient normalization (whose empirical counterpart is already included in Table 3)
>
> ## 2. Federated-Specific Design and Selective Tracking Mechanism.
>
> A very important focus of our work the potential communication and computation overhead such adaptive optimization with control variate may bring to FL, thus our approach introduces a selective tracking mechanism that determines which clients update both model parameters and control variates. This design, unique to our framework, enables an explicit trade-off between convergence efficiency and communication overhead (see Fig.5). The work in [1] assumes identical client participation for all updates and does not address this federated-level optimization control, while our work has a thorough discussion on the communication benefits using multiple metrics such as simulated runtime and communication volume metric as shown in Table 2, Figure 2,3,4 and, Appendix G,I,J.
>
>  ## 3. Theoretical Analysis of Post- vs. Pre-Estimation Correction.
>
> We provide four new theorems (Thm 5.2,5.4 for pre-estimation, and Thm C.2, C.3 for post-estimation) and accompanying empirical analyses illustrating why **post-estimation correction** leads to unstable convergence in adaptive FL, while **pre-estimation correction** yields provable stability and improved performance. To our knowledge, such a theoretical and experimental comparison of the two correction paradigms has not been discussed in prior works, including [1].
>
> ## 4. Comprehensive Evaluation Across Adaptive Optimizers and FL Settings.
>
> Although the main text focuses on Adam-based results for clarity, our framework is designed to be able to generalize naturally to other adaptive optimizers by directly replacing the swapping the Adam-based operations in Algorithm 1 (see Appendix J). We also include experiments comparing directly against the $\beta_2 = 0$ variant numerically corresponding to the method in [1] as reported in Table 3. These results consistently show that our full adaptive formulation achieves higher accuracy and faster convergence under identical conditions.
>
> ---
> We have revised our claims to remove the statement of being the first to prove convergence of adaptive FL without bounded gradients and have incorporated [1] into both theoretical and experimental comparisons. The new experimental results will be added once simulations are completed. We now clarify that our contributions lie in: (i) extending pre-estimation correction to adaptive moment methods, (ii) introducing selective client tracking for communication-efficient FL, and (iii) providing theoretical and empirical insights into when pre- and post-estimation correction succeed or fail. All modified text has been marked in blue.
>
> We sincerely appreciate the reviewer’s valuable comments, which have improved the clarity and positioning of our work. Please let us know if this response sufficiently addresses your concerns or if further clarification would be helpful.
>
> [1] Yan, Wenjing, et al. "Problem-Parameter-Free Federated Learning." The Thirteenth International Conference on Learning Representations. 2025.

---

> ### Author Response · Authors · 2025-11-13
> **Additional response to questions**
>
> We would also like to clarify the following questions regarding our work:
>
> > What are the primary technical hurdles that prevent the extension of the bounded-gradient-free analysis from the β2 = 0 case to the general case where β2 > 0?
>
> The key enabler for bounded-gradient-free analysis is gradient normalization, whereby the normalized vector $g/\|g\|$ always has unit norm and is thus inherently bounded regardless of the underlying gradient's magnitude. However, this boundedness property is lost when introducing adaptive second-moment estimation ($\beta_2 > 0$), as the update direction now depends on both the gradient and its running squared average, potentially leading to arbitrarily scaled steps if either moments are large or small. Consequently, the critical step for the analysis no longer applies, and extending the bounded-gradient-free guarantees to the general Adam setting presents substantial technical obstacles.
>
> > The empirical results for the language tasks (e.g., QQP, QNLI) show exceptionally large performance gains for FAdamGC. Do the authors have a hypothesis for why this specific correction mechanism is so particularly effective in the context of Parameter-Efficient Fine-Tuning (PEFT) with LoRA?
>
> One possible explanation for the strong empirical gains from our gradient correction mechanism in PEFT settings like LoRA is the inherent structure and heterogeneity of language data. While our synthetic heterogeneity setup used labels to construct Dirichlet distributions for both the image tasks and NLP tasks, the true heterogeneity in NLP also contributed from variability in input text itself (across clients or data partitions), which can cause more pronounced client drift and parameter bias during supervised fine-tuning than commonly seen in image or tabular tasks. As such, the gradient correction technique is particularly effective at mitigating these larger discrepancies, resulting in the substantial improvement observed for language tasks such as QQP and QNLI.
>
> > In Figure 4, it is shown that the number of clients participating in tracking updates (Se) can be reduced significantly with only a minor impact on performance. Is there a theoretical relationship between the degree of data heterogeneity (α) and the minimum required Se to maintain stable convergence?
>
> Currently, we do not have a formal theoretical relationship quantifying how data heterogeneity ($\alpha$) determines the minimum required subset of participating clients for robust convergence. Our selective tracking scheme was primarily motivated by empirical observations of the tradeoff between communication cost and performance. The development of such a theory remains an important direction for future work.
>
> ---
>
> We sincerely thank the reviewer for these insightful and technically questions, which have helped us better articulate the theoretical limitations, empirical observations, and future research directions of our work.

---

> ### Comment · Reviewer_Gf9b · 2025-11-27
> **Reply to Author-Response**
>
> Thank you for your thoughtful response. While I appreciate the clarifications provided, I must respectfully maintain my concerns about the paper's contributions, which remain insufficiently addressed:
>
> 1. Algorithmic Novelty: This algorithm in this paper seems a straightforward application of ADAM to federated learning. The incremental nature of the algorithmic contribution remains a fundamental weakness.
>
> 2. Theoretical Rigor: This paper only provides theoretical guarantees for the $\beta_2=0$ case and the result is weaker than existing work [1]—severely undermines the theoretical contribution. The authors should either:
>
> Extend their theory to the general β₂>0 case, or
> Clearly justify their theoretical advantages than [1].
>
> 3. The statement that the algorithm in [1] is a special case of algorithm in this paper when $\beta_2 = 0$ is particularly concerning, due to the distinction between the gradient norm (in [1]) versus point-wise squared gradients (in this paper).
>
>
> 4. Experimental Completeness: Given that [1] is not a special case of the proposed method, its absence from experimental comparisons is a critical omission. All experiments must include [1] as a baseline for fair evaluation.
>
> 5. Contribution Assessment:
>
> The first contribution (adaptive learning) already exists in [1]
> Selective client tracking appears to be a minor engineering detail rather than a fundamental advance
> While the comparison of correction paradigms has merit, it alone cannot carry the paper

---

> ### Author Response · Authors · 2025-11-27
>
> We sincerely thank the reviewer for these insightful comments. Our detailed responses are as follows:
>
> **Reply to 2.** We would like to clarify that our theoretical analysis is **not restricted to the case $\beta_2 = 0$**. The manuscript provides complete theoretical results for **both $ \beta_2 = 0 $ and $\beta_2 > 0 $**. This new analysis is one of our key contributions that distinguishes our work from [1], and we will revise the text to ensure this point is made explicit.
>
> **Reply to 3.** In both our prior replies and the revised manuscript, our intention was only to note that the $ \beta_2 = 0 $ special case of our method and that of [1] are *conceptually similar*, rather than identical. We will further refine the wording to avoid any ambiguity.
>
> **Reply to 4.** We have already updated Table 2 accordingly to include comparison with [1] and will continue revising the experimental section to include comprehensive comparisons with [1] across all experiments.
>
> **Reply to 1 and 5.** While we acknowledge that our work shares certain foundational ideas with [1], our contribution includes additional development. Specifically, we provide an extensive discussion of pre- and post-estimation correction terms, along with new theoretical results on pre-estimation strategies that highlight their advantages. These aspects are unique to our work and not investigated in [1].
>
> We appreciate the reviewer’s constructive feedback and will incorporate these clarifications and revisions in the updated manuscript.
>
>
> [1] Yan, Wenjing, et al. "Problem-Parameter-Free Federated Learning." The Thirteenth International Conference on Learning Representations. 2025

---

> > ### Comment · Reviewer_Gf9b · 2025-11-28
> >
> > Thank you for your clarification. I now understand that the theorem applies when β > 0, though it requires the restrictive bounded gradient assumption. I will reconsider my score after reviewing the feedback and discussions from other reviewers.

---

### Official Review · Reviewer_k38c · 2025-10-31

**Soundness:** 3
**Presentation:** 3
**Contribution:** 2
**Rating:** 4
**Confidence:** 4

**Summary:**

This paper introduces FAdamGC, a novel federated optimization algorithm that integrates client-drift compensation into adaptive methods like Adam. The key idea is to inject a "pre-estimation" gradient correction term before the moment accumulation step to mitigate the effects of data heterogeneity. The authors provide a convergence analysis for non-convex settings, demonstrating that FAdamGC achieves a standard O(1/√SKT) rate without requiring a bounded data heterogeneity assumption in certain cases. Comprehensive experiments on both image classification and LLM fine-tuning tasks show that FAdamGC outperforms existing SGD-based and adaptive FL baselines.

**Strengths:**

* The paper is well-written and clearly organized, making the proposed method and its analysis easy to follow.
* The experimental evaluation is comprehensive, covering both image classification and large language model (LLM) fine-tuning tasks against several relevant baselines.
* The core idea of injecting a gradient correction term before moment accumulation is intuitive and well-motivated.

**Weaknesses:**

* The method requires transmitting both model parameters (x) and correction terms (y) in each round, which doubles the communication payload compared to methods like LocalAdam. While the paper provides Simulated Run Time (SRT) and total volume metrics, a direct plot of accuracy versus communication bits/volume would more clearly illustrate the efficiency trade-offs.
* The paper empirically shows that β₂ > 0 is better than β₂ = 0, but an intuitive explanation for why the second-moment information is particularly beneficial in this corrected framework is expected.
* When β₂ = 0, the update in Eq. (2) resembles gradient normalization, making the algorithm conceptually similar to the one in [a]. Consequently, the convergence analysis in Theorem 5.2 and the empirical performance would be expected to be similar. A direct comparison and discussion with this highly relevant work are needed to better position the paper's contribution.
* Following the point above, if a key contribution is the inclusion of the second-moment term (β₂ > 0), its impact needs clearer justification. In the experiments, β₂ is set to 0.99, which is very close to 1. Additionally, an ablation study on the value of β₂ would be beneficial to understand its sensitivity and demonstrate its importance.

[a] Yan, W., Zhang, K., Wang, X., & Cao, X. (2025). Problem-Parameter-Free Federated Learning. In The Thirteenth International Conference on Learning Representations.

**Questions:**

* Table 1 provides a theoretical comparison of convergence rates. What would be the theoretical convergence rate for SCAFFOLD-M, and how would it compare to FAdamGC? This would provide a more complete picture, especially since SCAFFOLD-M is a key baseline in the experiments.
* In lines 251-257 and Eq. (4), the paper argues that injecting the correction term ∇f(x*) − ∇fi(x*) makes the global optimum x* a fixed point. Could the authors provide a more detailed intuition for this? Furthermore, why here the correction is applied to the gradient for the first moment (m) but not for the second moment (v) in the denominator?

---

> ### Author Response · Authors · 2025-11-19
> **responses (1/3)**
>
> We thank the reviewer for the time, effort, and valuable feedback.
>
> > ### weakness 1
> The paper empirically shows that $\beta_2 > 0$  is better than $\beta_2$ = 0, but an intuitive explanation for why the second-moment information is particularly beneficial in this corrected framework is expected.
>
> We appreciate the reviewer’s suggestion.
> Including second-moment information ($\beta_2 > 0$) stabilizes corrected updates in two key ways. First, it adaptively scales gradients, preventing large or erratic steps. Second, the estimation process makes it gain a level of robustness against data heterogeneity, as shown in the experiments, the Adam-based methods consistently have better performance than the SGD-based methods. This effect is absent when $\beta_2 = 0$, which explains the empirical performance gap. A new paragraph has been added to the revised manuscript to clarify this connection to standard Adam-style interpretations.
>
> ```
> Including second-moment information ($\beta_2 > 0$) further stabilizes corrected updates by adaptively scaling gradients to prevent erratic steps and by enhancing robustness against data heterogeneity, as evidenced by consistently better performance of Adam-based methods over SGD-based ones. This benefit disappears when $\beta_2 = 0$, explaining the observed empirical performance gap between {\tt FAdaMFed} and {\tt FAdamGC}.
> ```
>
> > ### weakness 2
> > The method requires transmitting both model parameters (x) and correction terms (y) in each round, which doubles the communication payload compared to methods like LocalAdam. While the paper provides Simulated Run Time (SRT) and total volume metrics, a direct plot of accuracy versus communication bits/volume would more clearly illustrate the efficiency trade-offs.
>
> Thank you for this helpful suggestion. We agree that illustrating the trade-off between accuracy and communication volume would make the comparison clearer. In the revised manuscript, we have added Figure 11 showing accuracy versus total transmitted bits for all methods, including ours. This figure directly visualizes the communication–performance relationship and complements the existing SRT and total-volume metrics. We have also added the following discussion into the appendix.
>
> ```
> Additionally, Fig. 11 presents the training curves of multiple algorithms evaluated in terms of total transmitted gigabytes and simulated run time. We can observe that even considering the communication overhead of gradient correction methods, across different numbers of sampled clients, {\tt FAdamGC} still consistently outperforms existing methods. This demonstrates both stability and communication efficiency of the combination of gradient correction and adaptive optimization in FL.
> ```

---

> ### Author Response · Authors · 2025-11-19
> **responses (2/3)**
>
> > ### weakness 3
> > When $\beta_2$ = 0, the update in Eq. (2) resembles gradient normalization, making the algorithm conceptually similar to the one in [a]. ... discussion with this highly relevant work are needed to better position the paper's contribution.
>
> We thank the reviewer for pointing out this important connection. We agree that when $\beta_2 = 0$, the update in Eq. (2) becomes closely related to gradient normalization and therefore resembles the method proposed in [1]. In the revised manuscript, we now explicitly discuss this connection, revised our contributions, and cite [1] accordingly. We state four main differences between the two works: 1) Our algorithm is a generalized framework with flexible choice of $\beta_2$ design, which allows smoother second-moment upates to stabilize training performance. 2) Aiming to enable communication efficient FL, we introduce selective tracking, enabling explicit trade-offs between communication and gradient correction updates. 3) Our theory compares pre- and post-estimation corrections and why post-estimation fails in adaptive optimization, showing stability advantages not analyzed in [1]. 4) We thoroughly evaluate across optimizers other than Adam, demonstrating consistent superior accuracy and convergence. We have updated our contributions in the revised manuscript to reflect these differences.
>
> ```
> - We propose FAdamGC, an Adam-based federated optimization algorithm stabilized with a novel gradient correction mechanism. By leveraging control variables to track global gradient information and implementing a selective client tracking scheme to enhance communication efficiency, FAdamGC compensates for client drift internally without the need for extra fine-tuning, efficiently mitigating model biases caused by non-i.i.d. data distributions in FL (Sec. 4.2). Furthermore, our analysis provides both theoretical and empirical insights into difference across pre- and post-estimation correction strategies (Sec. 5).
>
> - We conduct a rigorous convergence analysis of our proposed algorithm, producing both a convergence guarantee for specialized gradient normalization without relying on the bounded gradient assumption and also generalized convergence guarantee for adaptive federated optimization. Our analysis provides insights into the stability and convergence speedup achieved by FAdamGC under data heterogeneity, and clarifies the distinct impact of applying parameter tracking at different stages of the local update process (Sec. 5).
>
> - We perform extensive experiments of FAdamGC across diverse datasets and multiple FL settings, including image classification tasks using CNNs and sequence classification tasks using LLMs. Our results demonstrate substantial improvements in training accuracy and resource utilization compared with baselines under varying levels of non-i.i.d. client data distributions (Sec. 6).
> ```
>
> > ### weakness 4
> >  Following the point above, if a key contribution is the inclusion of the second-moment term ($\beta_2 > 0$), its impact needs clearer justification. In the experiments, $\beta_2$ is set to 0.99, which is very close to 1. Additionally, an ablation study on the value of $\beta_2$ would be beneficial to understand its sensitivity and demonstrate its importance.''
>
> We have conducted an additional ablation study spanning a range of $\beta_2$ values ($0$, $0.3$, $0.7$) on three benchmark datasets. The results have been updated in Table 3, which indicate a consistent and substantial improvement in both communication efficiency and runtime as $\beta_2$ increases from 0 to moderate values.
>
> These findings demonstrate that the inclusion of a nonzero second-moment term is not only theoretically motivated, but also empirically impactful for improving convergence and communication performance. We thank the reviewer for prompting this important analysis.
>
> | Settings  | Dataset | LocalAdam | FA-NT ($\beta_2=0$) | FAdamGC ($\beta_2=0$) |FAdamGC ($\beta_2=0.3$) | FAdamGC ($\beta_2=0.7$) | FA-NT  | FAdamGC |
> |-|-|-|--|-|-|-|-|-|
> | **Total Communication Rounds** | CIFAR-10| 589.5 ± 74.0| 401.5 ± 23.9| 358.8 ± 21.4 | 334.3 ± 20.2| 318.8 ± 15.3 | 394.8 ± 31.3    | 310.0 ± 16.8|
> || CIFAR-100    | 678.3 ± 40.6|867.5 ± 25.3| 527.0 ± 16.5 | 413.3 ± 15.5 | 370.5 ± 14.7| 530.3 ± 17.6 | 323.8 ± 16.3|
> || TinyImageNet | 177.3 ± 8.3|164.3 ± 6.7| 85.5 ± 5.7| 75.3 ± 5.7| 74.0 ± 6.6| 157.0 ± 6.4| 66.3 ± 4.4 |
> | **Simulated Run Time (minutes)** | CIFAR-10| 72.38 ± 74.0| 83.44 ± 33.9| 74.56 ± 21.4| 71.78 ± 4.34| 68.44 ± 4.34| 82.07 ± 31.3| 64.42 ± 16.8|
> || CIFAR-100|83.36 ± 40.6|180.27 ± 25.3|109.56 ± 16.5|88.79 ± 3.33|79.57 ± 3.16| 110.21 ± 17.6   | 67.2 ± 16.3|
> || TinyImageNet | 21.78 ± 8.3  | 34.15 ± 6.7  | 17.77 ± 5.7| 16.17 ± 1.22| 15.89 ± 1.42| 32.63 ± 6.4| 13.78 ± 4.4|
>
> [1] Yan, Wenjing, et al. "Problem-Parameter-Free Federated Learning." The Thirteenth International Conference on Learning Representations. 2025.

---

> ### Author Response · Authors · 2025-11-19
> **responses (3/3)**
>
> > ### question 1
> > Table 1 provides a theoretical comparison of convergence rates. What would be the theoretical convergence rate for SCAFFOLD-M, and how would it compare to FAdamGC? This would provide a more complete picture, especially since SCAFFOLD-M is a key baseline in the experiments.
>
> Table 1 presents a theoretical comparison of convergence rates across various adaptive methods, primarily to highlight that the bounded-gradient assumption is necessary for establishing general results in adaptive optimization. Although SCAFFOLD-M does not include an adaptive component, we acknowledge that discussing its convergence rate remains important. The rate for SCAFFOLD-M can be expressed as $\left(\frac{L\mathcal{F}\sigma^2}{nKT}\right)^{\frac{1}{2}} + \frac{L\mathcal{F}}{T}\left(1 + n^{-\frac{1}{3}}\right)$. This shows that SCAFFOLD-M achieves an $\mathcal{O}(1/\sqrt{nKT})$ convergence rate without requiring data heterogeneity or bounded gradient assumptions. However, this property does not directly extend to the family of adaptive methods because the analysis for adaptive optimizers must account for second moment terms, which fundamentally alter the convergence behavior compare to SCAFFOLD-M. We have included the SCAFFOLD-M rate in the revised Table 1, accompanied by a footnote clarifying that it is not an adaptive method.
>
> | Algorithms                              | Convergence Rate                                                                                                                                   | Additional Assumptions |
> |--|--|-|
> | SCAFFOLD-M| $$(\frac{L\mathcal{F}\sigma^2}{nKT})^{\frac{1}{2}} + \frac{L\mathcal{F}}{T}(1 + n^{-\frac{1}{3}})$$                                                 | -                     |
> | FedAdam        | $$(\frac{\mathcal{F}^2}{nKT})^{\frac{1}{2}} + \frac{L\sigma^2}{G^2\sqrt{nKT}} + \frac{\sigma^2}{GKT} + \frac{L\sigma^2\sqrt{n}}{G^2\sqrt{K}T^{3/2}} $$ | BG                    |
> | FedAMS        | $$(\frac{\mathcal{F}^2}{nKT})^{\frac{1}{2}} + \frac{L\sqrt{nK}G^2}{\sqrt{\epsilon}T} + \frac{L^2 K \sigma^2}{T} + \frac{G\sigma^2}{\sqrt{\epsilon^2nK}T}$$ | BG                    |
> | PAdaMFed           | $$\frac{(L + \sigma + \sqrt{L\sigma} + \mathcal{F})^2}{(nKT)^{\frac{1}{2}}} +\frac{(\sqrt{nK}\sigma + L)^2}{T}$$                                    | -                     |
> | FA-NT ($$\beta_2=\epsilon=0$$)| $$\frac{ L\mathcal{F}\sigma}{(nKT)^{\frac{1}{2}}} + \frac{(L\mathcal{F})^2}{T^2}+\frac{K^2(\sigma + L + nB)^2}{T^2}$$                                | BDH                   |
> | FA-NT          | $$(\frac{L\mathcal{F}\sigma^2}{nKT})^{\frac{1}{2}} + \frac{L\mathcal{F}}{T} +\frac{KG^6}{\epsilon^2T}+\frac{K^2(\sigma^2+(1+\epsilon^2)G^2)}{\epsilon^2T}$$ | BG                    |
> | FAdamGC ($$\beta_2 =\epsilon=0$$ }) | $$\frac{ L\mathcal{F}\sigma}{(nKT)^{\frac{1}{2}}}+ \frac{(L\mathcal{F})^2}{T^2}+\frac{K^2(\sigma + L)^2}{T}$$                                        | -                     |
> | FAdamGC      | $$(\frac{L\mathcal{F}\sigma^2}{nKT})^{\frac{1}{2}} + \frac{L\mathcal{F}}{T} +\frac{KG^6}{\epsilon^2T}+\frac{K(\sigma^2+(1+\epsilon^2)G^2)}{\epsilon^2T}$$  | BG                    |
>
> > ### question 2
> > In lines 251-257 and Eq. (4), the paper argues that injecting the correction term ∇f(x*) − ∇fi(x*) makes the global optimum x* a fixed point. Could the authors provide a more detailed intuition for this? Furthermore, why here the correction is applied to the gradient for the first moment (m) but not for the second moment (v) in the denominator?
>
> The ideal correction term, after adjusting the local gradient, makes the numerator zero. This effectively reduces equation (4) to $x^* = x^* - \eta_l \times 0$, thereby ensuring that $x^*$ is a fixed point solution. In practice, the second moment should also be iteratively corrected to guarantee stable and proper convergence. However, when focusing purely on the fixed-point properties, the key point to demonstrate is that to make the numerator of the update direction zero, the correction must be applied before the estimation—i.e., directly on the gradient—rather than after the estimation.

---

### Official Review · Reviewer_t86o · 2025-11-01

**Soundness:** 3
**Presentation:** 3
**Contribution:** 3
**Rating:** 6
**Confidence:** 2

**Summary:**

The paper introduces a variant of the Adam optimizer that incorporates client-drift compensation, making it suitable for federated learning settings. This approach aims to address the limitations of naive client-drift handling and improves upon existing adaptive methods such as FedAvg and FA-NT.

**Strengths:**

The main contributions include (1) an algorithm with a novel gradient correction mechanism designed to stabilize updates across heterogeneous clients, (2) a theoretical convergence analysis that supports the proposed method, and (3) experimental results that demonstrate its effectiveness. The paper is clearly written and well-organized, making it easy to follow the motivation, methodology, and results. The authors present their technical novelty in a transparent and understandable manner.

**Weaknesses:**

That said, the literature review appears rather limited. The discussion could be strengthened by including a broader comparison with recent federated optimization approaches and by justifying the choice of Adam as the base optimizer. In modern machine learning applications, alternative optimizers such as Muon or other momentum-based methods often demonstrate superior empirical performance, so the rationale for building upon Adam should be clarified.

In terms of experiments, the reported test accuracies are surprisingly low, which is unexpected for methods that explicitly address client-drift. It would be helpful if the authors could provide an explanation—perhaps related to dataset characteristics, hyperparameter choices, or implementation details. Overall, the work is clearly presented and methodologically sound, but the empirical results and contextualization within the broader literature could be improved.

Note: there are a few formatting issues (margin violations), authors should double-check for those

**Questions:**

It would be valuable if the authors could address the questions and concerns outlined in the weaknesses section.

---

> ### Author Response · Authors · 2025-11-19
>
> Thank you for the time, effort, and all the valuable feedbacks!
>
> > ### weakness 1
> > That said, the literature review appears rather limited. The discussion could be strengthened by including a broader comparison with recent federated optimization approaches and by justifying the choice of Adam as the base optimizer. In modern machine learning applications, alternative optimizers such as Muon or other momentum-based methods often demonstrate superior empirical performance, so the rationale for building upon Adam should be clarified.
>
> We appreciate the reviewer’s suggestion to broaden the literature review by including newer optimizers such as Muon. The choice of focusing on Adam is mainly because Adam has traditionally served as a widely adopted baseline for federated adaptive methods. This allows us to compare the effectiveness of gradient correction with existing FL Adam-based works, while also offering a solid foundation for our gradient correction approach both empirically and theoretically. That said, our algorithmic framework is designed to be compatible with a broader class of adaptive optimizers. Indeed, we have already demonstrated effectiveness with other optimizers in Appendix J.
>
> Nonetheless, we have now included relevant recent literature on Muon and other advanced optimization methods in the revised manuscript. Related experiments using Muon are currently running and will be updated before the discussion period ends.
>
> ```
> More recently, novel optimizers such as Muon, which introduce orthonormalized momentum matrices, have shown promising results in large-scale deep learning scenarios.
> ```
>
> > ### weakness 2
> > In terms of experiments, the reported test accuracies are surprisingly low, which is unexpected for methods that explicitly address client-drift. It would be helpful if the authors could provide an explanation—perhaps related to dataset characteristics, hyperparameter choices, or implementation details. Overall, the work is clearly presented and methodologically sound, but the empirical results and contextualization within the broader literature could be improved.
>
> The observed performance drop when distributing datasets in highly heterogeneous ways under FL (especially under $\alpha \leq 0.1$) is a well-known phenomenon. Although client-drift mitigation techniques, including adaptive optimizers, can alleviate some negative effects of heterogeneity, there remains an inherent performance gap compared to centralized single-machine training. For example, [1] also report similar accuracy values that are notably lower than centralized baselines. This disparity arises largely due to non-iid data distributions and communication frequency constraints.
>
> [1] Reddi, Sashank J., et al. "Adaptive Federated Optimization." International Conference on Learning Representations.

---

### Official Review · Reviewer_rhHF · 2025-11-08

**Soundness:** 3
**Presentation:** 3
**Contribution:** 3
**Rating:** 6
**Confidence:** 4

**Summary:**

The paper introduces FAdamGC, an adaptive gradient correction optimization algorithm that facilitates drift compensation using the Adam algorithm at the clients to address data heterogeneity. The algorithm utilizes a pre-estimation correction term based on Adam's moments. The theoretical analysis shows that FAdamGC provides better convergence guarantees than SGD-based corrective algorithms like SCAFFOLD under milder assumptions. The method achieves linear speedup convergence to stationary points. FAdamGC outperforms existing baselines empirically, although it requires additional communication costs. The paper also presents a selective tracking mechanism that chooses a subset of participating clients for updating correction terms, thereby reducing communication overhead.

**Strengths:**

The paper provides sufficient motivation for all the ideations. It justifies using Adam at the client level because FedAdam is more sensitive to gradient noise. The paper also explains why naively adapting a SCAFFOLD-like strategy for client drift mitigation in adaptive algorithms yields suboptimal results, and it carefully constructs a control variate strategy for the Adam optimiser.

The paper proposes a drift compensation method for the Adam optimizer used by clients in federated learning, based on a fixed-point structure, which brings a new perpective to improve the horizontal FL.

The algorithm incorporates a drift compensation term by averaging the gradient of a client over local rounds and utilises this correction to construct a data heterogeneity-robust algorithm. The paper demonstrates that their algorithm is more robust compared to the naïve approach, as it converges with fewer global communication rounds. The analysis also supports that under milder assumptions, the algorithm achieves the convergence rate for general non-convex federated learning objectives.

The paper is well-written, with clearly stated assumptions, detailed notations and results, and a well-articulated motivation for the research. The theorems and results were systematically laid out.

**Weaknesses:**

The paper noted that drift compensations should intuitively include the fixed-point structure needed for consistent convergence across different clients. However, the fixed-point structure discussed relates to a single client, since the update rule used for fixed-point problem formulation corresponds to each client's local optimizer. A discussion on how this fixed-point structure is maintained in the multi-node FL optimization problem would clarify how this intuition applies to FL, where multiple local updates and clients are involved.

**Questions:**

In Algorithm 1, $y^{(1)}$ and $y_i^{(1)}$ are not assigned any initial value. Can you recheck and justify?

Practically, $y_i^{(t)}$ is the local rounds' average of batch gradients for the client. Then, $y^{(t+1)}$ is computed using the aggregate of the differences between $y_i^{(t+1)}$ and $y_i^{(t)}$. How does this serve as a proxy for the theoretical pre-estimation correction term \nabla $f(x^\ast)-\nabla f_i(x^\ast)$?

The performance of FAdamGC is claimed to be superior to existing algorithms, and the experiment setup is explained in detail. However, the experiments are conducted by setting a target accuracy for each dataset, and then the performance of the algorithms is measured based on the total number of global rounds and the simulated runtime (in minutes) required to reach that target accuracy. Why does the paper not compare training loss and test accuracy in the main text, which is standard?

Figure 1 (3) shows that the gradient correction enables updates to move toward the global objective, but I do not understand how this demonstrates that client drift is being mitigated. Could you please elaborate?

In Theorem 5.2, the proof uses $\beta_2=0$ compared to $\beta_1=0$ in the proof of FedAdam, and there is no restriction on momentum terms in FedAMS. How are the assumptions considered milder, only because the gradient boundedness assumption is not needed in the proof if $\beta_2=0$? In that case, for FedAdam and FedAMS, do those algorithms also not require a gradient bound for convergence if we take $\beta_2=0$ in their analysis?

I would like to understand why the substitution $\tilde{m}$ is valid in equation (16).

$c^{(k)}$ used in the proof of Theorem 5.2; is it $c^{(k,0)}$ as used in the proof of Theorem 5.4?

How is the expectation dropped in the first term of equation (92), and then how is it resolved to the square root of a squared term in equation (94) if the term should have an expectation there?

Better remove extra equation numbers for every line while formatting the appendix.

In [1], it is discussed that proper tuning of local and global step sizes in the implementation of FedAdam mitigates the effect of data heterogeneity but does not eliminate it. How does learning rate tuning for FAdamGC affect the performance?

[1] Sashank Reddi, Zachary Charles, Manzil Zaheer, Zachary Garrett, Keith Rush, Jakub Konĕcnỳ, Sanjiv Kumar, and H Brendan McMahan. Adaptive federated optimization. arXiv preprint arXiv:2003.00295, 2020.

---

> ### Author Response · Authors · 2025-11-19
> **responses (1/3)**
>
> We sincerely thank the reviewer for carefully reading our work and providing important guidance.
>
> > ### weakness 1
> > The paper noted that drift compensations should intuitively include the fixed-point structure needed for consistent convergence across different clients. However, the fixed-point structure discussed relates to a single client, since the update rule used for fixed-point problem formulation corresponds to each client's local optimizer. A discussion on how this fixed-point structure is maintained in the multi-node FL optimization problem would clarify how this intuition applies to FL, where multiple local updates and clients are involved.
>
> In FL with multiple local updates, variance around the fixed-point analysis scenario does occur due to the server not aggregating after each local update. However, these variances are generally controllable through appropriate choices of local step sizes at each client (in theory through $\eta_l =\mathcal{O}(1/KL)$). As for concerns regarding multiple clients in the system, this analysis can be applied to any node since the interaction rules between each individual node and the server are the same. The fixed-point analysis mainly serves as an intuitive guide for algorithm design, ensuring that the effect of data heterogeneity is properly accounted for when studying the theoretical behavior of the method. We have added additional discussion to the revised manuscript.
>
> ```
> In settings with multiple local updates $(K > 1)$, variance around the fixed-point scenario naturally arises because the server does not aggregate after every step. However, these deviations remain controllable under appropriate choices of local step sizes. Thus the fixed-point analysis serves as an intuitive guide: it highlights the structural conditions under which client drift is minimized and explains why the proposed pre-estimation correction provides robustness against data heterogeneity in realistic multi-step FL training.
> ```
>
> > ### question 1
> > In Algorithm 1, and are not assigned any initial value. Can you recheck and justify?
>
> Thank you for pointing out this mistake, we have added the intialization $y_i^{(1)} = \nabla f_i(x^{(1)})$, $y^{(1)} = \nabla f(x^{(1)})$ in the revised manuscript.
>
> > ### question 2
> > Practically, $y_i^t$ is the local rounds' average of batch gradients for the client. Then, $y^{t+1}$ is computed using the aggregate of the differences between
> $y_i^{t+1}$ and $y_i^t$. How does this serve as a proxy for the theoretical pre-estimation correction term ?
>
> The design of $y_i^{(t)}$ and $y^{(t)}$ aim to represent local  and global gradients computed on the local device and across the whole network. While we cannot compute the true theoretical correction term $\nabla f(x^*) - \nabla f_i(x^\*)$ (since $x^\*$ is unknown), using these aggregate differences provides a practical and tractable proxy that approximates the same effect, allowing the correction mechanism to track the gradient difference across the whole system and individual clinets.
>
> > ### question 3
> > The performance of FAdamGC is claimed to be superior to existing algorithms, and the experiment setup is explained in detail. However, the experiments are conducted by setting a target accuracy for each dataset, and then the performance of the algorithms is measured based on the total number of global rounds and the simulated runtime (in minutes) required to reach that target accuracy. Why does the paper not compare training loss and test accuracy in the main text, which is standard?
>
> While it is true that training loss and test accuracy are standard metrics reported in many works, we chose to focus on the total communication rounds and simulated runtime to reach target accuracy because these metrics capture more directly the practical efficiency of adaptive optimization algorithms in distributed settings.
>
> Adam-based optimizers are known to converge faster and more stably compared to vanilla methods. Reporting the global rounds and wall-clock time to reach a fixed accuracy implicitly captures this fast convergence ability, providing a clearer practical perspective on cost and efficiency in federated networks where communication is often the bottleneck.
>
> Nonetheless, to ensure completeness, we have included training loss and accuracy curves with mean and variance in the appendix (see Appendix F), and these confirm consistent improvements of FAdamGC over baselines. We hope this clarifies our metric selection strategy.

---

> ### Author Response · Authors · 2025-11-19
> **responses (2/3)**
>
> > ### question 4
> > Figure 1 (3) shows that the gradient correction enables updates to move toward the global objective, but I do not understand how this demonstrates that client drift is being mitigated. Could you please elaborate?
>
> In common approaches with correction, local updates on individual clients are driven by their respective local objectives, this means that parameter trajectories tend to diverge as each client adapts to its own data distribution. The system then relies on periodic global averaging to coalesce these trajectories, which causes client drift when multiple rounds of local updates are performed.
>
> The key insight demonstrated in our figure is that when gradient correction terms are introduced, they explicitly adjust local updates by taking into account the gap between the local and global objectives. As a result, the direction of parameter updates aligns more closely with the true global optimum (red point), rather than being pulled towards local optima (black point).
>
> > ### question 5
> > In Theorem 5.2, the proof uses $\beta_2 = 0$ compared to $\beta_1 = 0$ in the proof of FedAdam, and there is no restriction on momentum terms in FedAMS. How are the assumptions considered milder, only because the gradient boundedness assumption is not needed in the proof if $\beta_2 = 0$? In that case, for FedAdam and FedAMS, do those algorithms also not require a gradient bound for convergence if we take $\beta_2 = 0$ in their analysis?
>
> Algorithms with $\beta_2 = 0$ may not require a bounded gradient assumption for convergence. The key reason is that setting $\beta_2 = 0$ causes the algorithm to employ simple gradient normalization rather than second-moment estimation. This normalization prevents the accumulation of large squared gradients, which can lead to unbounded growth if second-moment updates is used without a gradient bound. Removing the gradient boundedness assumption is considered milder because, in its presence, the impact of data heterogeneity $B$ can no longer be properly derived or analyzed. As a result, the theoretical effectiveness of adaptive methods against data heterogeneity cannot be demonstrated when a gradient bound is imposed. By relaxing this assumption, we can show the benefits of gradient correction in adaptive methods (see the comparison between Theorem 5.2 and Theorem C.3). Nonetheless, the same analysis approach should generalize to other algorithms such as FedAdam and FedAMS.
>
> > ### question 6
> > I would like to understand why the substitution $\tilde{m}$ is valid in equation (16).
>
> The substitution $\tilde{m}$ is valid in equation (16) because $\tilde{m}$ is defined as the deterministic counterpart of $m$, it replaces all stochastic gradients in $m$ with their deterministic values, as specified in Equation (7). In equation (15), $m$ appears within an inner product rather than a squared norm. Therefore, any stochastic component in $m$ contributes only additive noise to this term. By Assumption 5.1, this noise has zero expectation. Thus, for the purpose of expectation analysis, $m$ can be substituted by $\tilde{m}$ without affecting the result.
>
> > ### question 7
> > $c^{k}$ used in the proof of Theorem 5.2; is it  $c^{k,0}$ as used in the proof of Theorem 5.4?
>
> Thank you for pointing out the missing explanation, the proper definition of these terms are currently in Appendix D in FA-NT's proof, where $c^k$ as the sum of all moving average coefficients to compute the first order moment $m_i^{(t,k)}$: $c^k =  \sum_{k'=1}^k c^{(k,k')}$, we have added additional definition to the beginning of the proof for Theorem 5.2 and 5.4.
>
> > ### question 8
> > How is the expectation dropped in the first term of equation (92), and then how is it resolved to the square root of a squared term in equation (94) if the term should have an expectation there?
>
> Thank you for highlighting this oversight. You are correct, the expectation should not be dropped in the first term of equation (92). We have corrected this in the revision by explicitly including the expectation in the proof.

---

> ### Author Response · Authors · 2025-11-19
> **responses (3/3)**
>
> > ### question 9
> > Better remove extra equation numbers for every line while formatting the appendix.
>
> Thank you for your suggestion. We have removed the extra equation numbers for clear referencing.
>
> > ### question 10
> > In [1], it is discussed that proper tuning of local and global step sizes in the implementation of FedAdam mitigates the effect of data heterogeneity but does not eliminate it. How does learning rate tuning for FAdamGC affect the performance?
>
> We observe that FAdamGC exhibits similar sensitivity to learning rate tuning as other FL methods employing local Adam optimizers, such as LocalAdam. While the choice of step sizes can impact overall training stability and convergence speed, our gradient correction-based adaptive FL method consistently outperforms baselines across a wide range of tuning settings. This suggests that although careful learning rate selection remains important, the inherent robustness of the gradient correction mechanism in FAdamGC effectively mitigates adverse effects from data heterogeneity better than competing algorithms.

---

### Meta-Review · Area_Chair_J8W8 · 2026-01-06

**Summary:**

The main point of contention raised by was comparision to [a] raised by Gf9b and and echoed by k38c. While the other three reviewers recommended a weak accept, this was before they considered [a]. The main idea proposed here -  of correcting gradients before moment estimation - has already been proposed in [a]. In fact, setting $\beta_2=0$ in Adam from current paper recovers the gradient normalization technique of [a]. Further, the theoretical novelty also seems to be overstated as pointed out by k38c. Finally, as k38c points out, since $\beta_2 > 0$ was the main novelty of this work, the paper should have been solely focused on justifying and understanding that.


[a] Yan, W., Zhang, K., Wang, X., & Cao, X. (2025). Problem-Parameter-Free Federated Learning. In The Thirteenth International Conference on Learning Representations.

**Reviewer Concerns:**

- Reviewers rhHF
  - Asked a lot of minor clarifications regarding proofs and algorithm descriptions that were largely addressed by authors' response.

- Reviewer t86o
  - Absolute test accuracy scores are surprisingly low - response attributed to dataset difficulty and highlighted significant relative improvement.
  - Comparision to related work - added in revision



- Reviewer k38c
  - Double communication cost (transmitting model parameters + correction terms) - response states it is mitigated by selective tracking and justified by performance gains.
  - Justification and intution for why $\beta_2 > 0$ is important - added some experiments showing $\beta_2 > 0$ outperforms $\beta_2 0 0$, however they don't demonstrate a clear mechanism for why this occurs.
  - Missing baselines (specifically Yan et al.) - response added Table 3 comparison showing FAdamGC outperforms Yan et al.


- Reviewer Gf9b
  - Pre-estimation correction lacks novelty and overlaps with Yan et al. (2025) - response acknowledged similarity for beta2=0 special case but emphasized generalization to Adam (beta2 > 0).
  - Bounded-gradient-free convergence proof only holds for known special case (beta2=0) - response clarified general case relies on standard assumptions but offers better empirical performance.
  - Selective tracking viewed as minor engineering detail - response argued it is a critical system-level optimization for communication efficiency.


- Reviewers XP38
  - rigorously define system model (empirical risk/distribution) - addressed
  - Request for explanation on why SGD works with naive correction despite heterogeneity - response explained that ideal correction effectively cancels local gradient bias at the optimum.
  - comparision against momentum-based gradient correction estimation - response acknowledged suggestion and promised to include the results later.
  - mean/std in plots - added in Fig 10.

**Reviewer Scores:**

Reviewers Gf9b and k38c would have likely maintained their low scores, while all other reviewers once took notice of [a] would likely have **decreased** their scores.

---

### Decision · Program_Chairs · 2026-01-26

Reject